# Prediction of cellular morphology changes under perturbations with a transcriptome-guided diffusion model

Xuesong Wang [1,2,5], Yimin Fan [1,2,5], Yucheng Guo [1,5], Chenghao Fu [2], Kinhei Lee [2], Khachatur Dallakyan [2], Yaxuan Li [2], Qijin Yin [1], Yu Li [2,3] ✉ & Le Song [1,4] ✉

Investigating cell morphology changes after perturbations using high-throughput image-based profiling is increasingly important for phenotypic drug discovery, including predicting mechanisms of action (MOA) and compound bioactivity. The vast space of chemical and genetic perturbations makes it impractical to explore all possibilities using conventional methods. Here we propose MorphDiff, a transcriptome-guided latent diffusion model that simulates high-fidelity cell morphological responses to perturbations. We demonstrate MorphDiff's effectiveness on three large-scale datasets, including two drug perturbation and one genetic perturbation dataset, covering thousands of perturbations. Extensive benchmarking shows MorphDiff accurately predicts cell morphological changes under unseen perturbations. Additionally, MorphDiff enhances MOA retrieval, achieving an accuracy comparable to ground-truth morphology and outperforming baseline methods by 16.9% and 8.0%, respectively. This work highlights MorphDiff's potential to accelerate phenotypic screening and improve MOA identification, making it a powerful tool in drug discovery.

Characterizing and predicting cell states under genetic and drug perturbations remains one of the most challenging and meaningful directions in the single-cell biology domain. By understanding how individual cells respond to specific genetic and drug perturbations, researchers can better map the pathways that contribute to particular cell states and identify potential targets for therapeutic intervention. Multiple aspects of cell states may change under perturbations, including but not limited to gene expression profile, proteomic composition, metabolic status, and cell morphology. Among the various modalities of cell states, the investigation of cell morphology responses to perturbations through high-throughput image-based profiling has gained significant interest due to its wide applications in phenotypic drug discovery. These applications encompass the prediction of MOAs (Mechanism of Actions), the prediction of compound bioactivity, and drug repurposing[1].

However, considering the vast number of synthesizable chemical compounds and genes in the perturbation space, it is infeasible to profile the cell morphology for all possible perturbations. Therefore, developing an in-silico method to simulate cell morphological responses to perturbations emerges as a challenging and meaningful topic[2]. With a powerful and reliable tool to infer the cell morphology change under perturbations, the exploration of the vast perturbation space can be significantly accelerated, further promoting downstream drug discovery pipelines, such as MOA prediction. Computational approaches have been proposed to predict the cell morphology[3,4]. However, the precision and fidelity of these methods remain inadequate. First, as indicated by existing work IMPA (IMage Perturbation Autoencoder)[3], prediction on unseen perturbations will only perform well if the model has seen a similar perturbation with a similar effect in

[1]BioMap Research, Palo Alto, CA, USA. [2]Department of Computer Science and Engineering, The Chinese University of Hong Kong, Hong Kong SAR, China. [3]The Chinese University of Hong Kong Shenzhen Research Institute, Shenzhen, China. [4]Mohamed bin Zayed University of Artificial Intelligence, Abu Dhabi, UAE. [5]These authors contributed equally: Xuesong Wang, Yimin Fan, Yucheng Guo. ✉e-mail: yuli@cuhk.edu.hk; le.song@mbzuai.ac.ae

the training dataset. As structurally similar drugs or co-expression genetic perturbations may have different effects[5], it is critical to faithfully encode and represent perturbations to improve the generalizability of prediction in unseen perturbations. Second, in addition to the perturbations, cellular morphology is influenced by a wide range of factors, including but not limited to batch and well position effects[6], which leads to a high noise level in the cell morphology data.

Therefore, we developed MorphDiff, a scalable transcriptome-guided diffusion model for predicting cell morphology responses to unseen perturbations. MorphDiff predicts the perturbed cell morphology with the perturbed L1000 gene expression profile as the condition. The following motivations inspire the choice of using gene expression profiles as the condition. First, gene expression plays a crucial role in determining cell morphology by directing the synthesis of proteins that regulate cellular structure and dynamics[7,8]. Although the relationship between cell morphology and gene expression is complex, there remains shared and complementary information between them[9], making the prediction between gene expression and cell morphology feasible[9–12]. Therefore, perturbed gene expression is more informative in determining cell morphology than the prior knowledge-guided encoding of the perturbation, such as drug SMILES (Simplified Molecular Input Line Entry Specification) representation and Gene2vec[13] embedding. Second, the L1000 assay offers a much larger pool of publicly available datasets compared to cell morphology profiling[10]. Thus, obtaining the gene expression of cells treated with unseen perturbations is much easier than obtaining the cell morphology, enabling our tool to be potentially applicable in broader scenarios.

The architecture of MorphDiff is based on the Latent Diffusion Model[14], an advanced generative model inspired by thermodynamics that learns to reverse the data from an unordered state to an ordered state. Intuitively, by learning to recursively add Gaussian distributions to noisy images, diffusion models could generate high-quality images. Concretely, we utilize the Morphology Variational Autoencoder (MVAE, see Methods Section) to compress the high-dimensional morphology of cells into low-dimensional embeddings, and then train a latent diffusion model[14] using these embeddings with gene expression profiles as the condition. Compared to traditional generative models such as the Generative Adversarial Network (GAN)[15] used in MorphNet[4] and IMPA[3], the diffusion-based architecture has the following advantages. First, the latent diffusion model is highly robust to noise, as, by design, diffusion models transform data into a noisy state and learn to reconstruct the original data, which makes it highly suitable for cell morphology datasets. Second, the latent diffusion model supports flexible conditioning, further enhancing the utility of MorphDiff. In addition to relying solely on the perturbed transcriptome as input, the pre-trained MorphDiff model can also take an unperturbed cell morphology as input while using the perturbed cell transcriptome as the condition. This allows MorphDiff to infer the continuous transition of cell morphology from unperturbed to perturbed without requiring additional training. Third, diffusion models usually perform better on general image synthesis tasks than GAN-based generative models[16].

To evaluate MorphDiff's capability for predicting cell morphology responses to novel perturbations, we established a comprehensive assessment framework. Our methodology tested the model's performance across three extensive datasets encompassing a wide range of genetic and drug perturbations. We assessed the model using both standard image generation metrics and measures of biological relevance by analyzing interpretable morphological features extracted through established computational tools such as CellProfiler[17] and DeepProfiler[18]. Furthermore, we examined MorphDiff's effectiveness in capturing correlations between transcriptional and morphological responses to perturbations, potentially providing insights into how changes in gene expression manifest as alterations in cellular morphology.

A key application we aim to investigate is MorphDiff's potential in phenotypic drug discovery, particularly for MOAs retrieval. While traditional approaches focus on drug structure analysis and transcriptome response analysis[19,20], we emphasize that cell morphology information-whether directly observed or computationally inferred-provides complementary and valuable signals for MOA identification. Through our designed MOA retrieval pipeline, we demonstrated that MorphDiff-generated morphologies achieve comparable performance to ground-truth morphologies and outperform existing baseline methods and gene expression-based approaches. Furthermore, we explored the potential of our approach to discover drugs with different molecular structures but similar MOA. In summary, MorphDiff is a powerful tool in phenotypic drug discovery by accurately generating cell morphology under unseen perturbations, with promising applications in facilitating the exploration of the vast phenotypic perturbation screening space and assisting in determining the MOAs of structurally diverse drugs.

## Results
### Method overview
MorphDiff maps L1000 gene expression to cell morphology images through a Latent Diffusion Framework[14]. Concretely, as shown in Fig. 1a, paired L1000 gene expression and cell morphology images are curated for the same perturbations. The cell morphology images are acquired by the Cell Painting[21] platform and are usually composed of five channels, including DNA, ER, RNA, AGP, and Mito (Supplementary Table 1). MorphDiff is made up of two main components (Fig. 1b), Morphology VAE (MVAE) and Latent Diffusion Model (LDM). MVAE consists of the encoder and decoder parts. The encoder takes the high-dimensional cell morphology images composed of five channels as input and outputs the low-dimensional latent representations, while the decoder reconstructs the original input images based on the latent representation. This step compresses the cell morphology images into meaningful low-dimensional representation so that training the diffusion models can be easier. The latent diffusion model is trained to generate cell morphology low-dimensional representation conditioned on the perturbed gene expression profiles. The Latent Diffusion Model (LDM) consists of the noising process and denoising processes. In the noising process, Gaussian random noise is sequentially added to the latent representation of morphology for multiple steps, from 0 to $T$. The final $\mathbf{Z_T}$ follows the standard Gaussian distribution. In the denoising process, in each step $t$, LDM is trained to recursively remove the noise from $\mathbf{Z_t}$ conditioned on the L1000 gene expression as $t$ decreases from $T$ to 0. LDM is implemented with denoising U-Net architecture (based on convolutional neural networks) augmented with an attention mechanism[22]. As the input condition, L1000 gene expression is combined with the model parameters of LDM, specifically the key and value of the attention mechanism in LDM. Training an LDM minimizes the variational upper bound, which serves as a proxy for reducing noise prediction error and ensures the generated samples match the ground-truth distribution. More technical details are described in the MorphDiff Model section. As illustrated in Fig. 1c, pre-trained MorphDiff can be applied in two ways. In the first mode (referred to as MorphDiff(G2I)), it takes the L1000 gene expression as the condition and denoises the corresponding cell morphology images from random noise distribution. In the second mode (referred to as MorphDiff(I2I)), it takes the L1000 gene expression for one specific perturbation as the condition and transforms the morphology images from the control cell morphology to the predicted perturbed morphology images. Compared to previous works in generating cell morphology images, MorphDiff is the only tool that supports generation from gene expression to morphology and transformation from unperturbed morphology to perturbed morphology. In Fig. 1d, we outlined the key applications of MorphDiff in practice. Primarily, MorphDiff can predict cell morphology resulting from unseen perturbations not encountered during training, which fulfills our main objective of exploring the vast perturbation space through in-silico inference. Furthermore, our framework incorporates CellProfiler[17] and

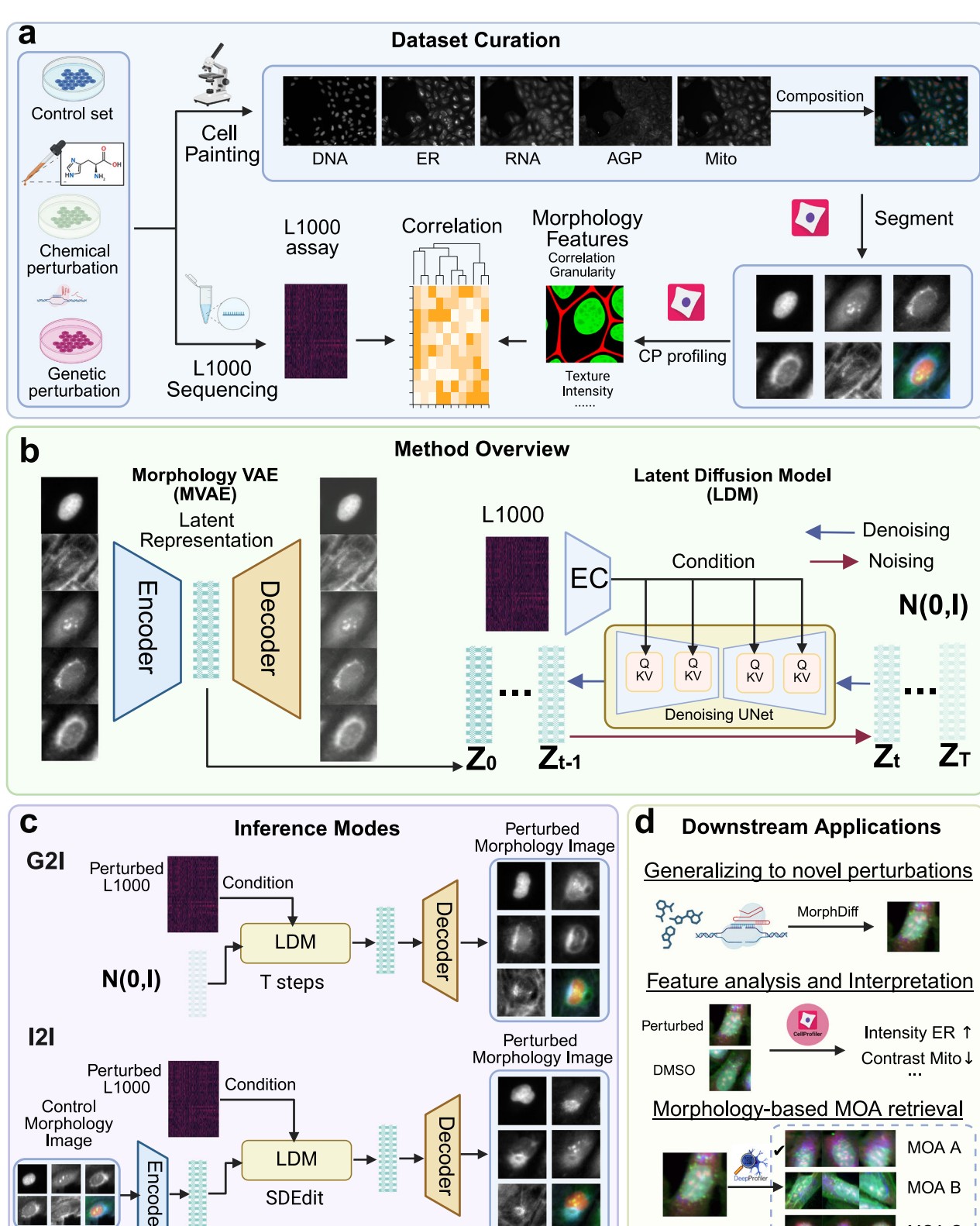

DeepProfiler[18] to generate biologically meaningful morphological features and embeddings, respectively. This integration enables researchers to probe specific morphological feature changes following perturbation, thereby enhancing interpretability and practical applicability. As a critical application, the generated morphological profiles can be utilized for retrieval and identification of drug MOA, establishing MorphDiff as a powerful tool in phenotypic drug discovery.

For model evaluation and applications, we collected three large-scale cell morphology image datasets for model training, evaluation, and analysis, which include 1028 drug perturbations from the CDRP dataset[23], 130 genetic perturbations from the JUMP dataset[24] on the U2OS cell line, and 61 drug perturbations from the LINCS dataset on the A549 cell line[25]. After pre-processing and splitting these datasets, we obtained the following datasets for training and evaluation: JUMP

**Fig. 1 | Overview of the MorphDiff framework. a** In this study, the multimodal dataset consists of the morphology images with five channels collected using CP (Cell Painting), and the gene expressions collected using L1000 assays. CellProfiler is then used to segment the cells and extract CellProfiler features at the single-cell level. Morphology images and gene expression together characterize the cell morphology responses to specific perturbations. The scale bar is 20 μm. **b** MorphDiff is composed of two main components: Morphology VAE (MVAE) and Latent Diffusion Model (LDM). The MVAE encoder encodes the multi-channel cell morphology images into latent representation, and the decoder reconstructs the original cell morphology images based on the latent representation. LDM is trained to denoise from Gaussian random noise $Z_T$ to morphology latent representation $Z_0$, recursively conditioned on L1000 gene expression. The scale bar is 20 μm. **c** MorphDiff can be applied in two ways to generate cell morphology images with perturbations: L1000 gene expression to cell morphology generation (G2I, Gene to Image) and perturbed L1000 gene expression combined with control morphology images to perturbed cell morphology images generation (I2I, Image To Image). I2I is implemented with SDEdit[62] without re-training. The scale bar is 20 μm. **d** Illustration of the downstream applications of MorphDiff, including prediction on unseen perturbations, feature analysis with CellProfiler, as well as morphology-based MOA retrieval with DeepProfiler. DMSO stands for Dimethyl sulfoxide, which is considered as control group without perturbation. The scale bar is 20 μm. **a**, **b**, **c**, **d** are created in BioRender. Group, A. (2025) https://BioRender.com/nu1zlqw.

Training set, JUMP ID (in-distribution) set, and JUMP OOD (out-of-distribution) set; the CDRP Training set, CDRP ID set, CDRP OOD set, and CDRP Target_MOA set; the LINCS Target leave-one-out set and LINCS MOA leave-one-out set. Details of the dataset splits are provided in the Datasets section.

For benchmarking and downstream tasks, we generated gene embeddings and drug embeddings as conditions for baseline methods, respectively, for genetic perturbation and drug perturbation prediction tasks, following the implementation described in IMPA[3]. The gene embeddings are generated with Gene2vec[13], and drug embeddings are generated with RDKit[26].

## MorphDiff accurately predicts changes in cell morphology with genetic perturbations

To illustrate the superiority of MorphDiff, we benchmarked the general generative performance on the JUMP OOD dataset. We compared MorphDiff with a wide range of baseline methods, including MorphNet[4], DMIT (Disentanglement for Multi-mapping Image-to-Image Translation)[27], DRIT++ (Disentangled Representation for Image-to-Image Translation)[28], StarGANv1[29], IMPA[3], VQGAN (Vector Quantized Generative Adversarial Network)[30], and MDTv2 (Masked Diffusion Transformers)[31]. For some baseline methods, including IMPA, DRIT++, DMIT, StarGANv1, VQGAN, and MDTv2, we followed the implementation in IMPA[3], using gene embeddings (Gene2vec[13]) as conditions. In contrast, MorphNet[4] generates morphology images based on gene expression profiles that have been corrected using scVI[32]. The details of these methods can be found in Supplementary Notes 1.

First, we conducted a visual assessment of the generation quality between MorphDiff methods and baseline approaches, as illustrated in Fig. 2a. For this visual comparison, we presented results from the *SREBF1* genetic perturbation, which represents an out-of-distribution (OOD) perturbation. Visualizations for the additional nine OOD genetic perturbations are available in Supplementary Figs. 1–3. The specific criteria used for selecting these images for visualization are detailed in Supplementary Notes 2. The generated outputs of both modes of MorphDiff display colors and morphological structures that closely resemble the ground truth. DMIT, IMPA, and MDTv2 produce reasonably good images that adequately capture the general morphological features, but they fall short in detail and clarity. MorphNet shows promise in generating reasonable images with some detail, but it lacks diversity. StarGAN produces diverse outputs with good morphological structures, but it exhibits subtle color bias compared to the ground truth. For DRIT++ and VQGAN, the visual quality appears worse than other methods, with either reduced clarity or noticeable color bias. Overall, MorphDiff (I2I), MorphDiff (G2I), IMPA, and DMIT demonstrate the highest image generation quality, with both MorphDiff variants producing images that closely resemble the ground truth.

Quantitatively, we evaluated these methods using several metrics, including FID[33], IS (Inception Score)[34], CMMD[35], as well as density and coverage[36]. FID measures the distance between the distributions of the ground-truth and generated images in the feature space, using features extracted from Inception V3[34]. The IS assesses the quality of the generated images based on how effectively they can be classified by the Inception V3 model. Both FID and IS utilize Inception V3[34], which is pre-trained on ImageNet[37]. To enhance the reliability and robustness of our comparisons, we incorporated CMMD, which calculates the Maximum Mean Discrepancy (MMD) distance based on features extracted by a CLIP model pre-trained on 400 million image-text pairs containing more complex scenes[35]. We also included density and coverage metrics, as these metrics are robust against outlier samples and could better measure the diversity and fidelity of the generated samples[36]. In general, lower FID and CMMD values indicate better performance, while higher density, IS, and coverage scores reflect better generation quality. For convenience of presentation, we used the reciprocals of FID and CMMD, denoted as FID(-1) and CMMD(-1), for comparison and analysis. To ensure statistical reliability and robustness of our results, we employed different sampling approaches based on method requirements. For methods requiring reference control DMSO morphology images as input (Supplementary Table 2), we generated outputs using 10 distinct groups of control images, each group derived from an independent plate ($n = 10$ biological replicates). Each plate represented a separate experimental unit, indicating that the statistical analysis derived from 10 independent biological replicates (10 distinct plates). For methods that do not rely on reference control DMSO images, we conducted 10 separate sampling iterations, each with a different random seed. We performed the one-sided Wilcoxon signed-rank test for each comparison with Bonferroni correction to further enhance the statistical significance of our tests. It should be noted that star ("∗") means that MorphDiff surpasses the baseline method with a *p* value <0.05. The "ns" represents the situation of no significant difference. The same notation for performance tests is used in the following sections. As depicted in Fig. 2b, MorphDiff (I2I) and MorphDiff (G2I) rank as the top two methods across these evaluation metrics compared with other baseline approaches, highlighting the effectiveness of our model. We also observed that all methods relying on reference control DMSO morphology images exhibit larger performance variance compared to those that do not, as the quality of the control cell morphology images may impact the model performance in terms of general evaluation metrics (Supplementary Notes 3). The results also show that MorphNet achieves a nearly zero coverage score, demonstrating that the generative output of MorphNet lacks diversity, which aligns with the visualization in Fig. 2a. Based on both qualitative visualizations and quantitative metrics, MorphDiff (both modes) demonstrates more superior performance in terms of generalization, fidelity, and diversity compared to other generative models, ultimately resulting in superior generation quality.

Furthermore, we performed an in-depth analysis of whether MorphDiff can accurately predict changes in cell morphology under genetic perturbations. We performed a quantitative analysis using the CellProfiler features to demonstrate the effectiveness of MorphDiff. CellProfiler features are a set of morphological features extracted with CellProfiler related to cell morphology, which measure the texture, intensity, and granularity within each morphology channel and the correlation between different morphology channels. In Fig. 2c, we used UMAP to project the morphological features into two dimensions

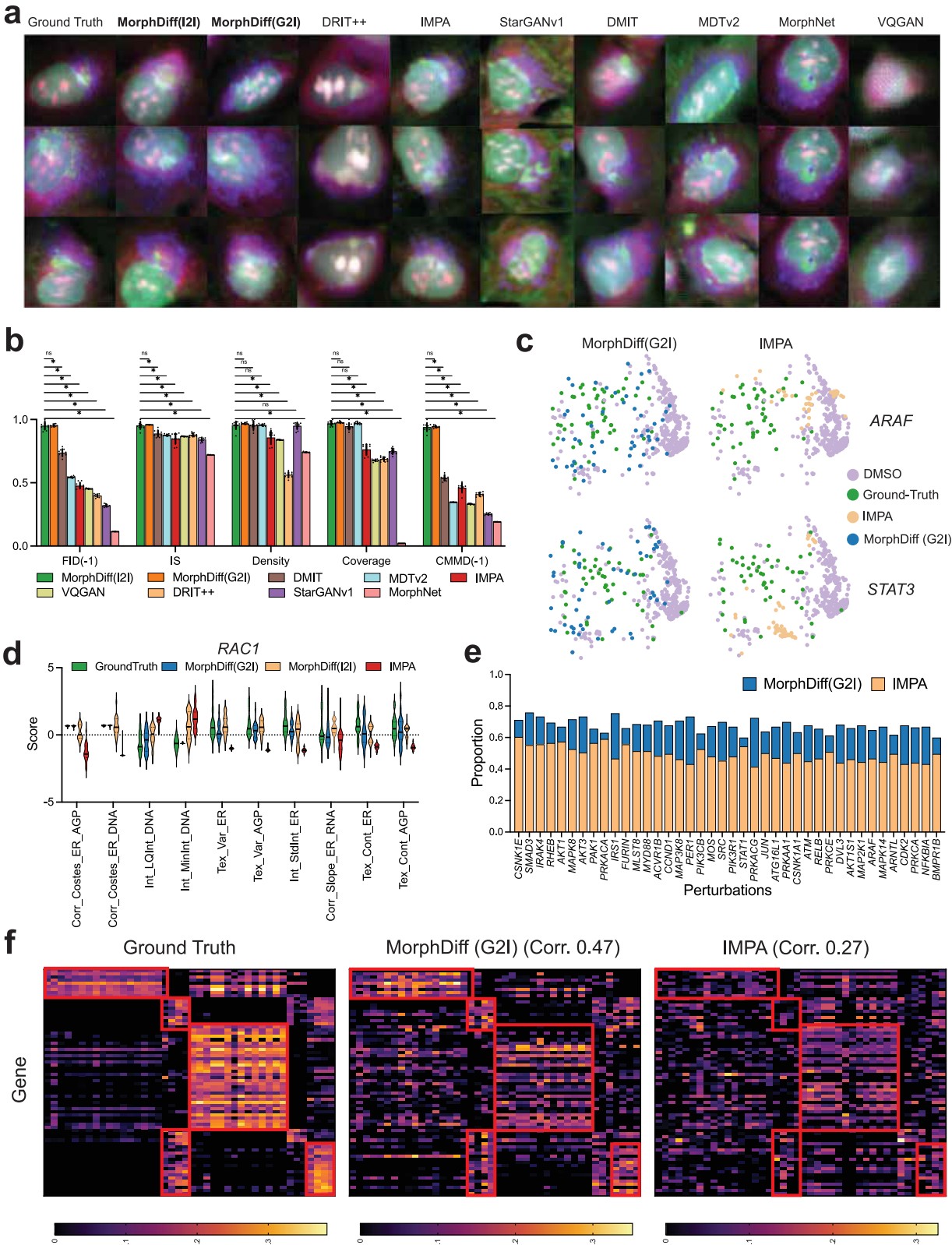

for visualization. We selected the 10 most important CellProfiler features discriminative of the perturbation types to form the morphological feature vector. The details of the feature selection strategy are discussed in the Evaluation Setup section (Morphological feature selection based on discriminability of perturbations). For *ARAF* and *STAT3* genetic perturbations, we can observe that the distribution of ground-truth perturbed cell morphology has an apparent variance.

The generated outputs of IMPA form a dense cluster and are far away from the ground-truth distribution. In contrast, the outputs of MorphDiff(G2I) are much more diverse and resemble the ground-truth distribution. We conducted the same analysis on other genetic perturbations, and the results in Supplementary Fig. 4 display a consistent effect. We also evaluated MorphDiff(I2I) using similar methods in Supplementary Fig. 5 and found a higher overlap between the ground-

**Fig. 2 | MorphDiff predicts changes in cell morphology with genetic perturbations. a** Visualization of the generated outputs from MorphDiff and baseline methods for the *SREBF1*. The corresponding information for the original channels and fluorophores can be found in Supplementary Table 1, and the weights of composition are detailed in the Evaluation Setup part of the Methods section. The scale bar is 20 μm. **b** Performance benchmarking on the JUMP OOD set. Each method was evaluated 10 times for robust assessment. Methods requiring reference control DMSO morphology as input used 10 distinct control image groups from independent plates (*n* = 10 biological replicates), while methods without this requirement used 10 sampling iterations with different random seeds. Linear normalization was conducted on the reciprocals of FID, CMMD, and on the other three metrics separately to convert their values to the range of 0–1. A larger value indicates better performance. One-sided Wilcoxon signed-rank test with Bonferroni correction was conducted between MorphDiff(I2I) and other methods. "*" indicates that MorphDiff(I2I) surpasses the baseline method with a *p* value <0.05. The "ns"

represents the situation of no significant difference. Data are presented as mean values ± SD. **c** UMAP visualization of CellProfiler features for IMPA and MorphDiff(G2I) under two genetic perturbations. DMSO (Dimethyl sulfoxide) serves as the untreated control. **d** CellProfiler feature distributions comparing ground truth with IMPA, MorphDiff(G2I), and MorphDiff(I2I) outputs for *RAC1* perturbation. Abbreviations: E ERSyto, EB ERSytoBleed, H Hoechst, Pg Ph_golgi, LQ lower quartile, Var variance, Tex texture, Corr correlation, Int intensity. **e** The two-sided Wilcoxon signed-rank test results to test the difference between the generated and ground-truth CellProfiler features. The *y*-axis indicates the proportion of generated CellProfiler features not significantly differ from the ground-truth CellProfiler features through a Wilcoxon signed-rank test (*p* value >0.05). 232 features are tested in total. **f** The heatmap of the correlation between the CellProfiler features of the ground-truth morphology, MorphDiff(G2I)-generated morphology, and IMPA-generated morphology with L1000 gene expression. Source data are provided as a Source data file for (**b**–**f**).

truth feature distribution and the MorphDiff (I2I) generated feature distribution.

Then, we turned to a more fine-grained analysis of how well the generated morphology aligns with the ground-truth morphology regarding the values of individual CellProfiler features. In Fig. 2d, we visualized the values of the 10 most important cell morphology features from the ground-truth perturbation morphology, the morphology generated by MorphDiff(G2I), MorphDiff(I2I), and the morphology generated by IMPA on *RAC1* genetic perturbation. We can see that the MorphDiff-generated morphology aligns well with the ground-truth morphology compared with the IMPA-generated morphology. The results of more perturbations on the top 20 most important CellProfiler features are shown in Supplementary Figs. 6 and 7. To have a more holistic view of how closely the MorphDiff-generated morphology aligns with the ground-truth distribution, we performed statistical tests for a more quantitative comparison. Concretely, for each feature out of all CellProfiler features extracted, we conducted the two-sided Wilcoxon signed-rank test across the CellProfiler features of the ground-truth and the generated output. If the *p* value of the Wilcoxon signed-rank test is larger than 0.05, we conclude that insufficient evidence shows significant differences between the generated perturbation morphological features and the ground-truth perturbation morphological features. As shown in Fig. 2e, the proportion of the generated CellProfiler features that align with the ground-truth features of MorphDiff(G2I) is much larger than IMPA. For MorphDiff(G2I), more than seventy percent of the generated CellProfiler morphology features align with the ground-truth perturbation CellProfiler features, which demonstrates the effectiveness of MorphDiff in accurately predicting the perturbation morphology. We also repeated the analysis in Fig. 2e with other types of statistical tests and metrics, including t-test, Kolmogorov-Smirnov test, and KL-divergence, in Supplementary Fig. 8 for MorphDiff(G2I), MorphDiff(I2I), and IMPA to further demonstrate the robustness and consistency of our conclusion.

Subsequently, we validated the effectiveness of MorphDiff in capturing the correlation between perturbed gene expression and perturbed morphology. For better clarity, we selected the CellProfiler features that MorphDiff(G2I) and IMPA can best recover. The details of feature selection are described in the Evaluation Setup Section (Morphological feature selection based on predictability). Figure 2f shows the correlation between gene expression and CellProfiler features of the ground truth, MorphDiff(G2I), and IMPA, respectively. In Fig. 2f, we observed that several groups of genes and CellProfiler features have a high correlation with each other. For example, the group on the top left of the heatmap mainly consists of features related to Correlation and Intensity associated with ER, Mito, and AGP channels (Supplementary Fig. 9). Another group on the bottom right mainly consists of features related to the intensity of DNA and ER channels (Supplementary Fig. 9). The detailed correlation of MorphDiff(G2I) and IMPA can be found in Supplementary Figs. 10 and 11. We computed the

correlation between the correlation map of the generated samples and ground-truth samples. We observed that the samples generated with MorphDiff(G2I) have a much higher correlation (0.47) than the samples generated with IMPA (0.27). Therefore, MorphDiff(G2I) effectively captures the pattern of cell morphological feature changes across different genetic perturbations, while IMPA fails to capture the correlation pattern between gene expression and cell CellProfiler features. We performed a similar analysis for MorphDiff(I2I) and IMPA in Supplementary Fig. 12 and also validated the superiority of MorphDiff(I2I).

## MorphDiff captures morphological changes in cells across thousands of drug treatments

We first benchmarked all methods on the CDRP OOD set with the general evaluation metrics, in which the meanings of stars are the same as in the previous section, and the hashtag ('#') indicates the situation where the baseline performs significantly better. As shown in Fig. 3a, both modes of MorphDiff substantially outperform other methods in most cases. While DMIT and MDTv2 achieve high scores in IS, density, and coverage, they obtain relatively lower scores in FID(-1) and CMMD(-1) compared to MorphDiff (both modes). This suggests that DMIT and MDTv2 struggle to produce high-quality samples. Similar to the results on the JUMP dataset, MorphNet performs well in terms of density score, but it is nearly ineffective in terms of coverage, demonstrating its lack of generalizability and diversity. Ideally, an effective generative model should achieve low FID, CMMD, and high scores in IS, density, and coverage to ensure that it produces outputs of high quality and fidelity, while also generating diverse outputs that closely align with the ground-truth distribution. MorphDiff (both modes) outperforms baseline methods considering these criteria, demonstrating its comprehensive generative capability and stability. To further demonstrate the generalizability of our model, we performed evaluation in a stricter setting. We conducted leave-one-out validation on the LINCS dataset (A549 cell line) for 10 different targets and 10 different MOAs. We selected targets and MOAs with sufficient cell morphology images for a robust evaluation. The detailed validation setting can be found in the Datasets section (LINCS dataset). The results displayed in Supplementary Fig. 13 demonstrate that MorphDiff consistently achieves the most comprehensive generative performance for different MOAs and targets in the leave-one-out setting.

We further investigated the responses of cell morphology to small molecular compounds by validating whether the generated output aligns well with the ground truth on the CellProfiler features. We extracted the CellProfiler features from the generated morphologies and compared them with the features extracted from the ground-truth morphologies. Concretely, we extracted CellProfiler features from 3000 ground-truth images sampled from the CDRP ID set and from the corresponding images generated by MorphDiff(I2I), MorphDiff(G2I), and IMPA. Over 200 informative CellProfiler features are extracted and analyzed. We calculated $R^2$ scores between the ground-truth

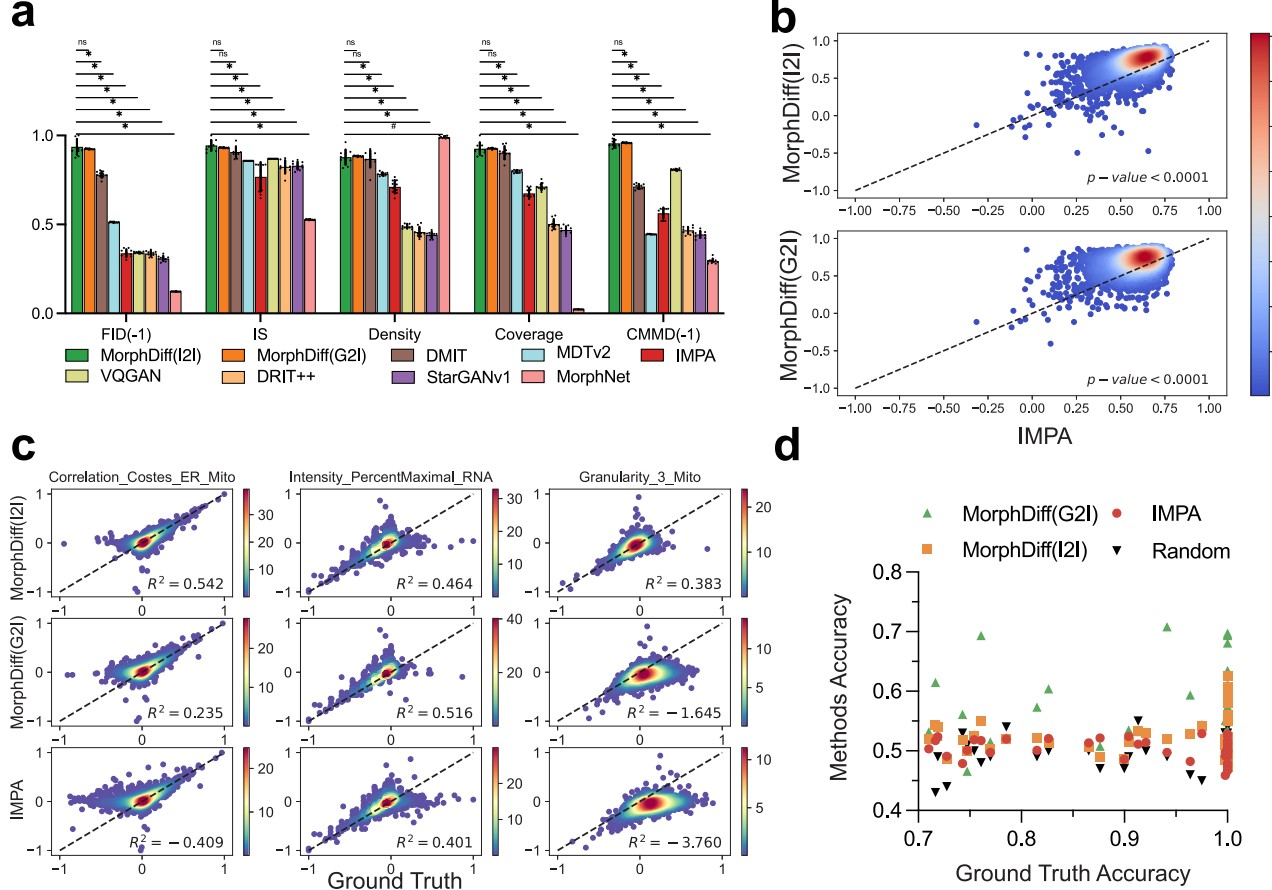

**Fig. 3 | MorphDiff captures morphological changes across thousands of drug treatments. a** General generative evaluation of pre-trained MorphDiff and IMPA. Methods requiring reference control images used 10 distinct control image groups from independent plates ($n = 10$ biological replicates), while other methods used 10 sampling iterations with different random seeds. Linear normalization converted FID and CMMD reciprocals, plus three other metrics, to 0–1 range. Statistical comparisons used one-sided Wilcoxon signed-rank tests with Bonferroni correction ($p$ value < 0.05). "*" indicates MorphDiff(I2I) superiority; "ns" indicates non-significance; hashtag ("#") indicates the baseline performs better significantly. Data are presented as mean values ± SD. **b** The $R^2$ score between the ground-truth CellProfiler feature vectors and the generated CellProfiler feature vectors on the CDRP ID set. The $x$-axis represents the $R^2$ scores between IMPA and ground truth for each sample. The $y$-axis represents the $R^2$ scores of MorphDiff(G2I) and

MorphDiff(I2I) against ground truth respectively. The $p$ values calculated by one-sided Wilcoxon signed-rank test indicate the significance of the distribution of the $y$-axis being greater than that of the $x$-axis. **c** The difference between control and perturbations for specific CellProfiler features on the CDRP ID set. The $x$-axis displays the difference between the ground-truth perturbations and the control, while the $y$-axis displays the difference between the generated perturbations and the control. The higher the point density along the diagonal, the closer the generated difference is to the ground truth. More results can be found in the Supplementary Fig. 14b, c. The $R^2$ score measuring the similarity between the predicted difference and ground-truth difference. **d** The $x$-axis represents the ground-truth classification accuracy scores between different pairs of perturbations, and the $y$-axis means the classification accuracy scores of the corresponding samples generated by different methods. Source data are provided as a Source data file for (**a–d**).

CellProfiler feature vectors and the corresponding generated CellProfiler feature vectors. A higher $R^2$ score indicates better proximity between the generated and the ground-truth CellProfiler feature vectors. We compared the $R^2$ score of MorphDiff versus ground truth ($y$-axis) and IMPA versus ground truth ($x$-axis) in Fig. 3b, with each point indicating a unique cell morphology. Figure 3b shows that MorphDiff(I2I) and MorphDiff(G2I) both outperform baselines significantly with a $p$ value < 0.0001 through the one-sided Wilcoxon signed-rank test. In more detail, 89% of the generated samples from MorphDiff(I2I) exhibit an $R^2$ score greater than 0.5, with 27.2% surpass 0.8. For MorphDiff(G2I), 87.6% of the samples have an $R^2$ score greater than 0.5, and 16.2% exceed 0.8. In contrast, 78.3% of the samples from IMPA exceed 0.5, but none reach 0.8. This indicates that MorphDiff (both modes) can well capture the feature distribution of the drug perturbation dataset. We conducted the same analysis for the CDRP OOD set (Supplementary Fig. 14a) and for the LINCS leave-one-out set (Supplementary Fig. 16a), and found that both modes of MorphDiff consistently outperform the baseline method with the $p$ value < 0.0001, demonstrating the generalizability of our method.

Furthermore, we performed a fine-grained analysis on the effectiveness of our tool in predicting the features that undergo the most significant changes after perturbation. For each CellProfiler feature, we conducted a chi-square test between the ground-truth control CellProfiler features and ground-truth perturbed CellProfiler features to determine which features change significantly after perturbation. We picked the features with $p$ value < 0.05 as significantly changed and calculated the $R^2$ score between the ground-truth and generated samples for each feature across 3000 images. To intuitively demonstrate the superiority of MorphDiff, we displayed 10 most significantly changed features (with the lowest $p$ value) by using a scatter plot to illustrate the degree of proximity between the ground truth and the generated images as shown in Fig. 3c and Supplementary Fig. 14b, c. The $x$-axis represents the difference between the ground-truth CellProfiler features and control CellProfiler features, while the $y$-axis represents the difference between the generated CellProfiler features and control CellProfiler features. Considering the high density on the diagonal and the corresponding $R^2$ scores, it is evident that MorphDiff(I2I) can accurately predict morphological changes on these

significantly changed CellProfiler features. To further evaluate the generalizability of MorphDiff, we performed the same analysis on the CDRP OOD set and LINCS leave-one-out set in Supplementary Fig. 15 and 16c, respectively. Our conclusion is consistent across different settings and datasets.

Moreover, it is crucial to evaluate whether the model can capture the diversity among different perturbations and generate cell morphology with specificity to perturbations. We utilized DeepProfiler[18] to extract morphological embeddings from 10 CDRP OOD perturbations with the most images, and we term them as DeepProfiler embeddings. Subsequently, we trained an SVM binary classifier[38] for each pair of perturbations on the ground-truth cell morphology and calculated the classification accuracy for the DeepProfiler embeddings generated by MorphDiff(I2I), MorphDiff(G2I), and IMPA. The results in Fig. 3d clearly show that MorphDiff(G2I) achieves the highest accuracy among the three methods, which indicates that it best captures the perturbation-specific cell morphology patterns. We showed the SMILES of the corresponding drugs in Supplementary Fig. 17a. We conducted the same analysis on 10 drugs selected from the LINCS leave-one-out set with the most ground-truth images. The results shown in Supplementary Fig. 16b further confirm the superiority of MorphDiff. The SMILES for these drugs can be found in Supplementary Fig. 17b. The results also indicate that the generated output of MorphDiff(G2I) has higher perturbation specificity than that of MorphDiff(I2I) at the drug level. This is because MorphDiff(I2I) additionally uses control images as input, which can be influenced by technical variations that may introduce substantial biases and further affect the detection of perturbation-specific change, especially when the phenotypic changes caused by the perturbation are subtle[18].

## Pre-trained MorphDiff as a promising tool in phenotypic drug discovery

Small-molecule compounds are agents that modify a target to affect its functionality[39]. These compounds can either reduce or accelerate the activity of a target, which is typically a biomolecule like a protein, enzyme, receptor, or gene. These biomolecules are involved in signaling or metabolic pathways, often specific to diseases[40]. Biomolecules play a pivotal role in disease development and progression, primarily through communication facilitated by interactions between proteins and nucleic acids or proteins themselves. These interactions often lead to signal amplification or metabolic process alterations, further affecting disease development. Typically, a target can be associated with multiple drugs. For instance, amcinonide, dexamethasone, and betamethasone can all impact the gene *NR3C1*, while LFM-A12, RG-14620, WHI-P154, and chrysophanol can affect the gene *EGFR*. Based on these observations, we aim to investigate whether drugs acting on the same target will exhibit similar influences on morphological changes compared to control images, and we wonder whether MorphDiff can capture such patterns. We selected 10 targets (CDRP Target_MOA set, see the Datasets section) with sufficient corresponding compounds to include enough cell morphology for analysis, and these data are not present in the CDRP Training set for MorphDiff and IMPA. Firstly, we assessed the models' generative performance. We conducted a benchmark test with the pre-trained MorphDiff(I2I), MorphDiff(G2I), and IMPA on the CDRP Target_MOA dataset in the same way as in the previous section. These models were pre-trained on the CDRP Training set. As illustrated in Fig. 4a, MorphDiff(I2I) significantly outperforms IMPA across all image generation quality metrics, with a corrected $p$ value of <0.05, as determined by the one-sided Wilcoxon signed-rank test and Bonferroni correction. For the comparison between MorphDiff(I2I) and MorphDiff(G2I), there is no significant difference. We further investigated how the performance of MorphDiff methods varies across different MOAs and targets. Specifically, we evaluated how performance is affected by the distance between the CDRP Training set and the evaluated CDRP Target_MOA dataset. As detailed analysis in

Supplementary Notes 4, we found that model performance is influenced by the similarity between training and evaluation datasets, with performance degrading as the evaluation compounds become more distant from the training data. Consequently, when applying MorphDiff methods, researchers should consider potential performance limitations for test compounds that differ significantly from those in the training set.

To determine whether the images of perturbations related to these targets are significantly different from the control images, we extracted CellProfiler feature vectors from the ground-truth and the generated images, then projected the CellProfiler features through UMAP[41], as depicted in Supplementary Figs. 18 and 19. We observed that the ground-truth CellProfiler feature vectors induced by various drugs are clearly distinct from the ground-truth control set in the CellProfiler feature space. The CellProfiler feature vectors generated by the pre-trained MorphDiff (both modes) drive a consistent effect with the ground truth after UMAP projection. On the other hand, IMPA generates multiple clusters of perturbations targeting the same gene, while these perturbations should have a similar effect according to the ground truth, which indicates its poor generalization performance. Additionally, the Wasserstein distance computed between the perturbed and control CellProfiler features quantitatively demonstrates the advantages of MorphDiff.

Moreover, we investigated whether our method could identify the significantly changed CellProfiler features after being treated with drugs that target different genes. First, we adopted the chi-square test between the ground-truth control and the perturbed morphology to identify significantly changed CellProfiler features with the $p$ value threshold at 0.05. Then, we calculated the F1 score between the significantly changed CellProfiler features identified with the generated and ground-truth morphology. We also sampled 10 times for each model to determine the significance of the results. Figure 4b demonstrates that MorphDiff(I2I) effectively identifies significantly changed CellProfiler features, outperforming IMPA for 6 of 10 targets with corrected $p$ value < 0.05. MorphDiff(I2I) also outperforms MorphDiff(G2I) for 5 targets, while MorphDiff(G2I) outperforms MorphDiff(I2I) for 2 targets significantly. In addition, we also validated whether the relative distance between cell morphology from different targets could be preserved with our generative model. We used DeepProfiler[18] to extract DeepProfiler embeddings of cell morphology from different targets, and computed Wasserstein distance to measure the pairwise distance between them. We found that the Wasserstein distance computed from the generated morphology by MorphDiff(I2I) and MorphDiff(G2I) is highly consistent with the ground truth, as Fig. 4c shows, where the correlation score is up to 0.97. These results demonstrate that MorphDiff, in both modes, exhibits strong generalization ability. It effectively learns complex and in-depth relationships between drug perturbations and morphology, predicting morphological changes and capturing diversity at the target level.

To further explore the potential utility of MorphDiff as a valuable tool in advancing drug discovery, we then focused on an important direction in drug discovery: identifying the MOA of a compound. We investigated whether cell morphology provides necessary information in the identification of drug MOA and whether the cell morphology inferred by MorphDiff also contains such information. Concretely, as demonstrated in Fig. 4d, we designed a framework for MOA retrieval with cell morphology images. Given query cell DeepProfiler embeddings, we retrieved possible MOAs based on the minimum distance between query morphology images and known morphology images corresponding to certain MOAs. Due to the highly diverse and noisy properties of cell morphology, we adopted for DeepProfiler embedding retrieval to better account for the variance of cell morphology. The same framework can also be applied to MOA retrieval with gene expression by simply replacing the DeepProfiler embeddings with gene expression vectors and using the Euclidean distance metric.

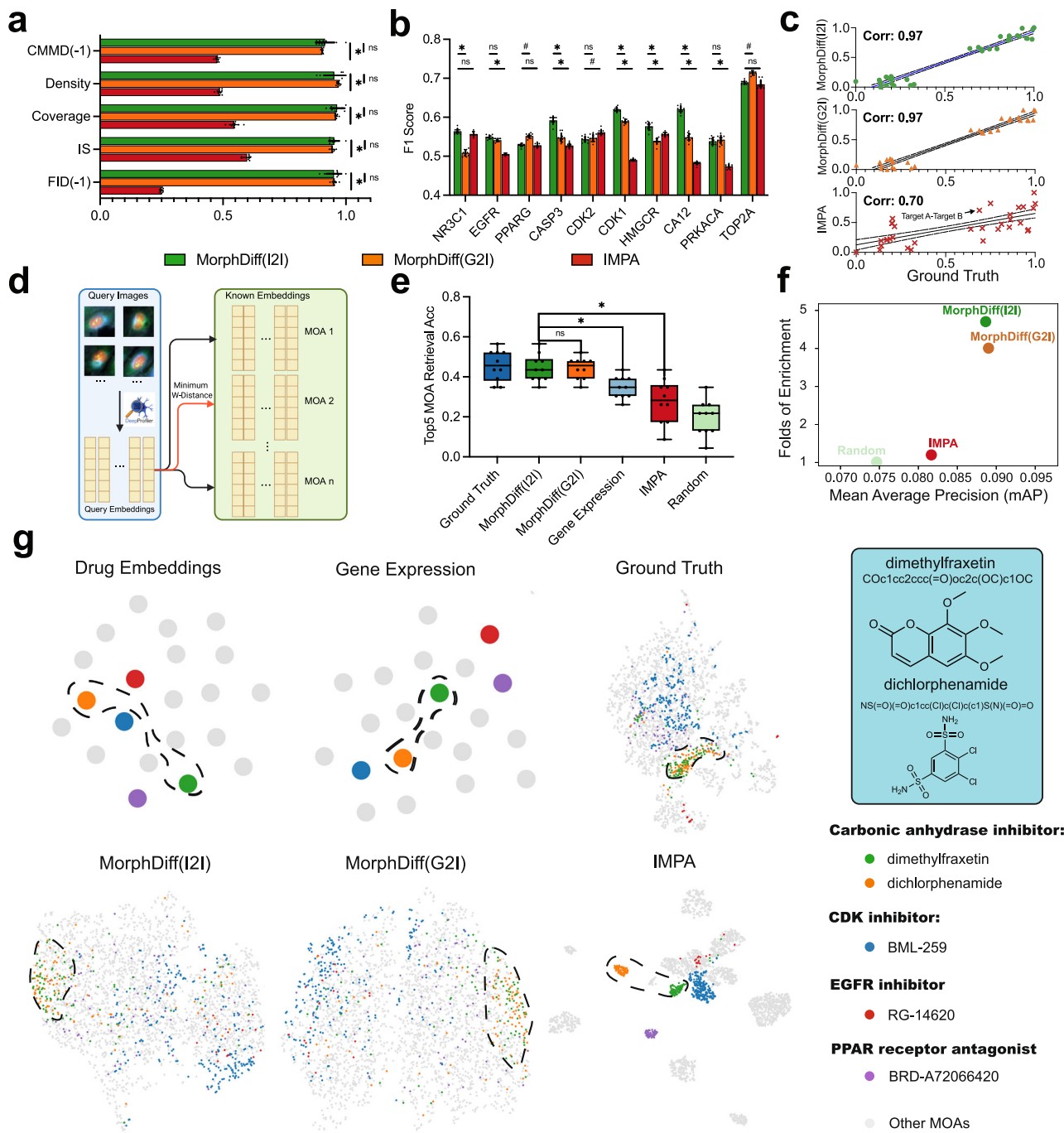

**Fig. 4 | Pre-trained MorphDiff potentially promotes drug discovery. a** General generative evaluation of pre-trained MorphDiff and IMPA. Methods requiring reference control images used 10 distinct control image groups from independent plates (*n* = 10 biological replicates), while other methods used 10 sampling iterations with different random seeds. Statistical comparisons used one-sided Wilcoxon signed-rank tests with Bonferroni correction (*p* value < 0.05). **b** F1 scores measure performance in identifying whether more than 200 CellProfiler features undergo significant changes under drug perturbations across various targets (*n* = 10 biological replicates). The tests are one-sided Wilcoxon signed-rank tests with Bonferroni correction (*p* value < 0.05). For **a**, **b**, "*" indicates significance; "ns" indicates non-significance; hashtag ("#") indicates the situation that the baseline performs better significantly. Data presented as mean values ± SD. **c** Wasserstein distance between different pairs of target-level perturbations for the ground-truth and generated cell morphology, computed with DeepProfiler embeddings. For

each pair of target-level perturbations, the ground-truth Wasserstein distance is the *x*-axis, and the corresponding generated Wasserstein distance is the *y*-axis. **d** Schematic overview of retrieval workflow. Created in BioRender. Group, A. (2025) https://BioRender.com/492h3v7. **e** Top 5 MOA retrieval results across modalities. Sixty-nine drugs were randomly divided into reference (47) and query (22) sets across 10 iterations. Boxplots show median, quartiles, and range. The whiskers in the boxplots mark the minima to maxima. **f** MOA matching performance using generated cell morphology DeepProfiler embeddings: Mean Average Precision (*x*-axis) versus Folds of Enrichment (*y*-axis). Points represent means across all query morphologies using 0.90 threshold (top 10%). **g** UMAP projections by drug MOAs across modalities: drug structure embeddings, gene expression, and DeepProfiler embeddings from ground-truth and generated cell morphology. BML-259, RG-14620, and BRD-A72066420 structures shown in Supplementary Fig. 23. Source data are provided as a Source data file for (**a**–**g**).

The retrieval experiments were conducted on the CDRP Target_MOA dataset. For the 69 drugs that act on 10 targets, we used different random seeds to divide them into two groups: one group of 47 (reference set) and another of 22 (query set). We retain the cases where the MOAs corresponding to the query drugs are present in the reference set for validation purposes. These 69 drugs correspond to 35 distinct Mechanisms of Action (MOAs), with some drugs associated with multiple MOAs. This complexity makes the retrieval task challenging. By calculating the Wasserstein distance (DeepProfiler embeddings) and MSE (Gene Expression) between the query set and the reference set, we selected the top $k$ closest drugs in the reference set as retrieval results for each query drug. If the correct MOA(s) are the same as or intersect with the MOA(s) of the top $k$ nearest drugs, we consider it a correct retrieval. We conducted the top 5 retrieval of MOA(s) on ground-truth DeepProfiler embeddings, MorphDiff(I2I)-generated DeepProfiler embeddings, MorphDiff(G2I)-generated DeepProfiler embeddings, IMPA-generated DeepProfiler embeddings, gene expression, and random retrieval results. To assess the significance of the experimental results, we randomly split the dataset 10 times with different random seeds. Figure 4e demonstrates that MorphDiff not only outperforms gene expression but also significantly eclipses IMPA regarding retrieval accuracy, as indicated by $p$ value of <0.05, based on one-sided Wilcoxon signed-rank test with Bonferroni correction. The average accuracy of MorphDiff-generated output outperforms IMPA and gene expression-based retrieval by 16.9% and 8.0% respectively. To further validate the robustness of our methods, we also experimented with $k = 6, 7, 8, 9, 10$ as Supplementary Fig. 20 shows and found that both modes of MorphDiff consistently outperform the baseline methods.

In addition to the above evaluation for MOA retrieval, to further demonstrate the effectiveness of the methods, we also evaluated MOA matching. We used two metrics, folds of enrichment[18] and mAP[42]. The details of these two metrics are shown in the Evaluation Metrics section. Intuitively, for each generated cell morphology, these metrics measure whether the top similar ground-truth cell morphology is treated by the drug belonging to the same MOA. As shown in Fig. 4f, we found that both modes of MorphDiff consistently outperform the baseline methods when the threshold is set to the top 10%. We also provided the results for different thresholds in Supplementary Fig. 21 to validate the robustness of our method. As the L1000 gene expression database is much larger than the current cell morphology database, we further explored the potential application of our model to generate the morphology images using the L1000 gene expression data without the corresponding ground-truth morphology images. We constructed a reference set of existing ground-truth morphology images covering 60 types of MOAs and 26112 images, which covers all drugs with available MOAs in the collected CDRP dataset and does not overlap with the CDRP Training set. Subsequently, we generated 2930 cell morphology images for 8 drugs whose MOAs are included in the reference set, but the ground-truth cell morphology images are not yet profiled, allowing us to evaluate the exploration capabilities through MOA matching. Detailed information on these drugs can be found in the Datasets section. The results shown in Supplementary Fig. 22 indicate that MorphDiff, in both modes, achieves higher scores on two metrics in different threshold settings, demonstrating its potential to explore the massive perturbed gene expression space.

Furthermore, drugs with similar structures usually have the same target and MOA(s) in most cases[43]. However, it would be more interesting to investigate the drugs with the same MOA(s) but that are structurally dissimilar[44,45] to improve biological safety and drug development flexibility. Therefore, we explored whether other modalities, such as gene expression and cell morphology, can provide complementary information to identify these drugs, and we also investigated whether the cell morphology generated by MorphDiff on specific perturbations contains such complementary information.

As shown in Fig. 4g, we showed that dimethylfraxetin and dichlorphenamide both belong to Carbonic anhydrase inhibitor, yet their structures are distinctly dissimilar. Meanwhile, the drug embeddings and gene expression of dimethylfraxetin and dichlorphenamide are of substantial distances. In the drug embedding space, BML-259, annotated with the CDK inhibitor, and RD-14620, annotated with the EGFR inhibitor, exhibit structural similarities to dichlorphenamide. Meanwhile, BRD-A72066420, with the PPAR receptor antagonist, is closer to dimethylfraxetin. The structures of these drugs can be found in Fig. 4g and Supplementary Fig. 23. In contrast, in the cell morphology space characterized by the DeepProfiler embeddings, we observed that cell morphology treated with dimethylfraxetin and dichlorphenamide have similar feature distributions and can be clearly separated from cell morphology treated with drugs from other MOAs, which indicates that cell morphology could provide complementary information in addition to drug structure and gene expression to identify the MOAs of drugs. We also observed that the cell morphology generated with both modes of MorphDiff also exhibited these patterns, with an overlapping distribution of cell morphology features treated with dimethylfraxetin and dichlorphenamide. The baseline method IMPA, as it only relies on the drug structure to generate the perturbed cell morphology, only generates biased predictions with cell morphology treated with dimethylfraxetin and dichlorphenamide separated far away. An additional example is bezafibrate, BRD-K54708045, and ciglitazone, which are all PPAR receptor agonists. In Supplementary Fig. 24, we observed that their drug embeddings and gene expression are not closely distributed, but BRD-K64402243, belonging to Caspase inhibitor, is closer to ciglitazone, annotated with PPAR receptor agonist. However, in the cell morphology space, we can see that the DeepProfiler embeddings of these three drugs from the same MOA clustered together. Similar to the previous example, MorphDiff(I2I) and MorphDiff(G2I) both successfully capture this pattern, while IMPA generates biased cell morphology. Based on these observations, we found that the relationship between MOAs and gene expression, drug structure, and morphology is complex. None of these modalities alone can perfectly predict the MOA; thus, the retrieval and prediction of MOAs is still a challenging task[10]. Integrating shared and complementary information across these modalities will be a key to improving the identification of MOAs of drugs[9] and further advancing drug discovery. In summary, we validated the potential capability of MorphDiff to retrieve and identify MOAs of specific drug perturbations, with consistently better accuracy than IMPA and gene expression and comparable performance with the ground truth. Furthermore, we showed that cell morphology contains complementary information to identify structurally dissimilar drugs with the same MOA, with potential application in advancing phenotypic drug discovery.

## Discussion

The exploration of cellular state transformation under genetic and drug perturbations enables numerous applications in drug discovery and biological research. Morphological profiling can cost-effectively capture thousands of features across perturbations by disease, mutation, or drug treatments and provide a unique view of cell state. Therefore, we propose MorphDiff, a diffusion-based generative model that can generate high-quality perturbed cell morphology conditioned with perturbed gene expression, efficiently leveraging the shared information between gene expression and cell morphology. We compared MorphDiff with several baseline methods and demonstrated its superiority with extensive experiments on three large-scale datasets. For genetic perturbation datasets, we analyzed the correlation between genes and morphological features, and the results show that MorphDiff effectively captures the pattern between gene expression and morphological profile, establishing a bridge to study the shared information between them. In the case of drug perturbations, MorphDiff successfully predicts the morphological changes caused by various drug perturbations. Furthermore, we applied the pre-trained

MorphDiff model to unseen perturbations affecting specific targets and demonstrated its ability to capture diversity at the target level. This showcases the generalizability of MorphDiff and its potential to facilitate the discovery of drug targets. Moreover, our experiments demonstrate further applications in drug discovery, enabling the investigation of drug MOAs through our advanced retrieval pipeline.

MorphDiff operates in two distinct modes: I2I and G2I. The G2I mode utilizes gene expression data exclusively as input, whereas the I2I mode incorporates both gene expression data and control cell morphology images. We have observed that the I2I mode exhibits greater variance in performance compared to the G2I mode. This variability can be attributed primarily to the quality and distribution characteristics of the selected control cell morphology sets used as input for the I2I mode. Our analysis indicates that the quality of the control cell morphology may potentially influence the quality of the generated output. Detailed analysis and tips for filtering DMSO control images can be referred to the Supplementary Notes 3 and Supplementary Fig. 25. Both modes of MorphDiff satisfy the criteria for effective generative modeling, achieving low FID, CMMD, and high scores in Inception Score, density, and coverage metrics. For most application scenarios, either mode is appropriate. However, when users specifically aim to analyze morphological feature changes relative to certain control cells-as discussed in Figs. 3c and 4b MorphDiff (I2I) represents the more suitable option. Nevertheless, users should carefully examine the quality and distribution of selected control morphology images, as these factors can potentially impact the generative performance.

Due to the data heterogeneity and lack of high-quality and massive datasets with matched time-point and concentration information, we do not explicitly encode these properties in our framework. In the future, if there are new datasets with massive L1000 gene expression profiles and cell morphology datasets with high-quality matched time-point and concentration information, our framework can also be easily extended to support explicit encodings of this information. As our model is based on the latent diffusion framework[14], which supports flexible conditioning, we can also add time-point and concentration embeddings as input conditions along with the gene expression profile so that our model could explicitly encode this information as input.

For future work, first, MorphDiff is conditioned on perturbed gene expression, which means that MorphDiff cannot directly be applied to perturbations unmeasured by transcriptomic profiling experiments. Although the L1000 assay[46] already profiles a vast number of diverse perturbations, there are still expectedly diverse new perturbations in the drug development process. Therefore, integrating MorphDiff with methods that can predict changes in gene expression under unseen perturbations, such as GEARS[47], is a possible extension. Second, the inference of MorphDiff is still costly as it is based on the diffusion model, which is known to be relatively slow in generating new samples. Therefore, the adaptation of more advanced sampling techniques for diffusion models in the future may further improve the efficiency of MorphDiff. Furthermore, though existing works[9–12] and the analyses in this paper have demonstrated remarkable precision in the prediction between gene expression and cell morphology, this problem still remains challenging in some cases. Some perturbations only significantly affect either the transcriptome or cell morphology, while having only subtle and difficult-to-observe effects on the other[10]. This phenomenon may arise from the underlying mechanisms of the cell responses to perturbations and various technical confounders in the experimental setup. Such complexity increases the difficulty of accurate prediction and analysis between gene expression and cell morphology, so addressing this challenge could be a meaningful future direction. Thus, incorporating more diverse input conditions, including but not limited to text description, drug structure, and chromatin accessibility in addition to gene expression, may further enhance the utility and generalizability of MorphDiff, making it a more powerful tool in predicting the cell morphology response under

various perturbations. In addition, while extensive benchmarks have demonstrated that MorphDiff outperforms other baseline methods in predicting out-of-distribution perturbations, we observed that its performance may degrade when applied to perturbations that deviate significantly from the training dataset distributions (Supplementary Notes 4). Although this limitation is expected for most deep learning methods in predicting perturbation response, including previous works such as CellOT[48] and IMPA[3], the curation of large-scale, high-quality, comprehensive datasets and the development of advanced learning paradigms may further enhance generalization performance when predicting responses to distant perturbations, and could be a promising direction for future research. Lastly, MOA prediction remains "notoriously challenging" as a major bottleneck in drug discovery, and there is no assay that can reliably be used to identify a majority of known MOA classes successfully[10]. Therefore, integrating multimodal information, including but not limited to gene expression, morphological profile, and drug structures, may improve MOA identification accuracy, and combining MorphDiff-generated cell morphology profile is a possible direction in achieving this goal.

The recent explosion of generative AI (GenAI)[14,49,50] has significantly transformed many areas. Vivid and high-resolution images and videos can now be generated with simple text descriptions at ultra-speed and impressive quality. We anticipate that cell morphology, as a particular type of image, will also benefit significantly from the advancements of GenAI. Our work MorphDiff, has demonstrated the potential of large-scale generative models to generate high-quality and high-fidelity cell morphology and has explored several potential biological applications. It is anticipated that large-scale pre-training on the increasingly available high-throughput cell morphology profiling datasets[6], combined with rapidly advancing GenAI methods, will enhance the precision and fidelity of cell morphology generation and further advance phenotypic drug discovery.

## Methods
### Problem definition

We collected $M$ cell morphology $\{\mathbf{Y}_i\}_{i=1}^M$ corresponding to $N$ unique perturbations $\{p_j\}_{j=1}^N$ that can either be genetic perturbation or drug treatment. DMSO refers to cells cultured in Dimethyl sulfoxide, which serves as the control group without perturbation. We use $\mathbf{Y}_{\text{DMSO}}$ as the cell morphology for the control group. The perturbation for cell morphology $\mathbf{Y}_i$ is denoted as $q_i$, thus $q_i \in \{p_j\}_{j=1}^N$. The L1000 gene expression corresponds to perturbation $q_i$ is denoted as $\mathbf{I}_{q_i}$. Our main objective is to train a model $f$ that generates the cell morphology conditioned on L1000 gene expression with high fidelity. Concretely, we aim to

$$\min \sum_{i=1}^M \text{Distance}(\mathbf{Y}_i, f(\mathbf{I}_{q_i})) \tag{1}$$

The cell morphology of the control group may serve as prior information for modeling the cell morphology after perturbation. Thus, $f$ can also take $\mathbf{Y}_{\text{DMSO}}$ as input, which can be written as $f(\mathbf{I}_{q_i}, \mathbf{Y}_{\text{DMSO}})$.

### MorphDiff model

Following[14], MorphDiff is composed of two parts: Morphology VAE (MVAE) and Latent Diffusion Model (LDM). MVAE compresses the high-dimensional multichannel cell morphology images to low-dimensional latent representation with minimal information loss. LDM is trained to denoise from random Gaussian noise to low-dimensional latent representation by recursively adding Gaussian noise. In practice, a random vector is sampled from the Gaussian distribution and then passed to LDM for the denoise process. Subsequently, the predicted high-quality cell morphology can be decoded from the denoised output of LDM using the MVAE decoder. Such an approach offers a unique advantage: by compressing the high-dimensional cell morphology image to a low-dimensional latent

representation, MorphDiff is computationally efficient, as sampling is performed on a low-dimensional space.

**Morphology VAE (MVAE).** Morphology VAE (MVAE) comprises two parts, encoder $E$ and decoder $D$. Given the cell morphology $\mathbf{Y} \in \mathbb{R}^{H \times W \times 5}$, the encoder $E$ encodes $\mathbf{Y}$ into a latent vector $\mathbf{z} = E(\mathbf{Y})$ and the decoder $D$ reconstructs the cell morphology $\hat{\mathbf{Y}} = D(\mathbf{z})$. Thus, a reconstructive loss $L_{\text{rec}}$ can be calculated between $\mathbf{Y}$ and $\hat{\mathbf{Y}}$. MVAE is trained in an adversarial manner following[30], where a patch-based discriminator $D_\phi$ is optimized to classify the original cell morphology from the reconstructed morphology $D(\mathbf{z})$. The patch-based discriminator is implemented with a multilayer convolutional neural network. On the other hand, it is a common practice in VAE training to regularize the latent $\mathbf{z}$ to be zero-centered and enforce small variance by introducing a regularizing loss term $L_{\text{reg}}$. A Kullback-Leibler term $L_{\text{reg}}$ between $q_E(\mathbf{z}|\mathbf{Y}) = \mathcal{N}(\mathbf{z}; E_\mu, E_{\sigma^2})$ and the standard normal distribution $\mathcal{N}(\mathbf{z}; 0, \mathbf{I})$ is adopted to regularize the latent representation $z$. Additionally, a LPIPS loss[51] ($L_{\text{lpips}}$) is applied to minimize the difference between the latent features of ground-truth images and those of generated images. The overall objective of training MVAE can be summarized as follows:

$$L_{\text{MVAE}} = \min_{E,D} \max_\phi (L_{\text{rec}}(\mathbf{Y}, D(E(\mathbf{Y}))) + \log D_\phi(\mathbf{Y}) + L_{\text{reg}}(\mathbf{Y}; E, D)) + L_{\text{lpips}}(\mathbf{Y}, D(E(\mathbf{Y}))) \tag{2}$$

**Learned perceptual image patch similarity (LPIPS) loss in MVAE.** The origin framework of VAE in the Stable Diffusion model[14] employs LPIPS loss[51] to minimize the difference between ground-truth images and generated images. The principle of LPIPS loss is to extract features of images with 5 conv layers from a pre-trained VGG network[52], and compute $l_2$ distance between ground-truth features and the generated features output from each layer. The integration of LPIPS will significantly improve the fidelity of VAE generation. However, as the pre-trained VGG model was trained with 3-channel images (RGB input), it is inconsistent with our 5-channel input (DNA, ER, RNA, AGP, Mito input). Therefore, we duplicate each channel from a shape of (128, 128) to (3, 128, 128) to ensure that the images from each channel are compatible with the input shape required by VGG. We compute LPIPS loss for each channel, respectively, and average the sum of loss across five channels. The formulation can be written as:

$$d_i(\mathbf{x}_i, \mathbf{x}_{0i}) = \sum_l \frac{1}{H_l W_l} \sum_{h,w} ||\mathbf{w}_l \odot (\hat{\mathbf{y}}_{ihw}^l - \hat{\mathbf{y}}_{0ihw}^l)||_2^2 \tag{3}$$

$$d(\mathbf{x}, \mathbf{x}_0) = \frac{1}{5} \sum_i d_i(\mathbf{x}_i, \mathbf{x}_{0i}) \tag{4}$$

$l$ represents $l$th layer in 5-layer VGG, and $\mathbf{w}^l$ is the parameter used to scale the activations. $\mathbf{x}_i$ and $\mathbf{x}_{0i}$ represent the ground-truth image and the generated image in $i$th channel, respectively, while $\hat{\mathbf{y}}_i^l$ and $\hat{\mathbf{y}}_{0i}^l$ are adopted as corresponding features obtained from the $l$th layer. H$l$ and W$l$ are height and width of embeddings output from the $l$th layer.

**Latent diffusion model (LDM).** Latent diffusion model (LDM) contains a noising and denoising process. Given a sample $\mathbf{z}_0 \in q(\mathbf{z})$ where $q(\mathbf{z})$ is the distribution of cell morphology latent representation. The noising process progressively adds Gaussian noise for $T$ steps and obtains $\mathbf{z}_1, \mathbf{z}_2, \ldots, \mathbf{z}_T$. The formulation of the noising process can be written as:

$$q(\mathbf{z}_t|\mathbf{z}_{t-1}) = \mathcal{N}(\mathbf{z}_t; \sqrt{\alpha_t}\mathbf{z}_{t-1}, (1-\alpha_t)\mathbf{I}) \tag{5}$$

$$q(\mathbf{z}_{1:T}|\mathbf{z}_0) = \prod_{t=1}^T q(\mathbf{z}_t|\mathbf{z}_{t-1}) \tag{6}$$

where $q(\mathbf{z}_t|\mathbf{z}_{t-1})$ can be interpreted as obtaining $\mathbf{z}_t$ by adding Gaussian noise to $\mathbf{z}_{t-1}$ parameterized by $1-\alpha_t$. We denote $\bar{\alpha}_t = \prod_{i=1}^t \alpha_i$. The denoising process aims to train a denoising network to recursively remove the noise from $\mathbf{z}_t$ ($\mathbf{z}_t = \sqrt{\alpha_t}\mathbf{z}_{t-1} + (1-\alpha_t)\epsilon$) and $\epsilon \sim \mathcal{N}(0, \mathbf{I})$. Concretely, the denoising process can be formulated as follows, where $p_\theta(\mathbf{z}_{t-1}|\mathbf{z}_t)$ denotes the probability distribution of $\mathbf{z}_{t-1}$ given $\mathbf{z}_t$:

$$p_\theta(\mathbf{z}_{t-1}|\mathbf{z}_t) = \mathcal{N}(\mathbf{z}_{t-1}; \mu_\theta(\mathbf{z}_t, t), \Sigma_\theta(\mathbf{z}_t, t)) \tag{7}$$

$$p_\theta(\mathbf{z}_{0:T}) = p(\mathbf{z}_T) \prod_{t=1}^T p_\theta(\mathbf{z}_{t-1}|\mathbf{z}_t) \tag{8}$$

$\mu_\theta(\mathbf{z}_t, t)$ and $\Sigma_\theta(\mathbf{z}_t, t)$ are the Gaussian mean and variance by the denoising network $\theta$. Similar to other generative models, the loss function of LDM can be formulated as maximizing the log-likelihood of the samples generated belonging to the original data distribution ($\log(p_\theta(\mathbf{z}_0))$). However, direct optimization is intractable, as we need to integrate the $T$ steps into the latent space. For L1000 gene expression $\mathbf{l}$, we encode it as an intermediate representation and map it to UNet through cross-attention. Note that the encoding process is not learnable and is a simple linear transformation, which is different from the training method in conditional latent diffusion[14]. This is because L1000 gene counts are already processed in ref. 9, so we believe it is better to map it to UNet directly. The encoder is represented as EC. Following DDPM[53], we can simplify the learning objective as:

$$L = \mathbb{E}_{\mathbf{z}, \epsilon \sim \mathcal{N}(0, \mathbf{I}), t} ||\epsilon - \epsilon_\theta(\mathbf{z}_t, t, \text{EC}(\mathbf{l}))||_2^2 \tag{9}$$

$\mathbf{z}_t$ is a noisy version of $\mathbf{z}_0$ at step $t$ and $\epsilon_\theta(\mathbf{z}_t, t, \text{EC}(\mathbf{l}))$ is a denoising network intended to predict $\epsilon$ from $\mathbf{z}_t$ conditioned on L1000 gene expression $\mathbf{l}$. The cross-attention for the intermediate layer $ith$ in UNet can be represented as $\text{Attention}(\mathbf{Q}, \mathbf{K}, \mathbf{V}) = \text{softmax}(\frac{\mathbf{QK}^T}{\sqrt{d}}) \cdot \mathbf{V}$, where $\mathbf{Q} = \mathbf{W}_\mathbf{Q}^{(i)} \cdot \varphi_i(\mathbf{z}_t)$, $\mathbf{K} = \mathbf{W}_k^{(i)} \cdot \text{EC}(\mathbf{l})$, $\mathbf{V} = \mathbf{W}_\mathbf{V}^{(i)} \cdot \text{EC}(\mathbf{l})$. $\varphi_i(\mathbf{z}_t)$ denotes an intermediate representation in UNet, and $\mathbf{W}_\mathbf{Q}^{(i)}, \mathbf{W}_\mathbf{K}^{(i)}$ and $\mathbf{W}_\mathbf{V}^{(i)}$ are learnable matrices. Following DDPM[53], the detailed LDM training and sampling algorithm is shown as Algorithm 1 and Algorithm 2:

**Algorithm 1.** Training Algorithm of MorphDiff
1: **repeat**
2:   $\mathbf{z}_0 \sim q(\mathbf{z}_0)$
3:   $t \sim \text{Uniform}(\{1, \ldots, T\})$
4:   $\epsilon \sim \mathcal{N}(0, \mathbf{I})$
5:   Take the gradient descent step on $\nabla_\theta \| \epsilon - \epsilon_\theta (\sqrt{\bar{\alpha}_t}z_0 + \sqrt{1-\bar{\alpha}_t}\epsilon, t, \text{EC}(\mathbf{l})) \|^2$
6: **until** converged

**Algorithm 2.** Sampling Algorithm of MorphDiff
1:   $\mathbf{z}_T \sim \mathcal{N}(0, \mathbf{I})$
2: **for** $t = T, \ldots, 1$ **do**
3:   **if** $t > 1$ **then**
4:     $\mathbf{z} \sim \mathcal{N}(0, \mathbf{I})$
5:   **else**
6:     $\mathbf{z} = 0$
7:   **end if**
8:   $\mathbf{z}_{t-1} = \frac{1}{\sqrt{\alpha_t}} \left( \mathbf{z}_t - \frac{1-\alpha_t}{1-\sqrt{\bar{\alpha}_t}}\epsilon_\theta(\mathbf{z}_t, t, \text{EC}(\mathbf{l})) \right) + \sigma_t \mathbf{z}$
9: **end for**
10: **return** $z_0$

**Implementation of LDM.** LDM ($\epsilon_\theta(\mathbf{z}_t, t, \mathbf{l})$) is implemented with the UNet[54] architecture augmented with an attention mechanism following[14]. Concretely, the base UNet architecture uses a stack of residual layers and downsampling convolutions, followed by a stack of

residual layers with upsampling convolutions, with skip connections connecting the layers with the same spatial size. We use the UNet variant from[16] and replace the self-attention layer with alternating layers of (a) self-attention, (b) position-wise MLP, and (c) cross-attention layer. To inject the L1000 gene expression $l$ as a condition in the denoising UNet, we use the encoder(EC) same as when training to project $\mathbf{l}$ into an intermediate representation EC($\mathbf{l}$). The intermediate representation EC($\mathbf{l}$) is then mapped to each cross-attention layer (Attention($\mathbf{Q}, \mathbf{K}, \mathbf{V}$) = softmax$\left(\frac{\mathbf{QK}^T}{\sqrt{d}}\right)\mathbf{V}$) of UNet with

$$\mathbf{Q} = \mathbf{W}_{\mathbf{Q}}^{(i)} \cdot \phi_i(\mathbf{z}_t), \mathbf{K} = \mathbf{W}_{\mathbf{K}}^{(i)} \cdot \tau_\theta(\mathbf{l}), \mathbf{V} = \mathbf{W}_{\mathbf{V}}^{(i)} \cdot EC(\mathbf{l}) \qquad (10)$$

$\phi_i(\mathbf{z}_t)$ is the learnable intermediate representation in the latent space of the denoising UNet and $\mathbf{W}_{\mathbf{Q}}^{(i)}$, $\mathbf{W}_{\mathbf{V}}^{(i)}$ and $\mathbf{W}_{\mathbf{K}}^{(i)}$ are learnable matrices.

## Datasets

**Image processing.** The images collected from CDRP[23], JUMP[24], and LINCS[25] are all bulk-level cell plate images. To enable fine-grained analysis at the single-cell image level. We used the CellProfiler software[17] version 4.2.5 to segment bulk-level cell plate images and get the single-cell level images. Concretely, we used the `IdentifyPrimaryObejcts` and `IdentifySecondaryObjects` functions to identify individual objects(cells) in the images. The threshold strategy is set to global, and the thresholding method is set to minimum cross-entropy.

**JUMP dataset (U2OS cell line).** We collected cell morphology images of cells treated with 130 different gene overexpression perturbations from the JUMP Cell Painting dataset[24] with the script from https://github.com/jump-cellpainting/datasets/. Each image consists of five channels, namely mitochondria (MitoTracker; Mito), nucleus (Hoechst; DNA), nucleoli and cytoplasmic RNA (SYTO 14; RNA), endoplasmic reticulum (concanavalin A; ER), Golgi and plasma membrane (wheat germ agglutinin (WGA); AGP) and actin cytoskeleton (phalloidin; AGP). Meanwhile, we collected the bulk gene expression from the database[46] with the same cell line and perturbations. L1000 gene expression in this dataset contains 12328 genes. The gene expression and the cell morphology images can thus be aligned according to perturbation information. The total number of categories is 131, including 130 perturbations and control set. After that, we split the dataset into three sets, which are the training set, ID (in-distribution) set, and OOD (out-of-distribution) set. Firstly, we randomly picked 10 percent of perturbations as OOD sets, resulting in 13 perturbations. The remaining parts were randomly split into training set (90 percent) and ID set(10 percent), which indicates that the training set and the ID set contain both control and perturbation data. We termed them as JUMP Training set, JUMP ID set, and JUMP OOD set. Detailed information on these genes and their related pathways can be found in Supplementary Data 1.

**CDRP dataset (U2OS cell line).** We collected cell morphology images of cells treated with 959 different small molecules on 10 different plates (plate 25593, 25594, 25598, 25599, 26128, 26133, 26135, 26601, 266607, 26608) from CDRP[23], by script https://github.com/gigascience/paper-bray2017/blob/master/download_cil_images.sh. All images consist of five channels, recording information from nucleus, Endoplasmic reticulum, Nucleoli and cytoplasmic RNA, Golgi and plasma membrane, and Mitochondria. The concrete correspondence between dye names, cell structures, and CellProfiler names can be found in Supplementary Table 1. We also collected the bulk gene expression from https://broad.io/rosetta/ and L1000 gene expression in this dataset containing 977 genes. After aligning the gene expression data with morphological images for the same perturbations at identical concentrations, we identified a total of 960 categories, comprising 959 perturbations and a control set. Following the same split

principle as the JUMP dataset, we obtained 96 perturbations of the OOD set and 863 perturbations plus control for the training set and the ID set. We termed them as CDRP Training set, CDRP ID set, and CDRP OOD set. Detailed information on these drugs can be found in Supplementary Data 2.

To assess our model's ability to capture features on target and MOAs levels, we collected data with target and MOA labels. We additionally selected 10 targets (69 drugs annotated with 35 MOAs) from CDRP[23] (from plate 24277, 24278, 24294, 24301, 24305, 24310, 24319, 25739), for which enough corresponding drugs, gene expression[9], and images could be collected for analysis and test. These data do not overlap with the CDRP Train set mentioned above. The targets are *NR3C1, EGFR, PPARG, CASP3, CDK2, CDK1, HMGCR, CA12, PRKACA* and *TOP2A*. After the matching and cropping process, we got 2222 images for *NR3C1*, 707 images for *EGFR*, 793 images for *PPARG*, 466 images for *CASP3*, 435 images for *CDK2*, 342 images for *CDK1*, 475 images for *HMGCR*, 327 images for *CA12*, 496 images for *PRKACA*, 51 images for *EGFR*. We termed this set as CDRP Target_MOA set. Detailed information on these compounds can be found in the Supplementary Data 3.

**LINCS dataset (A549 cell line).** We collected the bulk gene profiles from[9] (LINCS part) and downloaded the corresponding images from[25]. In this dataset, we selected 61 drugs that have sufficient images, along with available MOA and target labels. The components of these images are the same as those in the U2OS dataset. We matched the gene expression and morphology images in the same way as CDRP. To construct the dataset for model validation, we first divided the data into 10 folds according to the 10 targets corresponding to 21 drugs. We termed it as LINCS Target leave-one-out set. For each experiment, we trained the model on another 9 folds and validated it on the remaining fold. We performed the same operation for the 10 MOAs (42 drugs) and termed it as LINCS MOA leave-one-out set. The details of the drug can be found in the Supplementary Data 4 and 5.

**Exploratory analysis on L1000 dataset without corresponding morphology images.** As the L1000 datasets cover more perturbations compared with the Cell Painting datasets, we added additional evaluation on datasets containing L1000 perturbed gene expression profiles, and the perturbations covered in the dataset have not been applied to Cell Painting assays. Thus, there are no ground-truth cell morphology datasets regarding these perturbations. For evaluation purposes, we only retained the drugs whose MOA annotations are present in our collected CDRP dataset.

We constructed a reference set containing perturbed morphology images, comprising 251 drugs annotated with 60 MOAs, containing 26,112 single-cell images. These images were downloaded from CDRP[23] and processed using CellProfiler. In particular, these drugs do not overlap with the CDRP training set used to pre-train the model. For L1000 gene expression data without corresponding annotations (query drugs), we collected 8 drugs annotated with 6 MOAs (293 gene expressions) from[14], with their MOAs available in the reference set to facilitate validation of the MOA matching performance. We generated 2930 single-cell images for query drugs. Detailed information on query drugs and reference drugs can be found in the supplementary files Supplementary Data 6 and 7.

**Statistical tests for assessing the distribution difference.** We performed statistical tests to assess whether the OOD set and leave-one-out set exhibit statistically significant distribution differences from the training set. Specifically, we conducted two statistical tests: Energy Distance tests[55], and Maximum Mean Discrepancy tests[56] to evaluate the distribution differences between the DeepProfiler embeddings of the training set and the ID/OOD sets. For the Energy Distance test, we employed the dcor.homogeneity.energy_test function from the dcor package[55]. The energy test assesses whether multiple groups originate

from the same distribution by examining the Euclidean distances between data points. This method compares point-pairwise distances within each group to those between different groups. If points are more closely clustered within their own groups than with points from other groups, this pattern indicates that these groups likely stem from different underlying distributions. For the Maximum Mean Discrepancy (MMD) tests[56], we determined if two groups come from different distributions by measuring their distance in a high-dimensional feature space. The key steps are: (1) transform data using an RBF kernel, (2) calculate the MMD statistic by comparing within-group similarities to between-group similarities, and (3) assess significance through permutation testing-repeatedly shuffling labels and recalculating MMD statistic to generate a null distribution. Due to the computational complexity of these two analysis, we randomly sampled 2000 cell morphology images from each dataset and conducted tests using 999 permutations. To ensure robustness and stability, we repeated all statistical tests for 10 rounds. The statistic values and $p$ values for the JUMP dataset, CDRP dataset, LINCS Target leave-one-out set, and LINCS MOA leave-one-out set across all three types of tests are presented in Supplementary Tables 3–10. The statistical testing results consistently demonstrate that the training set and the OOD or leave-one-out sets derive from distributions with statistically significant differences, while the training set and the ID set derive from distributions with no significant differences. This validation confirms that our performance evaluation on the OOD set and leave-one-out set effectively assesses model performance on distributions outside the domain of the training set.

## Evaluation setup

### Evaluation setup across baseline methods.
Among all the baseline methods evaluated in the manuscript, only MorphDiff and IMPA support training and generation with the original five-channel input images, while the other baseline methods only support three-channel RGB images as input and output. Therefore, to fairly assess different methods, we devised the following evaluation strategy following previous works[3]. During training, methods that directly support five-channel cell morphology images as input (MorphDiff and IMPA) were trained using these five-channel images. For methods that originally could only process RGB three-channel images, we modified their implementations to enable them to accept and generate five-channel images. For performance benchmarking, the five generative metrics can only be applied to three-channel RGB images. Therefore, we converted the five-channel output to three-channel images for benchmarking, following the methodology outlined in CellProfiler[17]. The weights assigned to the different channels are as follows: **ER**(1, 0, 0), **RNA**(0, 1, 0), **DNA**(0, 0, 1), **Mito**(0.5, 0.5, 0), **AGP**(0.5, 0, 0.5). Consequently, the resulting composited image is calculated as: (1∗**ER** + 0.5∗**Mito** + 0.5∗**AGP**, 1∗**RNA** + 0.5∗**Mito**, 1∗**DNA** +0.5∗**AGP**).

### Morphological feature extraction.
To evaluate the fidelity of the generated single-cell morphology images, we utilized widely used single-cell morphology feature extraction methods, including CellProfiler[17] and DeepProfiler[18]. CellProfiler is based on traditional image feature extraction methods, which could produce interpretable morphological features, while DeepProfiler is based on a trained deep neural network to extract low-dimensional dense embeddings of cell morphology images. CellProfiler is used to extract features from single-cell morphology images, allowing for a comparison between the features of ground-truth cell morphology and those of generated cell morphology. This evaluation is mainly for assessing the similarity between ground truth and generated cell morphology images on the interpretable feature sets because these features are usually involved in downstream analysis with respect to cell morphology images, such as BioMorph[57] and Cytominer[58]. Some functions provided by CellProfiler are merely used for the extraction of bulk-level image characteristics, thus, we selected several functions that can extract meaningful features from single-cell images. We employed `MeasureColocalization` (Measuring the correlation between any two channels), `MeasureGranularity` (Measuring the size and intensity of granular structures), `MeasureImageIntensity` (Measuring the intensity), `MeasureTexture` (Measuring the smoothness, coarseness, and regularity). Features with NaN values and zero variance are removed. Meanwhile, features with duplicated meaning and functions are also removed (e.g., sum and mean of intensity). After these processing and filtering steps, more than 200 CellProfiler features are kept for further analysis.

DeepProfiler[18] is a tool that can extract dense embeddings from morphology images with a five-channel input. We used it following the handbook provided by the authors. The handbook can be found at https://cytomining.github.io/DeepProfiler-handbook/docs/00-welcome.html.

### Morphological feature selection based on discriminability of perturbations.
CellProfiler Morphological features measure the cell morphology from multiple perspectives. Some morphological features may be highly correlated with the gene expression, while some may not depend on gene expression or perturbation. Thus, to select the most informative morphological features related to gene expression and perturbation for downstream analysis, we trained a random forest classifier from morphological features to predict the corresponding perturbations. We set the number of estimators to 100 and selected the top 10 features of the highest importance. The features of the highest importance in the classifier are considered the most discriminative features of perturbations.

### Morphological feature selection based on predictability.
We selected the most predictable morphological features to visualize the relationship between gene expression and cell morphological features. Concretely, we computed the $R^2$ score of each predicted cell morphological feature with respect to ground-truth cell morphological feature. Only features with $R^2$ score higher than 0.5 were kept. 36 features from the prediction of MorphDiff and 8 features from IMPA meet this criteria, and 42 most predictable features are kept considering the overlap between two feature sets.

## Evaluation metrics
We denote the ground truth images and the generated images as **X** and **X**′ respectively, and the number of images is $N$.

### Fréchet Inception Distance (FID).
The Fréchet Inception Distance measures the similarity between two sets of images. It was proposed by ref. [33] to quantify the similarity of the generated images to the ground-truth ones. Concretely, the generated images and ground-truth images are passed to the Inception V3 model pre-trained on ImageNet. The output encodings of ground-truth images and generated images are denoted as **E** and **E**′ with dimension ($N$, 2048), respectively ($N$ is the number of images in the dataset, and 2048 is the dimension of Inception V3 model encodings). The mean embedding vector of **E** and **E**′ are denoted as **m** and **m**′, and the covariance matrices **E** and **E**′ are denoted as **C** and **C**′. The final FID score is computed as $\mathrm{FID}(\mathbf{E},\mathbf{E}') = ||\mathbf{m} - \mathbf{m}'||_2^2 + \mathrm{Tr}(\mathbf{C} + \mathbf{C}' - 2(\mathbf{C}\mathbf{C}')^{1/2})$. Tr denotes the trace of the matrix. A lower FID score indicates better performance of the generative model.

### Inception Score.
Inception Score was introduced in ref. [34] to measure the diversity and quality of the generated images. Mathematically, the Inception Score can be written as $\mathrm{InceptionScore}(\mathbf{X}') = \exp(\mathbb{E}_{\mathbf{X}'} D_{\mathrm{KL}}(p(y|\mathbf{X}')||p(y)))$. $y$ is the marginal distribution of the Inception V3 model. First, the generated images are passed through the Inception V3 model to get the conditional distribution $p(y|\mathbf{x}')$, $\mathbf{x}' \in \mathbf{X}'$,

then the marginal distribution $p(y)$ is also computed by averaging $p(y|\mathbf{x}')$ over all the images. The KL divergence between $p(y)$ and $p(y|\mathbf{x}')$ is calculated and averaged on all images. The exponential of the averaged KL divergence is used as the Inception Score. A higher Inception Score means better performance of the generative model.

**Density and coverage.** Density and coverage introduced by[36] provide additional measures for assessing the fidelity and diversity of the generated images. Basically, we assume that $\mathbf{E} = \{\mathbf{E}_1, \mathbf{E}_2, \ldots, \mathbf{E}_N\}$ and $\mathbf{E}' = \{\mathbf{E}'_1, \mathbf{E}'_2, \ldots, \mathbf{E}'_N\}$ are the Inception V3 model encodings of the ground-truth images and generated images, respectively. We can approximate a manifold of $\mathbf{E}$ as

$$\text{manifold}(\mathbf{E}_1, \mathbf{E}_2, \ldots, \mathbf{E}_N) = \bigcup_{i=1}^{N} B(\mathbf{E}_i, \text{NND}_k(\mathbf{E}_i)) \qquad (11)$$

The sphere, denoted as $B(\mathbf{x}, d)$, is characterized by its center at the point $\mathbf{x}$ and its radius $d$. $\text{NND}_k(\mathbf{x})$ denotes the distance from point $\mathbf{x}$ to the $k$th nearest neighbor $\mathbf{x}$ among excluding itself. Then, we can define density as

$$\text{Density} = \frac{1}{kN} \sum_{j=1}^{N} \sum_{i=1}^{N} \mathbf{1}_{\mathbf{E}'_j \in B(\mathbf{E}_i, \text{NND}_k(\mathbf{E}_i))} \qquad (12)$$

Density quantifies the concentration of generated samples within high-probability regions of the real data distribution. Specifically, it measures how well the generated samples populate the neighborhoods of real data points, reflecting the local consistency and quality of the generated distribution. Meanwhile, coverage assesses the extent to which the generated samples span the full support of the real data distribution. It evaluates whether the generative model captures all major modes of the true data, ensuring diverse and comprehensive generation. Coverage is defined as:

$$\text{Coverage} = \frac{1}{N} \sum_{i=1}^{N} \mathbf{1}_{\mathbf{E}'_i \in B(\mathbf{E}_i, \text{NND}_k(\mathbf{E}_i))} \qquad (13)$$

Intuitively, the density and coverage score will be higher if the generated distribution effectively aligns with the ground-truth distribution while of great diversity.

**CLIP embeddings and maximum mean discrepancy.** The CMMD (CLIP-MMD) metric represents the squared MMD (Maximum Mean Discrepancy) distance between the CLIP embeddings[59] of the ground-truth image set and those of the generated image set[35]. CLIP embeddings are effective in representing the diverse and intricate content in images, benefitting from training an image encoder and a text encoder together using 400 million image-text pairs.

Intuitively, given the $\mathbf{E} = \{\mathbf{E}_1, \mathbf{E}_2, \ldots, \mathbf{E}_m\}$ as CLIP-encoded embeddings of ground-truth images, and $\mathbf{E}' = \{\mathbf{E}'_1, \mathbf{E}'_2, \ldots, \mathbf{E}'_n\}$ representing CLIP-encoded embeddings of generated images, and $k$ denoted as kernel function, an unbiased estimator of $\text{dist}^2_{\text{MMD}}(\mathbf{E}, \mathbf{E}')$ is given by:

$$\text{dist}^2_{\text{MMD}}(\mathbf{E}, \mathbf{E}') = \frac{1}{m(m-1)} \sum_{i=1}^{m} \sum_{i \neq j}^{m} k(\mathbf{E}_i, \mathbf{E}_j) + \frac{1}{n(n-1)} \sum_{i=1}^{n} \sum_{i \neq j}^{n} k(\mathbf{E}'_i, \mathbf{E}'_j)$$
$$- \frac{1}{mn} \sum_{i=1}^{m} \sum_{i=j}^{n} k(\mathbf{E}_i, \mathbf{E}'_j) \qquad (14)$$

Note that a lower CMMD score means better generative performance.

**Mean average precision.** Mean average precision (mAP) is an evaluation metric assessing the probability that samples of interest will rank highly on a list of samples rank-ordered by some distance or similarity metric[42], such as cosine similarity, Euclidean, Mahalanobis, etc. The mAP framework is agnostic to the choice of this distance metric. Given $q$th sample as $q$ (generated cell morphology image), annotated with $l_q$ (drug, MOA, or genetic perturbation). Assume that there are $n$ ground-truth cell morphology images. $\text{TP}_k$ represents true positive when ranking the top $k$ ground-truth samples by distance with $q$, while $\text{TN}_k$, $\text{FP}_k$, and $\text{FN}_k$ are defined similarly. Then the precision and recall at each rank can be calculated as $P_k = \frac{\text{TP}_k}{\text{TP}_k} + \text{FP}_k$ and $R_k = \frac{\text{TP}_k}{\text{TP}_n + \text{FN}_n}$. We can then calculate AP and mAP:

$$\text{AP}_q = \sum_k (R_k - R_{k-1}P_k) \qquad (15)$$

$$\text{mAP} = \sum_{q=1}^{N} \frac{\text{AP}_q}{N} \qquad (16)$$

**Folds of enrichment.** Folds of Enrichment is an evaluation metric assessing whether the perturbed cell morphology and ground-truth cell morphology belonging to the same MOA or genetic perturbation will have high similarity[18]. Concretely, for each generated cell morphology image, we calculated the odds ratio of a one-sided Fisher's exact test. The test is calculated using a 2 × 2 table. The first row contains the number of ground-truth cell morphology treated with the same treatment (drug, drug with the same MOA, or genetic perturbation) (positive matches) and different treatments (negative matches) at a selected threshold of the list of results. The second row is the same, but for the treatments below the threshold (the rest). The odds ratio is the sum of the first row divided by the sum of the second row. It estimates the likelihood of observing the generated cell morphology is top similar to the ground-truth cell morphology with the same perturbation.

**Coefficient of determination ($R^2$ score).** The coefficient of determination ($R^2$ score)[60] was created to evaluate prediction of various models and testing of hypotheses. It can evaluate how well the generated results fit the ground truth. Consider a ground truth set of $n$ values marked $y_1, y_2, \ldots, y_n$, each associated with a predicted (or generated) value $g_1, g_2, \ldots, g_n$. Define residuals as $r_i = y_i - g_i$ and $\bar{y}$ as the mean of ground-truth set:

$$\bar{y} = \frac{1}{N} \sum_{i=1}^{N} y_i \qquad (17)$$

Then we can measure the variability of the data with two sums of squares formulas. The sum of square of residuals can be calculated as:

$$S_{\text{res}} = \sum_i (y_i - g_i)^2 = \sum_i r_i^2 \qquad (18)$$

The total sum of squares is:

$$S_{\text{tot}} = \sum_i (y_i - \bar{y})^2 \qquad (19)$$

The general definition of $R_2$ score is:

$$R^2 = 1 - \frac{S_{\text{res}}}{S_{\text{tot}}} \qquad (20)$$

In the best case, the $S_{\text{res}} = 0$ and $R^2 = 1$. Generally, the $R^2$ located in the interval [0, 1] means the degree of fitting, and the closer the value is to 1, the better the performance of the model. Values of $R^2$ being negative occur when the predicted data doesn't fit the ground truth at all.

**Wasserstein distance.** Wasserstein distance[61] can measure the distance between different distributions. Intuitively, the origin of the Wasserstein distance lies in the optimal transport problem, which can be imagined as moving a pile of stones represented by probability distributions to form another target shape, while minimizing the cumulative distance of the movements. This is the central concern of the optimal transport problem. Let $\mu$ and $\nu$ be two distributions, and let $d$ be a way of calculating the distance, then the Wasserstein distance can be defined as:

$$W_p(\mu, \nu) = \inf_{\gamma \in \Gamma(\mu, \nu)} (\mathbb{E}_{(\mathbf{x}, \mathbf{y}) \sim \gamma} d(\mathbf{x}, \mathbf{y})^p)^{\frac{1}{p}} \tag{21}$$

where $\Gamma(\mu, \nu)$ is the set of all couplings of $\mu$ and $\nu$.

**Pearson correlation.** Pearson correlation measures the linear relationship between two vectors. Concretely, given two vectors $\mathbf{X}$ and $\mathbf{Y}$, Pearson correlation $\rho$ can be computed as

$$\rho_{\mathbf{X}, \mathbf{Y}} = \frac{\text{cov}(\mathbf{X}, \mathbf{Y})}{\sigma_{\mathbf{X}} \sigma_{\mathbf{Y}}} \tag{22}$$

where $cov(\mathbf{X}, \mathbf{Y})$ is the covariance between $\mathbf{X}$ and $\mathbf{Y}$ and $\sigma_{\mathbf{X}}$ denotes the standard deviation of $\mathbf{X}$.

**F1 score.** In this study, we adopt balanced F-score (F1 score). After the chi-square test on CellProfiler features extracted from ground truth morphology, we term the features with $p$ values $< 0.05$ as positive samples (regarded as significantly changed features), and term other features as negative samples. For generated morphology, we conduct the same test on generated CellProfiler features, then we acquire predicted positive samples and negative samples generated by each method. In this way, we can calculate true positive (TP), false positive (FP), and false negative (FN) for each method. The F1 score can be calculated as:

$$F_1 = \frac{2\text{TP}}{2\text{TP} + \text{FP} + \text{FN}} \tag{23}$$

**Reporting summary**
Further information on research design is available in the Nature Portfolio Reporting Summary linked to this article.

## Data availability

All the data used in this work is publicly available. The morphology images of CDRP dataset are from[23] and can be downloaded with the scripts from https://github.com/gigascience/paper-bray2017/blob/master/download_cil_images.sh. The corresponding L1000 gene expression data is collected from https://broad.io/rosetta/. The morphology images of JUMP dataset are collected from[24] and can be downloaded with the script from https://github.com/jump-cellpainting/datasets/. The corresponding L1000 gene expression data is collected from[46]. For LINCS, the bulk morphology images are collected from[25] and the corresponding gene expression data is downloaded from https://broad.io/rosetta/. For all the genes and related pathways involved in the aforementioned genetic perturbations, as well as the drug names, SMILES (Simplified Molecular Input Line Entry System), targets (if applicable), and MOAs (if applicable) related to drug perturbations, we have provided detailed information, which is included in the Supplementary Data 1–7. Source data is provided with this paper. Source data are provided with this paper.

## Code availability

Code is available at: https://github.com/biomap-research/MorphDiff. Notebooks for parts of analysis can be found at Code Ocean (https://doi.org/10.24433/CO.4389762.v1).

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

## Acknowledgements

This study was supported by the Chinese University of Hong Kong (CUHK; award numbers 4937025, 4937026, 5501517, 5501329 and SHIAE BME-p1-24 to Y.L.) and the IdeaBooster Fund (IDBF23ENG05 and IDBF24ENG06 to Y.L.) and partially supported by a grant from the Research Grants Council of the Hong Kong Special Administrative Region (Hong Kong SAR), China (project no. CUHK 24204023 and 14208525 to Y.L.) and a grant from the Innovation and Technology Commission of the Hong Kong SAR, China (project no. GHP/065/21SZ and ITS/247/23FP to Y.L.). This research was also funded by the Research Matching Grant Scheme at CUHK (award numbers 8601603 and 8601663 to Y.L.) from the Research Grants Council, Hong Kong SAR, China. We thank Ouyang Jian from BioMap for helping to draw small molecule structures with ChemDraw.

## Author contributions

X.W., Y.F., Y.G., Y.L., and L.S. conceived the study. X.W., Y.F. designed the methodology. X.W., Y.F., and C.F. implemented the models. X.W., Y.F., C.F., K.L., K.D., and YX.L. ran the baseline experiments. X.W. and Y.F. interpreted the results. X.W., Y.F. performed analysis on the data. X.W., Y.F., and Y.G. organized and processed the publicly available data. X.W. and Y.F. visualized the experimental results. L.S., Y.L. supervised the

research. X.W., Y.F. wrote the manuscript. Y.G. and Q.Y. help review the paper. All authors read and approved the final manuscript.

## Competing interests

The authors declare no competing interests.
