## [Transparent Peer Review file · Nature Communications]

Prediction of cellular morphology changes under perturbations with transcriptome-guided diffusion model

Corresponding Author: Professor Yu Li

Version 0:

Reviewer comments:

Reviewer #1

(Remarks to the Author)

The manuscript introduces MorphDiff, an innovative transcriptome-guided latent diffusion model designed for precise prediction of cellular morphology responses to various perturbations. Nonetheless, the current manuscript may render it difficult for readers without a machine learning background to fully grasp the concepts. To bolster accessibility and ensure that a wider audience, including those from diverse scientific disciplines, can fully benefit from the insights offered, it is strongly recommended that the authors revise the paper to enhance its clarity and readability.

Some minor issues

1. Numerous abbreviations have been utilized without providing their corresponding full terms. Notable examples include IMPA, FID, IS, and DMIT, among others.
2. The acronym MOAs (Mechanisms of Action) has been redundantly employed twice within the abstract, a redundancy that also appears throughout the manuscript.
3. The following sentence in the abstract is particularly challenging to interpret: "We also validated that complementary information provided by cell morphology generated by MorphDiff can help discover drugs with dissimilar structures but the same MOAs." It is recommended to rephrase for enhanced clarity.
4. The results from G2I (Gene to Image) and I2I (Image to Image) within the MorphDiff framework exhibit discrepancies. Could you provide an explanation?
5. The relationships among the terms MorphDiff(g), MorphDiff(g+i), G2I, and I2I require clarification to avoid any potential confusion.
6. The study utilizes CellProfiler features. Were other feature extraction techniques considered? Further, how can the effectiveness of the proposed methods be validated using alternative feature sets?
7. The distribution of CellProfiler features is compared against the ground truth, IMPA, and MorphDiff. The implications of this direct comparison remain unclear to me, as some features might complement each other in achieving the task. Could you elucidate the rationale behind this comparative approach?
8. The authors propose a LDM-based method and compare it with several conventional GAN-based img2img methods, such as StarGANv1, DMIT, and DRIT++. However, it is recommended that the authors also consider comparing stronger generative network paradigms, such as VQGAN [1] based on the Generative Pretrained Transformer (GPT) paradigm, and Masked Diffusion Transformers based on the Masked Image Modeling (MIM) paradigm, to validate the unique significance of the LDM in this task.
9. Both FID and IS rely on InceptionV3, a network pretrained on ImageNet, which classifies natural images. However, medical images, particularly cell images, are out-of-distribution for InceptionV3, rendering these metrics unreliable. Therefore, more reliable metrics such as CMMD [3] based on CLIP should be considered.

[1] Taming Transformers for High-Resolution Image Synthesis, cvpr21

[2] Masked Diffusion Transformer is a Strong Image Synthesizer, iccv23

[3] Rethinking FID: Towards a Better Evaluation Metric for Image Generation. cvpr23

(Remarks on code availability)

The code has been tested.

Reviewer #2

(Remarks to the Author)

Wang et al proposes a machine learning model (MorphDiff) for predicting cell morphology induced by compound or genetic perturbations given gene expression measured upon the corresponding perturbation. The model is applied to a public data from U2OS cells, and prediction performance is benchmarked relative to a set of baseline methods.

Results are presented that indicate that the model produces representations of perturbations that result in higher accuracy when used for retrieving mechanism of action (MOA) of a query set of drugs as compared to morphological representations using other models or when using gene expression profiles alone.

The model in effect represents perturbations by induced morphological features (conditional on gene expression), however, as noted by eg Way et al Cell Syst 2022, there are many perturbations which induce distinct morphological but not transcriptional changes and vice versa. Thus, it is unclear how general the findings in the present manuscript are as the examples presented are limited (eg only U2OS cells, limited targets/mechanisms). Summary statistics on the mechanisms/targets (drugs) and pathways (gene oe) represented in the data set would at least provide a context in which to evaluate the results. Also, how are different concentrations and time points across the assays handled?

The authors themselves note in the Discussion that the model can only be applied when a transcriptomics profile of the perturbation is available. As also noted, there is a large publicly available pool of perturbation gene expression data available in the L1000 collection, however, corresponding morphological data is not as available. Thus, more explicit examples of potential uses for the proposed method would benefit the presentation.

The OOD datasets was constructed by a random split, could the authors clarify in what sense this constitutes out-of-distribution? Did the authors consider identifying sets of perturbations with well established mechanisms/targets and performing leave-one-out?

In Fig 4a and Appendix Fig 9 and 10, MorphDiff appears to produce more distinct separation between DMSO control and treatments than ground truth, do the authors have any comments on this? Also, how do drugs with different targets relate to each other when projected together?

As the improved accuracy in MOA retrieval of the proposed model over e.g. gene expression is a key result, it would seem prudent to present the results for different values of n in the n-nearest neighbour approach used to indicate the robustness of the results.

The example given with clocortolone pivalate is difficult to follow. Colocortolone pivalate is also glucocorticoid (albeit prodrug).

(Remarks on code availability)

Reviewer #3

(Remarks to the Author)

(Remarks on code availability)

The code can be reproduced under the given instruction, including the model training code for MorphDiff and the code for reproducing the image results in the paper.

Reviewer #4

(Remarks to the Author)

The authors present a model for predicting cellular morphology changes due to perturbations using a variational autoencoder (VAE) and diffusion model. They applied their model to two large scale datasets that explored morphological responses to drug and genetic perturbations and compared results against several baseline models. Evaluation consisted of comparisons of CellProfiler features as well as the ability to classify mechanisms of action (MOA). They found that the proposed model achieved the best image quality in terms of Frechet Inception Distance (FID).

The problem is well motivated and many extensive evaluations are performed to demonstrate the model's effectiveness. The details of the method are also presented clearly, so it is relatively easy to follow how the method works. My main criticism is that the manuscript includes so many analyses that it does not always fully explain the importance of the results. It can also be too highly dependent on the figures in the supplemental material, making it difficult to follow the paper without jumping back and forth between documents. I will include more specific examples below:

1. Appendix figure 1 is important to appreciate the results and should be included in the main manuscript. No figures within the main manuscript include actual images of the inputs and outputs of the model, so this figure provides context for the statistical analyses and discussions.
2. Figures 2a and 3a include several metrics, but none of these results are described in much detail. The authors note that their method is top two for all of them, but do not discuss the implications for each of them. For instance, DMIT is significantly higher in terms of two out of the four metrics (IS and coverage) in Figure 3a, so it is unclear that MorphDiff is the superior method in this analysis. The authors do note that FID is the most important feature, but then do not discuss the fact that MorphDiff(g) significantly outperforms MorphDiff(g+i) in Figure 2a.
3. There are two versions of MorphDiff described in the text (gene+control and gene only). The terminology for these is inconsistent (e.g., G2I and I2I in the figure 1 caption vs. g+l and g in figures 2 and 3 vs. Gene2Img in the body). Throughout most of the manuscript, reference to MorphDiff does not specify a version, so it is unclear which is being used. Presumably, it is g+i but that isn't made clear and it also isn't clear why only that version would be included in these analyses.
4. It is a bit unclear how the five-channel images are handled in some of the analyses. In supplemental figure 1, outputs are presented in RGB, so it is unclear how this mapping was performed. Also, the authors note that only their model and IMPA can produce five-channel outputs, so it is also unclear how the other baseline models can be applied to five channel input or how their outputs can be compared to the five-channel ground truth in figures 2a and 3a.

While the trends in the results seem compelling, I have some concerns about how statistical testing is performed and evaluated:

5. Averaging the metrics in figures 2a and 3a is not meaningful. The authors note that FID "is the most prevalent and significant metric," so it isn't appropriate to create an aggregate score (i.e., the average) that treats each identically.
6. The performance of statistical hypothesis testing is appreciated, but there are many different comparisons in each of these analyses. It is unclear whether correction was performed (e.g., Bonferroni).
7. The authors state that they use a t test to "have a more holistic view of how close the MorphDiff-generated morphology aligns with the ground truth distribution." A t-test will only test whether there are significant differences between the centers of distributions. To investigate the similarity of distributions, it would be more appropriate to use a Kolmogorov Smirnov test or use a metric such as KL divergence.
8. The authors report using t-tests, but the distributions they are comparing do not appear to be normal. It is probably more appropriate to use nonparametric tests (e.g., Wilcoxon).
9. The main statistical claim that the authors make is that their model "outperforms the baseline method and gene expression-based retrieval by 29.1% and 9.7% respectively" (mentioned three times and the only numerical result in the abstract). While this statement is technically accurate, it is a bit deceiving because the metric the authors are comparing is a percentage (accuracy). Stating a percentage increase is therefore often interpreted as an accuracy that is 29.1 and 9.7 percentage points higher, which is not the case. It would be better to report the actual accuracy values from the different models and simply state that the improvement is statistically significant.
10. Some of the descriptions of trends in the data uses vague or imprecise wording. For instance "Figure 3b shows that most scatter points lie above the diagonal, signifying that MorphDiff outperforms IMPA in most of the 3000 samples. Meanwhile, the y-coordinate corresponding to the highest density is close to one." Use precise statistics rather than vague and subjective words like "most" or "close."

The manuscript also requires significant copy-editing. There are grammatical issues throughout, including a typo in the title ("celluar"). I'll list several specific issues below with examples, but this list is not exhaustive:

11. Acronyms used before they are defined (e.g., "fid" on page 4), defined multiple times (e.g., "MOA" is defined twice in the abstract), or not defined at all (e.g., "SMILES").
12. Colloquial or vague language (e.g., "can be pretty noisy").
13. Incorrectly used words (e.g., "our framework corporates" should be "incorporates"; "in the following article" should probably be "section" or "analysis").
14. Plurality mismatch (e.g., "training and ID set covers" should be "training and ID sets cover").
15. Missing articles (e.g., "architecture augmented with attention mechanism" should say "...with an attention mechanism"; "We selected most predictable morphological features" should say "...the most predictable...").
16. Awkward phrasing (e.g., "achieves an excellent performance").
17. Editorializing (e.g., "which leads to a meager coverage score").

Other comments:

18. In figure 4c, in one of the differences marked as significant (CDK2) IMPA is higher, but it is marked with an asterisk indicating MorphDiff is higher.

19. The authors mention that they "expand each channel to 3 dimensions to match the input shape of VGG," but it is unclear how this was achieved. Were channels duplicated?

Minor comments:

20. In appendix figure 1, including the control image would provide helpful context for the results.

21. Figure 1d should be a separate table rather than a subfigure.

22. It is unclear why q_i needs to be defined since it is identically p_i .

23. The better performing model includes Y_{DMSO} , so it probably makes sense to have that version of the function in equation 1.

24. "time t" should instead say "step t."

25. It can be a little hard to read when describing statistical tests together with the results (e.g., second paragraph of 2.4). It would be better to describe the tests with the metrics in the methods section and only mention significance in the results.

(Remarks on code availability)

Version 1:

Reviewer comments:

Reviewer #2

(Remarks to the Author)

The revision by Li et al is extensive and provides additional results to demonstrate the superiority of the proposed methodology over comparators.

While I appreciate the significant effort the authors have made to revise the manuscript, I remain concerned about the lack of a clear context in which to interpret the results, such as for which of the investigated MOAs/targets morphology prediction is most accurate. This issue is closely tied to potential use cases and limits the applicability of the findings.

Some minor concerns:

Lines 203–205: The authors mention that the K-S test and KL divergence (Kullback-Leibler?) were 'experimented' with to further demonstrate the superiority of their method. Could the authors clarify how a significant K-S test or KL divergence would substantiate such claims of superiority? Additionally, the term 'experimenting' with statistical evaluations might imply exploratory analysis, which could be perceived as inconsistent with best practice.

The authors state that the K-S test was used to verify that the OOD datasets had different distributions. Could the authors specify which variable(s) the K-S test was applied to, or clarify if it was applied in a multivariate context?

(Remarks on code availability)

Reviewer #4

(Remarks to the Author)

The authors have made substantial changes to the manuscript in response to the previous review. I believe it is greatly improved, but I have some additional comments.

I appreciate the inclusion of example images in figure 1d, but its current presentation is not ideal. First, it does not fit with the rest of figure 1 as the rest of the figure shows flowcharts of the model architecture. The visualization is a result and works better in conjunction with the results from figure 2a because it can provide examples that demonstrate the trends shown in the metrics. For example, the explanation about why DMIT has higher density could be supported by pointing to specific examples. More details should also be included in the figure and/or caption (e.g., is there significance to the four examples chosen? Are there qualities of the proposed model's outputs that you are trying to highlight?).

I have some additional comments about the projection into three channels. First, how were these weights determined? Is there a biological justification for them or is it optimized in some way? The description of the projection is also overly complicated with several equations and variables defined that aren't used elsewhere. It is sufficient to just say the channels are $\{(1 * \text{DNA}), (1 * \text{ER} + 0.4 * \text{AGP} + 0.4 * \text{Mito}), (1 * \text{RNA} + 0.4 * \text{AGP} + 0.2 * \text{Mito})\}$ or use a matrix:

$C2 = [1, 0, 0, 0, 0;$

$0, 1, 0, 0, 4, 0.4;$

$0, 0, 1, 0, 4, 0.2] * C1$

Finally, is there a reason why this projection was used for the baseline models rather than adjusting the architecture to accept five channel input? It shouldn't be too difficult to modify these models to accept five channels. While pretrained model weights could not be used, I believe the models are being retrained for this task anyway. It seems like a significant disadvantage for these baselines to not be able to use the full input data.

Minor comments

The last two paragraphs of the introduction section largely summarize the results. The manuscript would flow better if the introduction were limited to setting up the goals and hypotheses and leaving the results to later sections.

Predicting cell morphological responses to perturbations using generative modeling is cited as biorxiv, but it was recently published in Nature Communications

The acronym DMSO appears several times (figure 1d, figure 2b, and section 4.1), but it is not defined and it also is not mentioned in the figure captions.

I looked over the responses to reviewer 1's comments and most of them seem to be well addressed. The only one that I think has some issues is the response to point 4:

4. The results from G2I (Gene to Image) and I2I (Image to Image) within the MorphDiff framework exhibit discrepancies. Could you provide an explanation?

I'm not completely clear on what the reviewer was asking because they weren't specific about what they meant by "discrepancies." I think the authors took the right approach here in trying to explain the differences in performance between the two models. I do think there are a couple of issues in their response, however.

The main source of confusion stems from the trends shown in response figure 1. In the results for the two held out datasets, we see opposite trends between G2I and I2I (i.e., I2I is higher for density and coverage and lower for FID and CMMD on the JUMP dataset, all of which are reversed in the CDRP dataset). While it is expected that the two methods would have differences in performance, it is unexpected that the trends between the two methods would reverse between these two datasets. This reversal suggests that there is something different about the datasets that makes the methods perform differently on them.

In the authors' revision, they include a discussion of the differences in performance between these methods on the two datasets in sections 2.2 and 2.3, respectively. These descriptions are independent and don't acknowledge that they are complete opposites of each other. For example, in section 2.2 they write:

"MorphDiff(I2I) performs better on the metrics of density and coverage, suggesting that it generates a more diverse set of samples, and the generated distribution better aligns with the ground truth."

while in section 2.3 they write:

"MorphDiff(G2I) outperforms on density, and coverage.. [meaning it] excels in generating samples of higher diversity and closer to the ground-truth distribution."

Presenting the results like this is confusing because it seems to attribute opposite strengths to the two methods depending on the dataset. This discrepancy makes it difficult to connect experimental results to the designs of the underlying models.

In the discussion section, the authors acknowledge the reversal in trends, but do not investigate it further. Instead, they speculate on factors that could have caused it:

"One reason for this discrepancy may be the quality of the control morphology images used as input in the I2I mode.

Although the control cell morphology provides valuable information about cellular components, it can also be affected by batch effects and technical variations in Cell Painting experiments."

It would be much more valuable to actually look at the quality of the images and see if there are differences between those provided by JUMP and CDRP. The authors could also provide specific examples to show where the inclusion of an input image had opposite effects for the two datasets.

(Remarks on code availability)

Version 2:

Reviewer comments:

Reviewer #2

(Remarks to the Author)

I appreciate the efforts by Wang et al in clarifying methodology and explicitly stating some of the limitations (albeit general to all machine learning) of the proposed methodology.

(Remarks on code availability)

Reviewer #4

(Remarks to the Author)

I appreciate the authors' thorough revisions to address my previous comments. I am satisfied with their responses and recommend publication.

(Remarks on code availability)

Prediction of cellular morphology change under perturbations with transcriptome-guided diffusion model

Xuesong Wang^{*†1,2}, Yimin Fan^{*†1,2}, Yucheng Guo^{†1}, Chenghao Fu², Kinhei Lee², Khachatur Dallakyan², Yaxuan Li², Qijin Yin¹, Yu Li^{‡2,4,5,6,7}, and Le Song^{‡1,3}

¹BioMap Research, California, USA

²Department of Computer Science and Engineering, CUHK, Hong Kong SAR, China

³Mohamed bin Zayed University of Artificial Intelligence, Abu Dhabi, UAE

⁴The CUHK Shenzhen Research Institute, Hi-Tech Park, Nanshan, Shenzhen, 518057, China

⁵Institute for Medical Engineering and Science, Massachusetts Institute of Technology, Cambridge, MA, USA

⁶Wyss Institute for Biologically Inspired Engineering, Harvard University, Boston, MA, USA

⁷Broad Institute of MIT and Harvard, Cambridge, MA, USA

Response to reviewers

Manuscript ID: NCOMMS-24-43912

Manuscript title: Prediction of cellular morphology change under perturbations with transcriptome-guided diffusion model

We really appreciate the editor and all the reviewers for their constructive and thoughtful comments, which are helpful for us to refine our manuscript. We have responded to the comments each by each in detail. In the revised version of our manuscript, we have thoroughly addressed all the concerns of the reviewers by conducting additional benchmarking, statistical tests, analyses, evaluations, and revisions. We believe that the revised manuscript will become more comprehensive and clarified with the insightful and valuable suggestions of all reviewers.

Blue: our responses to the reviewer's comments	Orange: references to the manuscript
Gray: reviewers' comments	Black: pointers to figures, tables, and sections

Summary of responses to reviewer comments

In response to the reviewers' comments, we present a summary of our key updates and responses below:

- 1. Extensive benchmarking and analysis including new datasets, baseline methods, and evaluation metrics:** We have added additional evaluation and benchmarking covering new datasets, baseline methods and evaluation metrics. For datasets, we collected a new drug perturbation dataset on the A549 cell line to further evaluate the robustness of our methods on different cell lines. We also split the datasets by the drug targets or MOAs and performed leave-one-out evaluations, to validate the effectiveness of our method in predicting the cell morphology when the drug treatment has significantly different targets or MOAs from the training set. For baseline methods, we added VQGAN [1] and Masked Diffusion Transformers [2] to ensure that the baseline methods cover diverse generative modeling paradigms. For evaluation metrics, we added a new evaluation metric CMMD [3] which is based on the CLIP model [4]. We comprehensively analyzed the benchmarking results, and found that MorphDiff performs best compared with all baseline

^{*}Work was done while interning at BioMap.

[†]Equal first authorship.

[‡]Corresponding Author. Email: liyu@cse.cuhk.edu.hk and songle@biomap.com

methods across diverse evaluation datasets and metrics, highlighting the superiority of our model in terms of generation quality and generalizability.

2. **Comprehensive evaluation and analysis on both modes of MorphDiff:** We have added comprehensive evaluations and analysis on two modes of MorphDiff, MorphDiff(G2I) and MorphDiff(I2I). In all the downstream analysis, we evaluated these two modes and analyzed their advantages respectively. Our findings indicate that both modes of MorphDiff meet the criteria for an effective generative model, achieving low FID, CMMD and high scores in IS, density, and coverage. In most application scenarios, either mode is suitable, except when users specifically aim to analyze morphological feature changes with respect to certain control cells.
3. **Detailed analysis and evaluation on the MOA retrieval and matching:** We have performed more comprehensive evaluation of the MOA retrieval and matching application of our model. In MOA retrieval, we evaluated different hyper-parameter settings to demonstrate the robustness of our method. We also added MOA matching analysis evaluated with two additional metrics, folds of enrichment and mean average precision. Furthermore, we added two application cases in which our method helped to identify drugs with different structures but annotated with the same MOAs. Our findings demonstrate the potential application of our method in MOA retrieval and phenotypic drug discovery.
4. **Rigorous statistical analysis on the evaluation datasets and results:** We performed rigorous statistical analysis of our evaluation datasets and results. For evaluation datasets, we performed the Kolmogorov-Smirnov test to demonstrate the significant differences between the training and OOD dataset distribution. For evaluation results, we have changed the t-test analysis used in several sections to the Wilcoxon signed-rank test, which is a non-parametric test and does not require the Gaussian assumption on the data distributions. We also performed Bonferroni correction to enhance the statistical significance of our analysis. In some analyses, as requested by the reviewers, we conducted analyses using different statistical tests. Our rigorous statistical tests suggest that MorphDiff demonstrates statistically significant and consistent advantages compared to the baseline methods.
5. **Exploratory analysis on the L1000 gene expression profiles without ground-truth Cell Painting images:** As the current public L1000 database is much larger than the Cell Painting database, we performed an additional evaluation of the L1000 gene expression profiles treated by drugs that have not been applied in Cell Painting assays. We evaluated the fidelity of our generated cell morphology with MOA matching, i.e., whether the generated cell morphology would be similar to ground-truth cell morphology treated by drugs with the same MOA. Our findings suggest that MorphDiff can generalize reasonably well to these L1000 gene expression profiles, while the baseline method has a similar performance as the random baseline.
6. **Careful editing of the writing issues of the manuscript** We have reorganized the figures and method descriptions throughout the manuscript to improve the readability. Furthermore, we clarified the definitions of several terms and metrics as requested by the reviewers. We carefully edited the manuscript for grammar and style issues.

Datasets we use in the reply letter and revised manuscript

1. **JUMP:** We collected 130 perturbations and a control set that include both gene expression data [5] and morphology images from [6]. We split the dataset into three sets, which are training set, ID (in-distribution) set, and OOD (out-of-distribution) set. We randomly picked 10 percent of perturbations as OOD sets, resulting in 13 perturbations. The remained parts were randomly split into training set (90 percent) and ID set(10 percent), which indicates that the training set and ID set contain both control and perturbation data. We termed them as **JUMP Training set**, **JUMP ID set** and **JUMP OOD set**.
2. **CDRP:** We collected 959 perturbations and control set from CDRP [7; 8]. Following the same split principle as JUMP dataset, we obtained 96 perturbations for OOD set, and 863 perturbations along with the control for training set and ID set. We termed them as **CDRP Training set**, **CDRP ID set** and **CDRP OOD set**. We additionally selected 10 targets (68 drugs annotated with 35 MOAs) from CDRP for MOA retrieval analysis, we termed it as **CDRP Target_MOA dataset**.

3. **LINCS**: To construct a dataset for model validation, we collected the bulk gene profiles from [8] (LINCS part) and downloaded the corresponding images from [9]. In this dataset, we selected 61 drugs that have sufficient images, along with available MOA and target labels. The components of these images are the same as those in the U2OS dataset. We first divided the data into 10 folds according to the 10 targets corresponding to 21 drugs. We termed it as **LINCS Target leave-one-out set**. For each experiment, we trained the model on another 9 folds and validated it on the remaining fold. We conducted experiments in a similar manner for the 10 MOAs (42 drugs) and termed it as **LINCS MOA leave-one-out set**.

Reviewer 1

The manuscript introduces MorphDiff, an innovative transcriptome-guided latent diffusion model designed for precise prediction of cellular morphology responses to various perturbations. Nonetheless, the current manuscript may render it difficult for readers without a machine learning background to fully grasp the concepts. To bolster accessibility and ensure that a wider audience, including those from diverse scientific disciplines, can fully benefit from the insights offered, it is strongly recommended that the authors revise the paper to enhance its clarity and readability.

Re:

Thanks for your insightful feedback and detailed and constructive comments. We have revised our manuscript and improved the quality and clarity so that more readers from diverse scientific disciplines can grasp the insights more easily. We added more explanations on the evaluation results as well as some intuitive explanations of our latent diffusion model framework to make these concepts easier to understand for those without a machine learning background. Some revisions we made to the manuscript are shown below, while the full revisions can be found in the revised manuscript.

...FID measures the distance between the distributions of ground-truth and generated images in the feature space, using features extracted from Inception V3 [10]. The Inception Score assesses the quality of the generated images based on how effectively they can be classified by the Inception V3 model. Both FID and IS utilize Inception V3 [10], which is pre-trained on ImageNet [11]. To enhance the reliability and robustness of our comparisons, we incorporated CMMD, which calculates the Maximum Mean Discrepancy (MMD) distance based on features extracted by a CLIP model pre-trained on 400 million image-text pairs containing more complex scenes [3]. We also included the density and coverage metrics, as these metrics are robust against outlier samples and could better measure the diversity and fidelity of the generated samples [12]...

...Intuitively, by learning to recursively add Gaussian distributions to noisy images, diffusion models could generate high-quality images. Concretely, we utilize the Morphology Variational Autoencoder (MVAE, see Section 4.2) to compress the high-dimensional morphology of cells into low-dimensional embeddings, and then train a latent diffusion model [13] using these embeddings with gene expression profiles as the condition...

...The latent diffusion model is trained to generate cell morphology images conditioned on the perturbed gene expression profiles...

..LDM is implemented with denoising U-Net architecture (based on convolutional neural networks) augmented with an attention mechanism [14]. As input condition, the L1000 gene expression is combined with the parameters of the LDM model, specifically the key and value of the attention mechanism in LDM. Training an LDM minimizes the variational upper bound, which serves as a proxy to reduce the noise prediction error and ensures that the generated samples match the ground-truth distribution...

Some minor issues

1. Numerous abbreviations have been utilized without providing their corresponding full terms. Notable examples include IMPA, FID, IS, and DMIT, among others.

Re:

Thank you for your comment. We have added the corresponding full terms to the places where they first appeared in the revised manuscript. We also listed their full terms here:

- IMPA: IMage Perturbation Autoencoder
- DRIT++: Disentangled Representation for Image-to-Image Translation
- VQGAN: Vector Quantised Generative Adversarial Network
- MDTv2: Masked Diffusion Transformer

- DMIT: Disentanglement for Multi-mapping Image-to-Image Translation
- FID: Frechet Inception Distance
- IS: Inception Score
- SMILES: Simplified Molecular Input Line Entry System

In other cases, such as MorphNet [15] and StarGANv1 [16], the authors referred to them without providing the full terms. We clarify that both terms are generative methods used for benchmarking evaluation. We added Appendix Table 7 (Response Table 1) in the revised Supplementary materials to display the acronyms and full names.

Acronyms	Full name
MOA	Mechanism Of Action
SMILES	Simplified Molecular Input Line Entry System
FID	Frechet Inception Distance
IS	Inception Score
IMPA	IMage Perturbation Autoencoder
DRIT++	Disentangled Representation for Image-to-Image Translation
VQGAN	Vector Quantised Generative Adversarial Network
G2I	Gene To Image
I2I	Image To Image
MVAE	Morphology Variational AutoEncoder
LDM	Latent Diffusion Model
LPIPS	Learned Perceptual Image Patch Similarity
CMMD	CLIP embeddings and Maximum Mean Discrepancy

Response Table 1: The acronyms and full names appear in the paper.

2. The acronym MOAs (Mechanisms of Action) has been redundantly employed twice within the abstract, a redundancy that also appears throughout the manuscript.

Re:

Thanks for pointing out this problem. We have removed redundant full names for MOAs throughout the manuscript, except the place where the acronym MOAs first appeared in the abstract and the main text.

Investigating the cell morphology change after perturbations with high-throughput image-based profiling is of growing interest, considering its wide applications in phenotypic drug discovery, including MOAs (**Mechanism Of Actions**) prediction,...

...These applications include the prediction of MOAs (**Mechanism Of Actions**), compound bio-activity,...

3. The following sentence in the abstract is particularly challenging to interpret: "We also validated that complementary information provided by cell morphology generated by MorphDiff can help discover drugs with dissimilar structures but the same MOAs." It is recommended to rephrase for enhanced clarity.

Re:

Thank you for your suggestion. We have rewritten this sentence as:

We also confirmed that MorphDiff has the potential to generate similar perturbed cell morphologies in cases where the drugs have dissimilar structures but the same MOAs, with patterns closely resembling the ground truth.

4. The results from G2I (Gene to Image) and I2I (Image to Image) within the MorphDiff framework exhibit discrepancies. Could you provide an explanation?

Response Figure 1: a. Performance benchmarking on JUMP OOD set. b. Performance benchmarking on CDRP OOD set. MorphDiff(G2I) denotes generating the perturbation morphology images with only the perturbation gene expression. MorphDiff(I2I) denotes generating the perturbation morphology with the perturbation gene expression and a control reference image. Linear normalization was conducted on the reciprocals of FID, CMMD, and the other three metrics separately to convert their values to the range of 0 to 1. A larger value indicates better performance.

Re:

Thanks for your comments. Basically, G2I (Gene to Image) mode is that MorphDiff generates cell morphology images straightforwardly conditioned on the gene expression, while I2I (Image to Image) generates cell morphology images based on a control cell morphology image (cultured with DMSO), conditioned on the gene expression. Intuitively, the I2I mode predicts how the morphology of a cell without perturbation transits to the perturbed cell morphology. Thus, the results of the I2I mode can be affected by the control cell morphology images, while the generated results of G2I are only determined by the gene expression. The control images could provide additional information to the model. However, they may also introduce experimental technical variations, such as *batch effect* (well-position effects and plate effects), and addressing them is still an open research problem [17]. The variation could confound the downstream analysis, which could lead to such discrepancies. To clarify for readers, we analyzed the discrepancies observed during comparisons and summarized the reasons in the Discussion Section. The revised performance comparison is as Response Figure 1 shows. We added the analysis in the revised Section 2.2 as follows:

For the two modes of MorphDiff, we can see that MorphDiff(I2I) performs better on the metrics of density and coverage, suggesting that it generates a more diverse set of samples, and the generated distribution better aligns with the ground truth. MorphDiff (G2I) scores higher in FID(-1) and CMMD(-1), indicating higher quality and fidelity. This highlights a trade-off where MorphDiff(G2I) prioritizes quality over diversity, while MorphDiff(I2I) emphasizes the diversity of generated outputs. In general, based on these five metrics, MorphDiff demonstrates more balanced performance in terms of generalization, fidelity, and diversity compared to other generative models, resulting in the best generation quality.

We also added the analysis in the revised Section 2.3 as follows:

For the two modes of MorphDiff, MorphDiff(G2I) outperforms on density, and coverage, while MorphDiff(I2I) performs better on FID(-1), IS and CMMD(-1). In this case, MorphDiff(G2I) excels in generating samples of higher diversity and closer to the ground-truth distribution, while MorphDiff(I2I) is more effective in generating high-quality samples. Ideally, an effective generative model should achieve low FID and CMMD, and high scores in IS, density, and coverage to ensure that it produces outputs of high quality and fidelity, while also being diverse and closely aligned with the ground truth distribution. MorphDiff (both modes) outperforms baseline methods considering these criteria, demonstrating its comprehensive generative capability and stability.

We added explanations of this discrepancy in the Discussion Section as follows:

MorphDiff can be applied in two modes: I2I and G2I. The G2I mode uses gene expression solely as input, while the I2I mode incorporates both gene expression and control cell morphology. Though

both modes outperform the baseline methods, we observed a discrepancy in quantitative performance between the two modes. For example, on the JUMP dataset, the I2I mode generates samples with greater diversity, while the G2I mode produces samples of higher quality. Conversely, on the CDRP dataset, the opposite is observed. On the LINCS dataset, the I2I mode performs better than the G2I mode except for the IS score. One reason for this discrepancy may be the quality of the control morphology images used as input in the I2I mode. Although the control cell morphology provides valuable information about cellular components, it can also be affected by batch effects and technical variations in Cell Painting experiments. Addressing these technical factors remains an open problem for Cell Painting images [17]. Nevertheless, both modes of MorphDiff meet the criteria for an effective generative model, achieving low FID, CMMD, and high scores in IS, density, and coverage. In most application scenarios, either mode is suitable, except when users specifically aim to analyze morphological feature changes with respect to certain control cells, as discussed in Figure 3c and Figure 4b; in this case, MorphDiff (I2I) is the more suitable option.

5. *The relationships among the terms MorphDiff(g), MorphDiff(g+i), G2I, and I2I require clarification to avoid any potential confusion.*

Re:

Thanks for your suggestion. We added clearer clarifications in the main text. In the original paper, MorphDiff(G2I) is equivalent to G2I/MorphDiff(g) and MorphDiff(I2I) is equivalent to I2I/MorphDiff(g+i). We have unified the names as MorphDiff(G2I) and MorphDiff(I2I) in all figures and text for clarity. We added a clarification in the revised Section 2.1 as follows:

In the first mode (referred as MorphDiff(G2I)), MorphDiff(G2I) takes the L1000 gene expression as the condition and denoises the corresponding cell morphology images from random noise distribution. In the second mode (referred to as MorphDiff(I2I)), MorphDiff(I2I) takes the L1000 gene expression for one specific perturbation as the condition and transforms the morphology images from the control set to the predicted morphology images for that perturbation.

6. *The study utilizes CellProfiler features. Were other feature extraction techniques considered? Further, how can the effectiveness of the proposed methods be validated using alternative feature sets?*

Re:

Thanks for your comment. Sorry for the potential confusion in the original manuscript. In our original study, we have already utilized CellProfiler [18] and DeepProfiler [17] in the manuscript. For example, we used CellProfiler features for the revised Figure 2(b-e), revised Figure 3(b-c), revised Figure 4b and, DeepProfiler embeddings for the revised Figure 3d and revised Figure 4(c-g).

First, the main rationale behind using the CellProfiler features and DeepProfiler embeddings can be described as follows. CellProfiler produces an interpretable feature set, including but not limited to the intensity, texture, and granularity of different channels in the cell morphology images. Due to their high interpretability, these features can be used in a series of downstream analysis pipelines such as BioMorph [19] and Cytominer[20] to discover features highly associated with cell states and drug targets. Therefore, we evaluated whether the output generated by MorphDiff is consistent with the ground truth in these interpretable feature sets. On the other hand, DeepProfiler is based on a CNN network, producing dense embeddings with respect to cell morphology. Unlike CellProfiler, the individual features of DeepProfiler are not directly interpretable. DeepProfiler is suitable for performing MOA classification and morphology retrieval, as it learns meaningful low-dimensional embeddings with deep learning [17]. Therefore, we used the DeepProfiler embeddings in the revised Figure 4(c-g) to quantify the phenotypic divergence between the target-level morphological responses, and to identify the drug MOAs according to the drug morphological responses. We also added a brief introduction of these two sets of features in the revised manuscript Method Section:

We utilized widely used single-cell morphology feature extraction methods, including CellProfiler [18] and DeepProfiler [17]. CellProfiler is based on traditional image feature extraction methods, which could produce interpretable morphological features, while DeepProfiler is based on pre-trained deep neural networks to extract dense low-dimensional embeddings of cell morphology images.

Furthermore, following your suggestion on evaluating using additional feature sets, besides CellProfiler and DeepProfiler embeddings, we used an additional cell morphology feature extraction method CLOOME [21], a recent

deep learning method for extracting dense embeddings from cell morphology images. We validated the effectiveness of our model using the newly added metrics: Folds of Enrichment and Mean Average Precision, as proposed in [17; 22]. We added the detailed descriptions of these two metrics in the revised Method Section as follows:

Mean Average Precision Mean Average Precision (mAP) is an evaluation metric assessing the probability that samples of interest will rank highly on a list of samples rank ordered by some distance or similarity metric [22], such as cosine, Euclidean, Mahalanobis, etc. The mAP framework is agnostic to the choice of this distance metric. Given a sample as q (generated cell morphology image), annotated with l_q (drug, MOA, or genetic perturbation). Assume that there are n ground-truth cell morphology images. TP_k represents *True Positive* when ranking the top k ground-truth samples by distance with q , while TN_k , FP_k , and FN_k are defined similarly. Then the precision and recall at each rank can be calculated as $P_k = \frac{TP_k}{TP_k + FP_k}$ and $R_k = \frac{TP_k}{TP_k + FN_k}$. We can then calculate AP and mAP :

$$AP_q = \sum_k (R_k - R_{k-1}P_k) \quad (1)$$

$$mAP = \sum_{q=1}^n \frac{AP_q}{n} \quad (2)$$

Folds of Enrichment Folds of Enrichment is an evaluation metric assessing whether the perturbed cell morphology and ground-truth cell morphology belonging to the same MOA or genetic perturbation will have high similarity [17]. Concretely, for each generated cell morphology image, we calculated the odds ratio of a one-sided Fisher’s exact test. The test is calculated using a 2×2 table. The first row contains the number of ground-truth cell morphology treated with the same treatment

Response Figure 2: Folds of enrichment and mean average precision of MorphDiff and IMPA computed with the DeepProfiler and CLOOME embeddings. The threshold in folds of enrichment is set to 1%.

(drug, drug with the same MOA or genetic perturbation) (positive matches) and different treatments (negative matches) at a selected threshold of the list of results. The second row is the same, but for the treatments below the threshold (the rest). The odds ratio is the sum of the first row divided by the sum of the second row. It estimates the likelihood of observing the generated cell morphology are top similar to the ground-truth cell morphology with the same perturbation.

For each generated cell morphology, we use these two metrics to measure whether the top similar ground-truth cell morphology has the same perturbations. We used cosine similarity as the similarity measure. The threshold for the folds of enrichment is set to 1%. We conducted the analysis on JUMP OOD set and CDRP OOD set.

We showed the results of the additional analysis with CLOOME embedding set in Response Figure 2. We found that though the scores are not exactly the same for DeepProfiler embeddings and CLOOME embeddings, the trend for both methods is consistent, which demonstrates that the effectiveness and robustness of MorphDiff can be validated across multiple different feature sets.

7. The distribution of CellProfiler features is compared with ground truth, IMPA, and MorphDiff. The implications of this direct comparison remain unclear to me, as some features might complement each other in achieving the task. Could you elucidate the rationale behind this comparative approach?

Re:

Thank you for your comments. Regarding your comment on why we directly compare the distribution of CellProfiler features, CellProfiler is a classical and widely used image feature extraction tool for Cell Painting image analysis [18]. The CellProfiler features have been applied in drug discovery [23] and analysis of genetic variation effect [24] from Cell Painting images. Furthermore, CellProfiler features are also the basis of several downstream analysis tools, such as BioMorph [19] and Cytominer [20]. As our objective is to develop a powerful in-silico method to simulate high-fidelity cell morphological responses to perturbations, it is essential that the CellProfiler features from our generated images are as close to the ground-truth CellProfiler features as possible, making our method more interpretable and reliable in potential downstream applications. More intuitively, for example, if the imaging experiments reveal that the mean intensity of the DNA channel increases after a certain perturbation, we may be interested in whether our predictions have the same trend. If the trend is the same, the model can more likely be trusted in downstream analysis and applications that involve the CellProfiler features. Therefore, we directly compare the feature distribution of the generated output with ground truth. The motivation for this step is further described in the revised manuscript as follows:

CellProfiler is used to extract features from single-cell morphology images, allowing for a comparison between the features of ground-truth cell morphology and those of generated cell morphology. This evaluation mainly aims to assess the similarity between ground-truth and generated cell morphology images in the interpretable feature sets because these features are usually involved in downstream analysis with respect to cell morphology images such as BioMorph [19] and Cytominer[20]

Regarding the problem you mentioned that some features might complement with each other, we agree that some features indeed have similar functions or meanings. For example, there are features describing the mean intensity of a particular channel as well as features describing the sum intensity of a particular channel. We have noticed this phenomenon in our initial analysis. In our analysis including Figure 2(b-e), Figure 3(b-c) and Figure 4b in the manuscript, we have filtered out these features with similar functions to reduce their impact on the results. This step was described in the original manuscript and is further highlighted in the revised manuscript as follows:

Features with NaN values and zero variance are removed. Meanwhile, features with duplicated meaning and functions are also removed (e.g., sum and mean of intensity).

8. The authors propose a LDM-based method and compare it with several conventional GAN-based img2img methods, such as StarGANv1, DMIT, and DRIT++. However, it is recommended that the authors also consider comparing stronger generative network paradigms, such as VQGAN based on the Generative Pretrained Transformer (GPT) paradigm and Masked Diffusion Transformers based on the Masked Image Modeling (MIM) paradigm, to validate the unique significance of the LDM in this task.

Re:

Thanks for suggesting conducting comparisons with stronger generative network paradigms. Following the suggestion, we added comparison experiments with VQGAN [1] and Masked Diffusion Transformers (MDTv2) [2].

The added results can be found in Response Figure 1. We observed that the newly added methods VQGAN and MDTv2 underperform MorphDiff in terms of all five evaluation metrics. Overall, MorphDiff(G2I) and MorphDiff(I2I) achieve the best performance. We also modified Figure 2a and Figure 3a in the manuscript accordingly as Response Figure 1 shows. We also added these two methods in the manuscript as follows:

VQGAN VQGAN(Vector Quantised Generative Adversarial Network) [25] combines the inductive bias of CNNs with the expressive power of transformers to model and synthesize high-resolution images. It employs CNNs to learn a context-rich vocabulary of image components and uses transformers to efficiently model their composition in high-resolution images. VQGAN is well suited for conditional synthesis tasks, accommodating both non-spatial information, like object classes, and spatial information.

MDTv2 MDTv2 (Masked Diffusion Transformer) [26] introduces a masked latent modeling scheme to learn contextual relationships among semantic parts of an image. During training, MDT masks certain tokens in the latent space. An asymmetric diffusion transformer predicts these masked tokens from the unmasked ones with the diffusion generation process. MDT can reconstruct complete image information from incomplete input, learning relationships among image tokens. A further improved MDT with a more efficient macro network structure and training strategy is named MDTv2 [2].

9. Both FID and IS rely on InceptionV3, a network pretrained on ImageNet, which classifies natural images. However, medical images, particularly cell images, are out-of-distribution for InceptionV3, rendering these metrics unreliable. Therefore, more reliable metrics such as CMMD based on CLIP should be considered.

Re:

Thank you for pointing out that. We added the results of the CMMD metrics [3] as shown in Response Figure 1 as well as in the revised manuscript Figure 2a and 3a. We showed that both modes of MorphDiff achieve the best performance in this metric, highlighting the effectiveness of our methods. We also added the description of the CMMD metric in the revised manuscript Method Section:

CLIP embeddings and Maximum Mean Discrepancy The CMMD (CLIP-MMD) metric represents the squared MMD (Maximum Mean Discrepancy) distance between the CLIP embeddings [27] of the ground-truth image set and those of the generated image set [3]. CLIP embeddings are extremely effective in representing the diverse and intricate content in images, benefitting from training an image encoder and a text encoder together using 400 million image-text pairs. Intuitively, given the $\mathbf{E} = \{\mathbf{E}_1, \mathbf{E}_2, \dots, \mathbf{E}_m\}$ as CLIP-encoded embeddings of ground truth images, and $\mathbf{E}' = \{\mathbf{E}'_1, \mathbf{E}'_2, \dots, \mathbf{E}'_n\}$ representing CLIP-encoded embeddings of generated images, an unbiased estimator of $\hat{dist}_{MMD}^2(\mathbf{E}, \mathbf{E}')$ is given by:

$$\hat{dist}_{MMD}^2(\mathbf{E}, \mathbf{E}') = \frac{1}{m(m-1)} \sum_{i=1}^m \sum_{i \neq j}^m k(\mathbf{E}_i, \mathbf{E}_j) + \frac{1}{n(n-1)} \sum_{i=1}^n \sum_{i \neq j}^n k(\mathbf{E}'_i, \mathbf{E}'_j) - \frac{1}{mn} \sum_{i=1}^m \sum_{i=j}^n k(\mathbf{E}_i, \mathbf{E}'_j) \quad (3)$$

Reviewer 2

1. Wang et al proposes a machine learning model (MorphDiff) for predicting cell morphology induced by compound or genetic perturbations given gene expression measured upon the corresponding perturbation. The model is applied to a public data from U2OS cells, and prediction performance is benchmarked relative to a set of baseline methods.

Results are presented that indicate that the model produces representations of perturbations that result in higher accuracy when used for retrieving mechanism of action (MOA) of a query set of drugs as compared to morphological representations using other models or when using gene expression profiles alone.

The model in effect represents perturbations by induced morphological features (conditional on gene expression), however, as noted by eg Way et al Cell Syst 2022, there are many perturbations which induce distinct morphological but not transcriptional changes and vice versa. Thus, it is unclear how general the findings in the present manuscript are as the examples presented are limited (eg only U2OS cells, limited targets/mechanisms). Summary statistics on the mechanisms/targets (drugs) and pathways (gene oe) represented in the dataset would at least provide a context in which to evaluate the results. Also, how are different concentrations and time points across the assays handled?

Re:

Thanks for the comments. We appreciate your detailed summary of our work and constructive feedback.

- First, we thank the reviewer for highlighting these aspects, and we agree there indeed exists perturbations that impact the transcriptional profile or morphological profile, but not both as noted in [28]. However, we would like to respectfully clarify that though morphological and transcriptional changes may not occur together in some cases, prediction between gene expression and cell morphology is feasible in most cases, benefiting from the existence of a shared subspace and sufficient shared information between two modalities (transcriptional profile and morphological profile) [8; 29; 30]. Specifically, the results shown in a previous work [8] demonstrate that the morphological profile is predictable from transcriptional profile in most cases (with R^2 score greater than 0.8 in some datasets). Meanwhile, the paper [28] also points out that “As well, to some degree, gene expression and morphology datasets contain sufficient information to predict changes in each other”. Therefore, prediction between gene expression and cell morphology is feasible in most cases. For clarity, we revised the Introduction Section in the revised manuscript to incorporate your comments and suggestions:

Although the relationship between cell morphology and gene expression is complex, there remains shared and complementary information between them [8], making the prediction between gene expression and cell morphology feasible [28; 8; 29; 30].

We also discussed this point in the Discussion Section to improve clarity:

Furthermore, though existing works [8; 28; 29; 30] and the analyses in this paper have demonstrated remarkable precision in the prediction between gene expression and cell morphology, this problem still remains challenging in some cases. Some perturbations only significantly affect either the transcriptome or cell morphology, while having only subtle and difficult-to-observe effects on the other [28]. This phenomenon may arise from the underlying mechanisms of the cell responses to perturbations and various technical confounders in the experimental setup. Such complexity increases the difficulty of accurate prediction and analysis between the gene expression and cell morphology, so addressing this challenge could be a meaningful future direction. Thus, incorporating more diverse input conditions, including but not limited to text description, drug structure, and chromatin accessibility in addition to gene expression, may further enhance the utility and generalizability of MorphDiff, making it a more powerful tool in predicting the cell morphology response under various perturbations.

- Second, following your concern about how general our model is, we performed additional evaluation with LINCS dataset from A549 cell line, which is different from the evaluation we have done related to U2OS cells. We conducted leave-one-out validation on the selected 10 MOAs and 10 targets. The selection

Response Figure 3: Leave-one-out validation performance on various MOAs and targets on the A549 cell line. The significance test used here is the Wilcoxon signed-rank test, corrected with the Bonferroni correction. **a.** Leave-one-out validation on different targets. **b.** Leave-one-out validation on MOAs.

criteria involve choosing mechanisms of action (MOA) and targets that include both gene expression and cell morphology datasets, along with a sufficient number of images. To ensure that the validation set in each fold is indeed out-of-distribution compared with the training set, we conducted the Kolmogorov-Smirnov test between the training set and the validation set, as shown in Response Table 5 and 6. The statistical results demonstrate that the validation set in each fold is significantly out-of-distribution compared to the training set. We added the details of constructing the leave-one-out evaluation set in the revised Method Section as follows:

To construct the dataset for validation, we first divided the data into 10 folds according to the 10 targets corresponding to 21 drugs. For each experiment, we trained the model on another 9 folds and validated it on the remaining fold. We did the same analysis for the 10 MOAs (42 drugs). The details of the drugs can be found in Supplementary files LINCS_moa_drug.xlsx and LINCS_target_drug.xlsx.

- The performance evaluations of leave-one-out validation are shown in Response Figure 3. The results show that MorphDiff(I2I) and MorphDiff(G2I) rank first and second on 4 out of 5 metrics, including cases in which they share the first place with other methods. Although MDTv2 performs well in terms of density, its shortcomings in other key metrics limit its overall performance. This suggests that MDTv2 may be more focused on generating similar samples, but it lacks generation diversity, which poses challenges for adaptability and reliability in application scenarios. Furthermore, we also conducted the same analysis as the revised Figure 3 on the LINCS datasets, which is displayed in Response Figure 4. MorphDiff consistently outperforms the baseline method. Response Figure 4a shows the R^2 score between the ground-truth CellProfiler feature vectors and the generated CellProfiler feature vectors. The x -axis represents the R^2 scores between the CellProfiler feature vector of IMPA and ground truth for each sample. The y -axis represents the R^2 scores of MorphDiff(G2I) and MorphDiff(I2I) with the ground truth respectively. The p -values obtained from the Wilcoxon signed-rank test demonstrate that MorphDiff generates cell morphologies significantly closer to the ground-truth compared to the baseline. 45% of the cell morphology generated from MorphDiff (I2I) have an R^2 value greater than 0.5, while 38.8% of MorphDiff(G2I) and 9.2% of IMPA exceed 0.5, highlighting that MorphDiff generates samples very similar to ground truth on the LINCS dataset. In Response Figure 4b, we trained an SVM binary classifier[31] for each pair of 10 selected perturbations with the ground-truth DeepProfiler embeddings and calculated the test classification accuracy for DeepProfiler embeddings generated by MorphDiff(I2I), MorphDiff(G2I) and IMPA. The x -axis represents the classification accuracy scores between pairwise perturbations of ground truth, and the y -axis means the classification accuracy scores of corresponding samples generated by different methods. The average accuracy scores for the ground truth, MorphDiff(I2I), MorphDiff(G2I), and IMPA are 97.3%, 55.9%, 59.9%, and 49.3%, respectively. The Wilcoxon signed-rank test results are as follows: MorphDiff(I2I) vs. IMPA: $p < 0.05$; MorphDiff(G2I) vs. IMPA: $p < 0.005$; MorphDiff(G2I) vs. MorphDiff(I2I): $p < 0.05$; MorphDiff(I2I) vs. Random: $p < 0.05$; MorphDiff(G2I) vs. Random: $p < 0.001$; IMPA vs. Random: $p = 0.81$ (no significant difference). Response Figure 4c assesses the predictions on the morphological changes between the control

Response Figure 4: Additional evaluation analysis on the LINCS dataset (For revised Figure 3 in the manuscript)

and perturbations for selected CellProfiler features in the LINCS leave-one-out set. The x -axis displays the changes between the ground truth perturbations and the control, while the y -axis displays the changes in the generated perturbations and the control. The R^2 score shows the degree of fit between ground truth and generated changes. The analysis further highlights the superiority of MorphDiff. Overall, MorphDiff is the most reliable tool on the LINCS dataset. We also added additional analysis in the revised manuscript as follows:

...To further demonstrate the generalizability of our model, we performed evaluation in a stricter setting. We conducted leave-one-out validation on the LINCS dataset (A549 cell line) for 10 different targets and 10 different MOAs. We selected targets and MOAs with sufficient cell morphology images for a robust evaluation. The detailed validation setting can be found in Section 4.3 (LINCS dataset). The results displayed in Appendix Figure 11 demonstrate that MorphDiff consistently achieves the most comprehensive performance across different targets and MOAs in the leave-one-out setting...

...We conducted the same analysis for CDRP OOD set (Appendix Figure 12a) and for LINCS leave-one-out set (Appendix Figure 14a), and found that both modes of MorphDiff consistently outperform the baseline methods with $p\text{-value} < 0.0001$, demonstrating the generalizability of our method...

...To further evaluate the generalizability of MorphDiff, we performed the same analysis on the CDRP OOD set and LINCS leave-one-out set in Appendix Figure 13 and Appendix Figure 14c respectively. Our conclusion is consistent across different settings and datasets...

Response Figure 5: Statistics of the pathways involved in the JUMP dataset

...We conducted the same analysis on 10 drugs selected from the LINCS leave-one-out set with the most ground-truth images. The results shown in Appendix Figure 14b further confirm the superiority of MorphDiff. The SMILES representations for these drugs can be found in Appendix Figure 15b. The results also indicate that MorphDiff(G2I) generates higher perturbation specificity than MorphDiff(I2I) at the drug level. This is because MorphDiff(I2I) additionally uses control images as input, which can be influenced by technical variations that may introduce substantial biases and further affect the detection of perturbation-specific change, especially when the phenotypic changes caused by the perturbation are subtle [17].

- Following your suggestion that the summary statistics on the MOAs/targets and pathways would at least provide a context to evaluate the results. We provided files with detailed information of MOAs/targets and pathways. For the CDRP drug dataset used in the revised Section 2.3, there are 68 drugs that act on 10 targets, annotated with 35 MOAs. These 68 drugs correspond to 35 distinct Mechanisms of Action (MOAs). In the LINCS leave-one-out dataset, there are 10 targets and 10 MOAs, covering a total number of 42 different drugs. During the exploration of MOAs of the U2OS L1000 profiles (we described the details in response to your Comment 2), we identified 8 types of drugs annotated with 6 MOAs and constructed a U2OS reference set that includes 251 drugs annotated with 60 MOAs. We have added this information in the revised Method Section to provide the context in which to evaluate the results. We added detailed information of all these drugs in the Supplementary files CDRP-BBBC047_959_drugs.xlsx, CDRP-BBBC047_target_moa_retrieval.xlsx, Explore_query_drug.xlsx, Explore_ref_drug.xlsx, LINCS_moa_drug.xlsx, LINCS_target_drug.xlsx. Regarding the pathway information, as each gene is involved in several different pathways, we counted the common pathways of the genetic perturbation target in the JUMP dataset as shown in Response Figure 5 and the detailed information can be found in the Supplementary file genetic_pathway_information.xlsx. We only counted those pathways that involve more than 10 genes in our JUMP genetic perturbation dataset.
- For the time-point information of the dataset, according to the data resource paper [8], the L1000 dataset and Cell Painting dataset are originally collected at different time points, while the collected time points are determined as optimal for each modality individually. Though the datasets are collected from different time points, both the previous work [8] and our framework have demonstrated the feasibility of predicting one modality using the other. Therefore, we did not integrate the time-point information into our framework. For the concentration information, due to the limited number of possible concentrations in the Cell Painting experiments and the heterogeneity of different datasets, we did not explicitly include the concentration information in the modeling framework. Instead, we matched the transcriptional and morphological profiles of the same perturbations with the same dosage to perform training and validation. The reasons for the design choice are discussed as follows. As shown in Response Figure 6, in the CDRP dataset, for most Cell Painting experiments, the concentrations of drug treatment are around 10. Therefore, it is infeasible to evaluate the generation results for different concentrations because of the limited availability of ground-truth experiments across different concentrations. Furthermore, for the newly collected LINCS dataset on the

Response Figure 6: Statistics on the concentrations with respect to the Cell Painting datasets and L1000 datasets for drug perturbations in CDRP-BBBC047-Bray (U2OS cell lines).

A549 cell line, both the L1000 datasets and the Cell Painting datasets have six concentrations (10, 3.3, 1.1, 0.33, 0.12, 0.04 μm). However, each concentration has only a limited number of experiments. Therefore, we did not encode the concentration information in modeling and evaluation to improve the generalizability of this framework, rather than being dataset-specific. Furthermore, we would like to state that, in the future, if there are new datasets with massive L1000 gene expression profiles and cell morphology datasets with high-quality matched time-point and concentration information, our framework can also be easily extended to support explicit encodings of these information. As our model is based on the latent diffusion framework [13] which supports flexible conditioning, we can also add time-point and concentration embeddings as input conditions along with the gene expression profile so that our model could explicitly encode this information as input. We added additional clarifications in the revised Discussion section:

Due to the data heterogeneity and lack of high-quality and massive datasets with matched time-point and concentration information, we do not explicitly encode these properties in our framework. In the future, if there are new datasets with massive L1000 gene expression profiles and cell morphology datasets with high-quality matched time-point and concentration information, our framework can also be easily extended to support explicit encoding of these information. As our model is based on the latent diffusion framework [13] which supports flexible conditioning, we can also add time-point and concentration embeddings as input condition along with the gene expression profile so that our model could explicitly encode this information as input.

2. *The authors themselves note in the Discussion that the model can only be applied when a transcriptomics profile of the perturbation is available. As also noted, there is a large publicly available pool of perturbation gene expression data available in the L1000 collection; however, corresponding morphological data is not as available. Thus, more explicit examples of potential uses for the proposed method would benefit the presentation.*

Re:

Thank you for your comment. Following your suggestion to evaluate the potential utility of our model in exploring the L1000 gene expression profiles without the corresponding cell morphology images, we have collected L1000 gene expression profiles without the corresponding perturbed cell morphology images for evaluation and validated the effectiveness of our method in utilizing these datasets for drug MOA matching. Concretely, we collected the L1000 data without the corresponding morphological images, from [8] (U2OS cell line). For quantitative evaluation, we conducted experiments to test whether our model could be applied to identify and match MOAs of drugs by generating cell morphology images from the L1000 gene expression profiles without the corresponding morphology images. We constructed a reference dataset of cell morphology treated by drugs with known MOAs, covering 251 drugs associated with 60 MOAs from CDRP, with the rationale that these MOAs cover a sufficient number of corresponding images for reference. Due to space limitations, detailed information on these drugs can

be found in the Supplementary file Explore_ref_drug.xlsx. For the L1000 gene expression profiles without cell morphology images, we only retained drugs with MOA annotations present in the reference dataset to evaluate the MOA matching performance, resulting in 293 L1000 gene expression profiles for 8 drugs annotated with 6 MOAs. Detailed information on these drugs can be found in Response Table 2. We generated 10 images for each L1000 gene expression profile to ensure the robustness of the evaluation. The evaluation metrics are folds of enrichment and mean average precision from [17; 22], which have been widely applied in evaluating MOA matching. The detailed descriptions of these two metrics are also discussed in the revised Method Section:

Drug Names	MOA
fluphenazine	dopamine receptor antagonist
thioridazine	dopamine receptor antagonist
BRD-K47448802	acetylcholine receptor agonist
BRD-K70693222	dyrk inhibitor
BRD-A54632525	lipoxygenase inhibitor
BRD-K06217810	dyrk inhibitor
BRD-K85853281	topoisomerase inhibitor
BRD-K48969316	caspase inhibitor

Response Table 2: Drugs and their corresponding MOAs of the explored L1000 gene expression profiles without corresponding morphological images.

Mean Average Precision Mean Average Precision (mAP) is an evaluation metric assessing the probability that samples of interest will rank highly on a list of samples rank ordered by some distance or similarity metric [22], such as cosine, Euclidean, Mahalanobis, etc. The mAP framework is agnostic to the choice of this distance metric. Given a sample as q (generated cell morphology image), annotated with l_q (drug, MOA, or genetic perturbation). Assume that there are n ground-truth cell morphology images. TP_k represents *True Positive* when ranking the top k ground-truth samples by distance with q , while TN_k , FP_k , and FN_k are defined similarly. Then the precision and recall at each rank can be calculated as $P_k = \frac{TP_k}{TP_k + FP_k}$ and $R_k = \frac{TP_k}{TP_n + FN_n}$. We can then calculate AP and mAP :

$$AP_q = \sum_k (R_k - R_{k-1}P_k) \quad (4)$$

$$mAP = \sum_{q=1}^n \frac{AP_q}{n} \quad (5)$$

Folds of Enrichment Folds of Enrichment is an evaluation metric assessing whether the perturbed cell morphology and ground-truth cell morphology belonging to the same MOA or genetic perturbation will have high similarity [17]. Concretely, for each generated cell morphology image, we

Response Figure 7: Folds of enrichment analysis on the morphological profiles, which are generated by the L1000 gene profiles without corresponding morphology images. The analysis are conducted on the U2OS cell line. top 1%, 5%, 10% refer to different thresholds.

calculated the odds ratio of a one-sided Fisher's exact test. The test is calculated using a 2×2 table. The first row contains the number of ground-truth cell morphology treated with the same treatment (drug, drug with the same MOA or genetic perturbation) (positive matches) and different treatments (negative matches) at a selected threshold of the list of results. The second row is the same, but for the treatments below the threshold (the rest). The odds ratio is the sum of the first row divided by the sum of the second row. It estimates the likelihood of observing the generated cell morphology are top similar to the ground-truth cell morphology with the same perturbation.

In Response Figure 7, with the above evaluation metrics, we can see that MorphDiff (both modes) outperforms the baseline and random cases, demonstrating its potential ability to infer the MOAs of the perturbed L1000 gene expression profile when the corresponding perturbation MOA labels are missing, by generating their morphology images. We added analyses of these experiments in the revised Section 2.4 as follows:

As the L1000 gene expression database is much larger than the current cell morphology database, we further explored the potential application of our model to generate the morphology images using the L1000 gene expression data without the corresponding ground-truth morphology images. We constructed a reference set of existing ground-truth morphology images covering 60 types of MOAs and 26112 images, which covers all drugs with available MOAs in the collected CDRP dataset and does not overlap with the CDRP Training set. Subsequently, we generated 2930 cell morphology images for 8 drugs whose MOAs are included in the reference set but the ground-truth cell morphology images are not yet profiled, allowing us to evaluate the exploration capabilities through MOA matching. Detailed information on these drugs can be found in Section 4.7. The results shown in Appendix Figure 20 indicate that MorphDiff, in both modes, achieves higher scores on two metrics in different threshold settings, demonstrating its potential to explore the massive perturbed gene expression space.

We provided a notebook for the L1000 exploration (L1000_exploration.ipynb) in code ocean for reference.

3. *The OOD datasets was constructed by a random split, could the authors clarify in what sense this constitutes out-of-distribution? Did the authors consider identifying sets of perturbations with well established mechanisms/targets and performing leave-one-out?*

Re:

We appreciate your constructive suggestions. In the original version, the out-of-distribution set indicates that the perturbations (drugs or genes) in the OOD set are unseen in the training set. We called it the OOD dataset because it covers perturbations out of the distribution of the training perturbations. Following your concern on whether the OOD set has a significant difference from the training set, we performed the Kolmogorov-Smirnov test to confirm that the OOD dataset has a different distribution from the training dataset. We used the DeepProfiler embeddings of the cell morphology images for testing. Concretely, the Kolmogorov-Smirnov test is a widely used non-parametric test for analyzing whether the two datasets come from the same distribution [32]. It compares the entire distributions of two datasets. If the p -value is smaller than a certain threshold (e.g. 0.05), it means the two datasets come from different distributions. As shown in Response Table 3 and Response Table 4, for the OOD datasets of JUMP and CDRP, they have different distributions from the training set with high statistical significance.

Furthermore, following your suggestion, we conducted leave-one-out experiments on 10 targets and 10 MOAs respectively within the LINCS datasets. We selected targets and MOAs with sufficient transcriptional profiles and cell morphology images for robust comparison and evaluation. We also performed the Kolmogorov-Smirnov test of the data distribution in the leave-one-out target and MOA setting in Response Table 5 and 6. We listed the MOAs, targets, and drugs selected for the analysis in Extended Response Table 1 and 2. We added detailed descriptions of the datasets and the Kolmogorov-Smirnov test in the revised manuscript (Section 4.3 Datasets) as follows:

JUMP dataset (U2OS cell line)...We conducted the Kolmogorov-Smirnov test between the JUMP Training set and the JUMP ID set with a value of p greater than 0.05, demonstrating no significant diversity between them. On the other hand, we performed the same test between the JUMP Training set and the JUMP OOD set with a p -value less than 0.0001, showcasing the significant distinction between the distributions of them...

CDRP dataset (U2OS cell line)...To assess the diversity of the out-of-distribution (OOD) set, we conducted Kolmogorov-Smirnov tests between the CDRP training set and the CDRP ID set, as well

as between the CDRP Training set and the CDRP OOD set. The resulting p -values were 0.4735 and less than 0.0001, respectively, indicating that the OOD set is sufficiently out of distribution...

...We consistently implemented the Kolmogorov-Smirnov test between the CDRP Training set and the CDRP Target_MOA set, a p -value equal to 0 illuminating the significant diversity of CDRP Target_MOA set...

LINCS dataset (A549 cell line) We collected the bulk gene profiles from [8] (LINCS part) and downloaded the corresponding images from [9]. In this dataset, we selected 61 drugs that have sufficient images, along with available MOA and target labels. The components of these images are the same as those in the U2OS dataset. We matched the gene expression and morphology images in the same way as CDRP. To construct dataset for model validation, we first divided the data into 10 folds according to the 10 targets corresponding to 21 drugs. We termed it as **LINCS Target leave-one-out set**. For each experiment, we trained the model on another 9 folds and validated it on the remaining fold. We performed the same operation for the 10 MOAs (42 drugs) and termed it the **LINCS MOA leave-one-out set**. We conducted the Kolmogorov-Smirnov test for each fold as Appendix Table 5 and 6 show, proving that the validation set in each fold is significantly out-of-distribution. The details of the drug can be found in the Supplementary files LINCS_moa_drug.xlsx and LINCS_target_drug.xlsx.

Set A	Set B	Statistic value	p -value
train set	ID set	0.0006	0.1848
train set	OOD set	0.0041	< 0.0001

Response Table 3: Kolmogorov-Smirnov test on the JUMP dataset. Set A and Set B indicate the two sample sets for performing the test.

Set A	Set B	Statistic value	p -value
train set	ID set	0.0002	0.4735
train set	OOD set	0.0032	< 0.0001
train set	Target_MOA set	0.0176	0.0

Response Table 4: Kolmogorov-Smirnov test on the CDRP dataset. Set A and Set B indicate the two sample sets for performing the test.

Set A	Set B	Statistic value	p -value
other targets	CTSK	0.0082	< 0.0001
other targets	ERBB2	0.0030	< 0.0001
other targets	F10	0.0070	< 0.0001
other targets	HRH1	0.0051	< 0.0001
other targets	MLNR	0.0022	< 0.0001
other targets	PREP	0.0052	< 0.0001
other targets	PTGS2	0.0010	0.0270
other targets	SLC6A1	0.0021	< 0.0001
other targets	TOP2A	0.0021	< 0.0001
other targets	TRPA1	0.0034	< 0.0001

Response Table 5: Kolmogorov-Smirnov test on the LINCS Target leave-one-out set. Set A and Set B indicate the two sample sets for performing the test.

Set A	Set B	Statistic value	p -value
other MOAs	EGFR inhibitor	0.0010	0.0028
other MOAs	HMGCR inhibitor	0.0023	< 0.0001
other MOAs	NFkB pathway inhibitor	0.0040	< 0.0001
other MOAs	adrenergic receptor agonist	0.0050	< 0.0001
other MOAs	adrenergic receptor antagonist	0.0020	< 0.0001
other MOAs	calcium channel blocker	0.0026	< 0.0001
other MOAs	dopamine receptor agonist	0.0100	< 0.0001
other MOAs	dopamine receptor antagonist	0.0010	< 0.0001
other MOAs	phosphodiesterase inhibitor	0.0018	< 0.0001
other MOAs	serotonin receptor antagonist	0.0034	< 0.0001

Response Table 6: Kolmogorov-Smirnov test on the LINCS MOA leave-one-out set. Set A and Set B indicate the two sample sets for performing the test.

The evaluation results for the leave-one-out settings are shown in Response Figure 3, where MorphDiff(I2I) and MorphDiff(G2I) rank first and second on 4 out of 5 metrics, including instances where they share the first place with other methods. The leave-one-out validation demonstrates that MorphDiff performs greatly across multiple metrics, generally reflecting a well-rounded model that effectively balances quality, diversity, and robustness, making it suitable for practical applications. We also added the benchmarking results for the leave-one-out setting in the revised manuscript as follows:

...We conducted leave-one-out validation on the LINCS dataset (A549 cell line) for 10 different targets and 10 different MOAs. We selected targets and MOAs with sufficient cell morphology images for a robust evaluation. The detailed validation setting can be found in Section 4.3 (LINCS dataset). The results displayed in Appendix Figure 11 demonstrate that MorphDiff consistently achieves the most comprehensive generative performance across different targets and MOAs in the leave-one-out setting...

4. In Fig 4a and Appendix Fig 9 and 10, MorphDiff appears to produce more distinct separation between DMSO control and treatments than ground truth, do the authors have any comments on this? Also, how do drugs with different targets relate to each other when projected together?

Re:

We appreciate your precious comments and are sorry for the confusion.

Response Figure 8: UMAP projection with different parameters on CellProfiler features, taking drugs targeting NR3C1 as the example.

- For the first question, We utilized UMAP to generate the original Figure 4a, Appendix Figure 9 and Figure 10. UMAP is famous for preserving the local topology while distorting the overall distance among all samples [33; 34]. Thus, the seemingly more distinct separation between DMSO control and drug perturbations generated by MorphDiff in the UMAP space is more likely to be attributed to the bias of UMAP in capturing the overall distance. We performed a thorough evaluation to validate this claim. Firstly, we used a quantitative metric, the Wasserstein Distance, to measure the distance between the perturbed DeepProfiler embeddings and the control (DMSO) DeepProfiler embeddings for ground truth, MorphDiff(I2I), MorphDiff(G2I) and IMPA in the Extended Response Figure 1 and Extended Response Figure 2. We can see that the Wasserstein Distance in MorphDiff(I2I) and MorphDiff(G2I) are quite close to the ground truth, and the distances between DMSO and target generated by our method are not higher than the ground truth. Secondly, we showed that the visual distance of the UMAP projection can be affected by the parameters in the UMAP function (we used the `scanpy.tl.umap` function integrated in Scanpy [35]). We found that two hyperparameters, *min-dist* and *spread*, could visibly affect the global distance in the UMAP embedding space. The *min-dist* parameter defines the effective minimum distance between the embedded points. Smaller values lead to a more clustered or compact embedding, bringing nearby points on the manifold closer together. In contrast, larger values result in a looser distribution of points. The *spread* parameter defines the effective scale of the embedded points. Together with *min-dist*, it determines the degree of clustering or dispersion of the embedded points. We set different numbers of these two parameters as Response Figure 8 shows, where the points in Response Figure 8a and b (*min-dist*=0.5) are more clustered than the points in Figure 8c and d (*min-dist*=1.5). On the other hand, the value of *spread* also affected the visualization. The points in Response Figure 8a (*spread*=1.5) are obviously more compact than the points in Response Figure 8b (*spread*=1.0), while there is no visually significant difference between Response Figure 8c (*spread*=1.5) and 8d (*spread*=1.0), along with larger values of *min-dist* (1.5). Therefore, the parameters of UMAP could affect the visualization quality, and some combination of parameters may amplify the subtle bias in UMAP projection. Overall, the results still demonstrate that our model generates a distribution that is closer to the ground truth than the baseline, and the “more distinct separation between DMSO and treatments than ground truth” is mainly caused by the parameters of UMAP. To demonstrate the effectiveness of our model in a quantitative way, we computed the distance between control and perturbations using an additional distance metric, Wasserstein Distance, as mentioned above. We updated the figures in Appendix Figure 16 (Extended Response Figure 1) and Appendix Figure 17 (Extended Response Figure 2) with proper UMAP hyperparameters. We added the following description in the revised Section 2.4:

Response Figure 9: UMAP projection on all targets for CellProfiler features and DeepProfiler embeddings.

...We observed that the ground-truth CellProfiler feature vectors induced by various drugs are clearly distinct from the ground-truth control set in the CellProfiler feature space. The CellProfiler feature vectors generated by the pre-trained MorphDiff (both modes) drive a consistent effect with the ground truth after UMAP projection. On the other hand, IMPA generates multiple clusters of perturbations targeting the same gene, while these perturbations should have a similar effect according to the ground truth, which indicates its poor generalization performance. Additionally, the Wasserstein Distance computed between the perturbed and control CellProfiler features quantitatively demonstrates the advantages of MorphDiff...

- For the second question, to explore how drugs with different targets relate to each other when projected together, we projected the corresponding CellProfiler features and DeepProfiler embeddings of all targets and DMSO control images to the UMAP embedding space as shown in Response Figure 9. In Response Figure 9a, we utilized the CellProfiler features, and we can see that the results generated by MorphDiff in both modes have a pattern similar to the ground truth, while the results from the baseline have a significantly different pattern. However, there is no significant visible distinction for drugs with different targets. In Response Figure 9b, we used DeepProfiler embeddings, and showed a separation of cell morphology treated with drugs with different targets from ground truth, MorphDiff(I2I) and MorphDiff(G2I), while the distribution of the generated cell morphology from IMPA is significantly biased from the ground truth. We noticed that DeepProfiler embeddings could better separate the cell morphology treated by drugs from different targets than CellProfiler embeddings. That is mainly because CellProfiler relies on traditional manual features and may have weaker representation capabilities than the deep learning-based feature extractor DeepProfiler [17]. We also noticed that the separation between cell morphology treated by drugs with different targets is not very clear even on the ground-truth cell morphology images. This may be attributed to the technical variation and batch effect in the dataset. The phenotypic differences caused by different perturbations are often subtle and may be overshadowed by technical variation. The technical variation includes several factors in the Cell Painting assay, including but not limited to the type of microscope used, the date and time of image acquisition, the technician involved, plate-to-plate variations (differences in assay preparation), and well-position effects (variations in humidity and temperature). The authors of DeepProfiler referred to these sources of variation as batch effects, and highlighted that addressing them remains an open research challenge for Cell Painting images [17]. As a result, although the ground-truth and the generated cell morphology images could to some extent capture the variation between the cell morphology treated by drugs with different targets, they could not yet fully separate the cell morphology in the UMAP space. As our model can already generate a highly similar distribution as ground truth, we expect the performance of MorphDiff in distinguishing drug targets to be continually improved with advancements in imaging platforms.

5. *As the improved accuracy in MOA retrieval of the proposed model over e.g. gene expression is a key result, it would seem prudent to present the results for different values of n in the n -nearest neighbour approach used to indicate the robustness of the results.*

Re:

Thanks for your comment. We agree that performing hyperparameter tuning on the retrieval method could further validate the robustness of our results. First, we tuned the number of k for the top k retrieval (indeed the same as the value of n in the n -nearest neighbor approach you mentioned). We varied the number of k from 5 to 10 and presented the results for the top 5 to 10 retrieval in Response Figure 10. We could observe that under different retrieval settings, both modes of MorphDiff consistently outperform baseline methods with statistical significance, which validates the robustness of our conclusion. We also revised the manuscript as follows:

To further validate the robustness of our methods, we also experimented with $k = 6, 7, 8, 9, 10$ as Appendix Figure 18 shows, and found that both modes of MorphDiff consistently outperform the baseline methods.

To further illustrate the robustness of our results in linking cell morphology with MOA, we also added additional evaluation on MOA matching using two additional evaluation metrics, the folds of enrichment and mean average precision. The descriptions of these two metrics are shown in the revised Method Section.

Response Figure 10: Top 5 to top 10 Mechanisms of Action (MOA) retrieval results for different modalities. Ground truth denotes retrieval with ground truth morphological embeddings, while MorphDiff(I2I), MorphDiff(G2I) and IMPA signify retrieval with respectively generated DeepProfiler embeddings. Gene Expression represents retrieval according to the MSE (Mean Square Error) between gene expression.

Mean Average Precision Mean Average Precision (mAP) is an evaluation metric assessing the probability that samples of interest will rank highly on a list of samples rank ordered by some distance or similarity metric [22], such as cosine, Euclidean, Mahalanobis, etc. The mAP framework is agnostic to the choice of this distance metric. Given a sample as q (generated cell morphology image), annotated with l_q (drug, MOA, or genetic perturbation). Assume that there are n ground-truth cell morphology images. TP_k represents *True Positive* when ranking the top k ground-truth samples by distance with q , while TN_k , FP_k , and FN_k are defined similarly. Then the precision and recall at each rank can be calculated as $P_k = \frac{TP_k}{TP_k + FP_k}$ and $R_k = \frac{TP_k}{TP_k + FN_k}$. We can then calculate AP and mAP :

$$AP_q = \sum_k (R_k - R_{k-1}P_k) \quad (6)$$

$$mAP = \sum_{q=1}^n \frac{AP_q}{n} \quad (7)$$

Folds of Enrichment Folds of Enrichment is an evaluation metric assessing whether the perturbed cell morphology and ground-truth cell morphology belonging to the same MoA or genetic perturbation will have high similarity [17]. Concretely, for each generated cell morphology image, we calculated the odds ratio of a one-sided Fisher's exact test. The test is calculated using a 2×2 table. The first row contains the number of ground-truth cell morphology treated with the same treatment (drug, drug with the same MOA or genetic perturbation) (positive matches) and different treatments (negative matches) at a selected threshold of the list of results. The second row is the same, but for the treatments below the threshold (the rest). The odds ratio is the sum of the first row divided by the sum of the second row. It estimates the likelihood of observing the generated cell morphology are top similar to the ground-truth cell morphology with the same perturbation.

The MOA matching experiment is conducted on the same dataset as MOA retrieval. As shown in Response Figure 11, under different thresholds, MorphDiff outperforms the baseline methods in the MOA matching task in terms of

Response Figure 11: Performance of generating feature representations MOA matching with two metrics: Mean Average Precision (mAP) in the x -axis and Folds of Enrichment in the y -axis. Each point indicates the mean of these metrics over all queries using the DeepProfiler embeddings. The threshold when computing the folds of enrichment here is set to top 10%, top 5%, top 2% and top 1% respectively.

mean average precision and folds of enrichment. Therefore, in different retrieval hyperparameters and evaluation settings, MorphDiff shows superior performance compared to baseline methods in identifying the MOAs of drugs. We added analysis in the revised Section 2.4:

In addition to the above evaluation for MOA retrieval, to further demonstrate the effectiveness of the methods, we also evaluated MOA matching. We used two metrics, folds of enrichment [17] and mean average precision [22]. The details of these two metrics are shown in Section 4.6. Intuitively, for each generated cell morphology, these metrics measure whether the top similar ground-truth cell morphology is treated by the drug belonging to the same MOA. As shown in Figure 4f, we found that both modes of MorphDiff consistently outperform the baseline methods when the threshold is set to the top 10%. We also provided the results for different thresholds in Appendix Figure 19 to validate the robustness of our method.

6. The example given with clocortolone pivalate is difficult to follow. Colocortolone pivalate is also glucocorticoid (albeit prodrug).

Re:

Thanks for your comment. The previous example with *clocortolone pivalate* may lead to confusion. In our revised manuscript, we have reorganized this section and added additional examples in Figure 4g (Response Figure 12a) and Appendix Figure 22 (Response Figure 12b). As shown in Response Figure 12a, we showed that *dimethylfraxetin* and *dichlorphenamide* both belong to *Carbonic anhydrase inhibitor*. However, the drug embeddings and gene expression of *dimethylfraxetin* and *dichlorphenamide* are of substantial distances. For example, *BML-259* from MOA *CDK inhibitor* has a more similar structure to *dichlorphenamide*, while *BRD-A72066420* with MOA *PPAR receptor antagonist* has a close distance with *dimethylfraxetin*. In contrast, in the cell morphology space characterized by the DeepProfiler embeddings, we observed that cell morphology treated with *dimethylfraxetin* and *dichlorphenamide* have similar feature distribution and can be clearly separated from cell morphology treated with drugs from other MOAs, indicating that cell morphology could provide complementary information in identifying the MOAs of drugs. We also observed that the cell morphology generated with both modes of MorphDiff also exhibits these patterns, with an overlapping distribution of cell morphology features treated with *dimethylfraxetin* and *dichlorphenamide*. The baseline method IMPA, as it exclusively relies on the drug structure to generate the perturbed cell morphology, only generates biased prediction where cell morphology treated with *dimethylfraxetin* and *dichlorphenamide* separated far away. An additional example is *bezafibrate*, *BRD-K54708045* and *ciglitazone*, which are all *PPAR receptor agonist*. In Response Figure 12b, we observed that their drug embeddings and gene expression are not in close proximity, but *BRD-K64402243* belonging to *Caspase inhibitor* is closer to *ciglitazone* from *PPAR receptor agonist*. However, in the cell morphology space, we can see that the DeepProfiler embeddings of these three drugs from the same MOA clustered together. Similar to the previous example, MorphDiff(I2I) and MorphDiff(G2I) both effectively capture this pattern, while IMPA generates biased cell morphology. With the observations mentioned above, we showed that cell morphology could provide complementary information in identifying the MOAs of drugs, and MorphDiff is potentially helpful in cases where researchers aim to discover an alternative drug structurally dissimilar to a known drug but of the same effect. We also revised the manuscript as follows:

As shown in Figure 4g, we showed that *dimethylfraxetin* and *dichlorphenamide* both belong to *Carbonic anhydrase inhibitor*, yet their structures are distinctly dissimilar. Meanwhile, the drug em-

Response Figure 12: The UMAP projection for drug embeddings, gene expression and DeepProfiler embeddings for treated cell morphology for drugs belonging to different MOAs. The DeepProfiler embeddings are extracted from ground truth images, generated images obtained from three methods respectively. a and b indicate two different examples.

beddings and gene expression of *dimethylfraxetin* and *dichlorphenamide* are of substantial distances. In the drug embedding space, *BML-259*, annotated with the MOA *CDK inhibitor*, and *RD-14620*, annotated with the MOA *EGFR inhibitor*, exhibit structural similarities to *dichlorphenamide*. Meanwhile, *BRD-A72066420*, with the MOA *PPAR receptor antagonist*, is closer to *dimethylfraxetin*. The structures of these drugs can be found in Figure 4g and Appendix Figure 21. In contrast, in the cell morphology space characterized by the DeepProfiler embeddings, we observed that cell morphology treated with *dimethylfraxetin* and *dichlorphenamide* have similar feature distributions and can be clearly separated from cell morphology treated with drugs from other MOAs, which indicates that cell morphology could provide complementary information in addition to drug structure and gene expression to identify the MOAs of drugs. We also observed that the cell morphology generated with both modes of MorphDiff also exhibited these patterns, with an overlapping distribution of cell morphology feature treated with *dimethylfraxetin* and *dichlorphenamide*. The baseline method IMPA, as it only relies on the drug structure to generate the perturbed cell morphology, only generates biased predictions with cell morphology treated with *dimethylfraxetin* and *dichlorphenamide* separated far

away. An additional example is *bezafibrate*, *BRD-K54708045* and *ciglitazone*, which are all *PPAR receptor agonists*. In Appendix Figure 22, we observed that their drug embeddings and gene expression are not closely distributed, but *BRD-K64402243* annotated with *Caspase inhibitor* is closer to *ciglitazone* annotated with *PPAR receptor agonist*. However, in the cell morphology space, we can see that the DeepProfiler embeddings of these three drugs from the same MOA clustered together. Similar to the previous example, MorphDiff(I2I) and MorphDiff(G2I) both successfully capture this pattern, while IMPA generates biased cell morphology.

Reviewer 3

I co-reviewed this manuscript with one of the reviewers who provided the listed reports. This is part of the Nature Communications initiative to facilitate training in peer review and to provide appropriate recognition for Early Career Researchers who co-review manuscripts. The code can be reproduced under the given instruction, including the model training code for MorphDiff and the code for reproducing the image results in the paper.

Re:

Thanks for your time in reviewing our manuscript and code. Thanks for your comments.

Reviewer 4

The authors present a model for predicting cellular morphology changes due to perturbations using a variational autoencoder (VAE) and diffusion model. They applied their model to two large scale datasets that explored morphological responses to drug and genetic perturbations and compared results against several baseline models. Evaluation consisted of comparisons of CellProfiler features as well as the ability to classify mechanisms of action (MOA). They found that the proposed model achieved the best image quality in terms of Frechet Inception Distance (FID).

The problem is well motivated and many extensive evaluations are performed to demonstrate the model's effectiveness. The details of the method are also presented clearly, so it is relatively easy to follow how the method works. My main criticism is that the manuscript includes so many analyses that it does not always fully explain the importance of the results. It can also be too highly dependent on the figures in the supplemental material, making it difficult to follow the paper without jumping back and forth between documents. I will include more specific examples below:

Re:

Thanks for your detailed and constructive comments and suggestions. We have carefully revised the manuscript and added necessary descriptions of the purposes and implications of these analyses to further highlight the importance of our results. We refined our analyses in each section by performing stricter statistical analysis and reorganizing figures. We also greatly appreciate your suggestion on making the manuscript less dependent on the figures in the supplementary materials. We have made sure that all important figures are present in the main manuscript for better readability. Some revisions we made to explain the importance of the results before explaining the details of the analyses are shown below, and the full revisions can be found in the revised manuscript.

...Then, we turned to a more fine-grained analysis of how well the generated morphology aligns with the ground truth morphology regarding the values of individual CellProfiler features...

...We also validated the effectiveness of MorphDiff in capturing the correlation between the perturbed gene expression and perturbed morphology...

...We further investigated the responses of cell morphology to small molecular compounds by validating whether the generated output aligns well with the ground truth on the CellProfiler features...

...Furthermore, we performed a fine-grained analysis on the effectiveness of our tool in predicting the features that undergo the most significant changes after perturbation...

...Moreover, it is crucial to evaluate whether the model can capture the diversity among different perturbations and generate perturbed cell morphology with specificity to perturbations.

...Moreover, we investigated whether our method could identify the significantly changed CellProfiler features after treated with drugs that target different genes...

...In addition to the above evaluation for MOA retrieval, to further demonstrate the effectiveness of the methods, we also evaluated MOA matching...

...Furthermore, drugs with similar structures usually have the same target and MOA(s) in most cases [36]. However, it would be more interesting to investigate drugs with the same MOA(s) but are structurally dissimilar [37; 38] to improve the biological safety and flexibility of drug development. Therefore, we explore whether other modalities, such as gene expression and cell morphology, can provide complementary information to identify these drugs...

1. Appendix figure 1 is important to appreciate the results and should be included in the main manuscript. No figures within the main manuscript include actual images of the inputs and outputs of the model, so this figure provides context for the statistical analyses and discussions.

Re:

Thanks for your suggestions. We agree that such visualization is essential for better readability. We have added Appendix Figure 1 to the main manuscript in Figure 1d. The revised Figure 1d is shown in Extended Response

Figure 3d. Due to the limitations of space, the visualization of some methods are placed in Appendix Figure 1 in the revised manuscript, and we also attached this figure as Extended Response Figure 4. For the previous table in Figure 1d. We have moved it to Appendix Table 1 for the utility comparison between methods. We also revised the manuscript as follows:

In Figure 1d and Appendix Figure 1, we visualized the generated output of MorphDiff and a series of baseline methods.

Response Figure 13: a. Performance benchmarking on the JUMP OOD set. b. Performance benchmarking on the Drug OOD set. MorphDiff(G2I) denotes generating the perturbation morphology images with only the perturbation gene expression. MorphDiff(I2I) denotes generating the perturbation morphology with the perturbation gene expression and a control reference image. Linear normalization was conducted on the reciprocals of FID, CMMD, and the other three metrics separately to convert their values to the range of 0 to 1. A larger value indicates better performance.

2. Figures 2a and 3a include several metrics, but none of these results are described in much detail. The authors note that their method is top two for all of them, but do not discuss the implications for each of them. For instance, DMIT is significantly higher in terms of two out of the four metrics (IS and coverage) in Figure 3a, so it is unclear that MorphDiff is the superior method in this analysis. The authors do note that FID is the most important feature, but then do not discuss the fact that MorphDiff(g) significantly outperforms MorphDiff(g+i) in Figure 2a.

Re:

Thanks for pointing that out. We have carefully modified the benchmarking performance analysis part following the comments from you and other reviewers. Following the suggestions of another reviewer, in the general generative performance evaluation parts, we added a new metric CMMD [3], and added two generative methods as comparative methods, which are VQGAN [1] and Masked Diffusion Transformers [2]. Furthermore, following your suggestion in Comment 5, we removed the average score in the comparison. The updated figures are shown in the Response Figure 13. Following your comments 6 and 8, we performed statistical analysis with the Wilcoxon test and Bonferroni correction. After that, DMIT only significantly outperforms MorphDiff(I2I) on coverage score, and shows no significant difference with MorphDiff(I2I) on IS (Response Figure 13b and revised Figure 3a). Following your suggestion, we revised the analysis of the performance evaluation metric and added the following sentence in the first paragraph of Section 2.2 of the revised manuscript:

...We evaluated these methods using several metrics, including FID (Fréchet Inception Distance) [39], IS (Inception Score) [10], CMMD [3], as well as density and coverage [12]. FID measures the distance between the distributions of the ground-truth and generated images in the feature space, using features extracted from Inception V3 [10]. The IS assesses the quality of the generated images based on how effectively they can be classified by the Inception V3 model. Both FID and IS utilize Inception V3 [10], which is pre-trained on ImageNet [11]. To enhance the reliability and robustness of our comparisons, we incorporated CMMD, which calculates the Maximum Mean Discrepancy (MMD) distance based on features extracted by a CLIP model pre-trained on 400 million image-text pairs containing more complex scenes [3]. We also included density and coverage metrics, as these metrics are robust against outlier samples and could better measure the diversity and fidelity of the generated samples [12]. In general, lower FID and CMMD values indicate better performance,

while higher density, IS, and coverage scores reflect better generation quality. For convenience of presentation, we used the reciprocals of FID and CMMD, denoted as FID(-1) and CMMD(-1), for comparison and analysis...

As depicted in Figure 2a, we can see that MorphDiff (I2I) and MorphDiff (G2I) rank first and second in all metrics except density, while DMIT ranks first in density. For DMIT, the higher density suggests that the generated samples are relatively concentrated in the feature space, indicating that they exhibit high similarity to specific areas of the ground-truth distribution. However, the relatively lower scores of DMIT in FID(-1), IS, and CMMD(-1) indicate that samples generated by DMIT are of worse quality and fidelity. For the two modes of MorphDiff, we can see that MorphDiff(I2I) performs better on the metrics of density and coverage, suggesting that it generates a more diverse set of samples, and the generated distribution better aligns the ground truth. MorphDiff (G2I) scores higher in FID(-1) and CMMD(-1), indicating higher quality and fidelity. This highlights a trade-off where MorphDiff(G2I) prioritizes quality over diversity, while MorphDiff(I2I) emphasizes the diversity of generated outputs. In general, based on these five metrics, MorphDiff demonstrates more balanced performance in terms of generalization, fidelity, and diversity compared to other generative models, resulting in the best generation quality.

We also added descriptions at the first paragraph in the Section 2.3 of the revised manuscript:

As Figure 3a shows, MorphDiff(I2I) ranks first on FID(-1) and CMMD(-1), and outperforms other methods significantly. The results show that MorphNet achieves a remarkably high density score but a nearly zero coverage score, showing an imbalance between the quality and diversity of the generated samples. After checking the images generated by MorphNet, we found that it produced almost identical images, regardless of the input condition, resulting in high density but low coverage, which is a typical example of mode collapse. On the other hand, DMIT ranks the best on coverage but obtains relatively lower scores on FID(-1) and CMMD(-1) compared to MorphDiff (both modes), suggesting that it struggles to produce high-quality samples. For the two modes of MorphDiff, MorphDiff(G2I) outperforms on density and coverage, while MorphDiff(I2I) performs better on FID(-1), IS and CMMD(-1). In this case, MorphDiff(G2I) excels in generating samples of higher diversity and closer to the ground-truth distribution, while MorphDiff(I2I) is more effective in generating high-quality samples. Ideally, an effective generative model should achieve low FID, CMMD, and high scores in IS, density, and coverage to ensure that it produces outputs of high quality and fidelity, while also being diverse and closely aligned with the ground-truth distribution. MorphDiff (both modes) outperforms baseline methods considering these criteria, demonstrating its comprehensive generative capability and stability.

We also added analyses between two modes of MorphDiff in the revised Discussion Section as follows:

MorphDiff can be applied in two modes: I2I and G2I. The G2I mode uses gene expression solely as input, while the I2I mode incorporates both gene expression and control cell morphology. Though both modes outperform the baseline methods, we observed a discrepancy in quantitative performance between the two modes. For example, on the JUMP dataset, the I2I mode generates samples with greater diversity, while the G2I mode produces samples of higher quality. Conversely, on the CDRP dataset, the opposite is observed. On the LINCS dataset, the I2I mode performs better than the G2I mode except for the IS score. One reason for this discrepancy may be the quality of the control morphology images used as input in the I2I mode. Although the control cell morphology provides valuable information about cellular components, it can also be affected by batch effects and technical variations in Cell Painting experiments. Addressing these technical factors remains an open problem for Cell Painting images [17]. Nevertheless, both modes of MorphDiff meet the criteria for an effective generative model, achieving low FID, CMMD and high scores in IS, density, and coverage. In most application scenarios, either mode is suitable, except when users specifically aim to analyze morphological feature changes with respect to certain control cells, as discussed in Figure 3c and Figure 4b; in this case, MorphDiff (I2I) is the more suitable option.

3. *There are two versions of MorphDiff described in the text (gene+control and gene only). The terminology for these is inconsistent (e.g., G2I and I2I in the figure 1 caption vs. g+I and g in figures 2 and 3 vs. Gene2Img in the body). Throughout most of the manuscript, reference to MorphDiff does not specify a version, so it is unclear*

which is being used. Presumably, it is $g+i$ but that isn't made clear and it also isn't clear why only that version would be included in these analyses.

Re:

Thanks for your comment. We have unified the terminology. In the original paper, MorphDiff(G2I) is equivalent to G2I/MorphDiff(g) and MorphDiff(I2I) is equivalent to I2I/MorphDiff(g+i). We have unified the names as MorphDiff(G2I) and MorphDiff(I2I) in all figures and text for clarity. Basically, MorphDiff(G2I) refers to the version that generates the cell morphology images solely conditioned on the gene expression, while MorphDiff(I2I) refers to the version that generates the cell morphology images based on a control cell morphology image and conditioned on the gene expression. We added a clear definition in the revised Section 2.1:

In the first mode (referred as MorphDiff(G2I)), MorphDiff(G2I) takes the L1000 gene expression as the condition and denoises the corresponding cell morphology images from random noise distribution. In the second mode (referred as MorphDiff(I2I)), MorphDiff(I2I) takes the L1000 gene expression for one specific perturbation as the condition and transforms the morphology images from the control set to the predicted morphology images for that perturbation.

In our revised version, we performed experiments and analysis for both MorphDiff(G2I) and MorphDiff(I2I) throughout the manuscript. We have clearly indicated the version we used for each analysis in the revised version.

4. It is a bit unclear how the five-channel images are handled in some of the analyses. In supplemental figure 1, outputs are presented in RGB, so it is unclear how this mapping was performed. Also, the authors note that only their model and IMPA can produce five-channel outputs, so it is also unclear how the other baseline models can be applied to five channel input or how their outputs can be compared to the five-channel ground truth in figures 2a and 3a.

Re:

Thanks for your comments. The raw data of the cell morphology images contain five channels. During the training, for the methods that could process five-channel input including MorphDiff and IMPA, we directly use the five-channel cell morphology images to train the model; for the methods that could not process five-channel input including the rest of the methods, we set a weight for each channel, and merge the five-channel images to three-channel RGB images. For example, for channel i , we assume the values of this channel as C_i and the weight of this channel as $w_i = (w_{ir}, w_{ig}, w_{ib})$. The three channel RGB image $C = \{C_r, C_g, C_b\}$ can be computed as $C_r = \sum_{i=1}^5 w_{ir} C_i$, $C_g = \sum_{i=1}^5 w_{ig} C_i$ and $C_b = \sum_{i=1}^5 w_{ib} C_i$. For visualization, since three-channel RGB images are the most intuitive, we converted the output of MorphDiff and IMPA to three-channel RGB images using weights w_i . For performance benchmarking, as many scores can only be applied to three-channel RGB images such as the Inception score, we also converted the output of MorphDiff and IMPA to three-channel images when benchmarking. The weights for different channels are DNA(1,0,0), ER(0,1,0), RNA(0,0,1), AGP(0,0.4,0.4) and Mito(0.4,0.4,0.2). These weights are unified for different methods. For clarity, we also added more descriptions in the Method Section of our revised manuscript as follows:

Evaluation setup across baseline methods Among all the baseline methods evaluated in manuscript, only MorphDiff and IMPA support training and generation with the original five-channel input images, while the other baseline methods only support three-channel RGB images as input and output. Therefore, to fairly evaluate across different methods, we devised the following evaluation strategy following previous works [40]. During the training, for the methods that could process five-channel input including MorphDiff and IMPA, we directly use the five-channel cell morphology images to train the model; for the methods that could not process five-channel input including the rest of the methods, we set a weight for each channel, and merge the five-channel images to three-channel RGB images. For example, for channel i , we assume the values of this channel as C_i and the weight of this channel as $w_i = (w_{ir}, w_{ig}, w_{ib})$. The three-channel RGB image $C = \{C_r, C_g, C_b\}$ can be computed as $C_r = \sum_{i=1}^5 w_{ir} C_i$, $C_g = \sum_{i=1}^5 w_{ig} C_i$ and $C_b = \sum_{i=1}^5 w_{ib} C_i$. For visualization, since three-channel RGB images are the most intuitive, we converted the output of MorphDiff and IMPA to three-channel RGB images using weights w_i . For performance benchmarking, as many metrics can only be applied to three-channel RGB images such as the Inception Score and FID, we also converted the output of MorphDiff and IMPA to three-channel images when benchmarking. The weights for different channels are DNA(1,0,0), ER(0,1,0), RNA(0,0,1), AGP(0,0.4,0.4) and Mito(0.4,0.4,0.2). These weights are unified across different methods.

We added a simple demo of compositing five-channel images into three-channel RGB images in code ocean (composite_523.ipynb) for inference.

While the trends in the results seem compelling, I have some concerns about how statistical testing is performed and evaluated:

5. Averaging the metrics in figures 2a and 3a is not meaningful. The authors note that FID “is the most prevalent and significant metric,” so it isn’t appropriate to create an aggregate score (i.e., the average) that treats each identically.

Re:

Thanks for your suggestions. We agree with your comment that the aggregate score is not appropriate. We have removed the aggregate score for the results and analyzed these metrics separately. Meanwhile, following the suggestions from Reviewer 1 and you, we added another performance evaluation metric CMMD [3] and analyzed the results. The revised results are shown in Response Figure 13. We have also added the revised manuscript paragraph in response to your Comment 2.

6. The performance of statistical hypothesis testing is appreciated, but there are many different comparisons in each of these analyses. It is unclear whether correction was performed (e.g., Bonferroni).

Re:

Thanks for your suggestion. Previously we did not perform correction. In the revised manuscript, we have added Bonferroni correction to the statistical tests involved in the performance comparison in the revised Figure 2a, Figure 3a, Figure 4a, Figure 4b, and Figure 4e and modified the manuscript accordingly. We also showed the corresponding updated Figure 2a and Figure 3a in Response Figure 13. We explicitly stated in the manuscript when the Bonferroni correction is applied:

...Each method was sampled 10 times using different seeds, and a Wilcoxon signed-rank test was performed to assess the significance of the results, with p-values corrected using Bonferroni correction...

7. The authors state that they use a t test to “have a more holistic view of how close the MorphDiff-generated morphology aligns with the ground truth distribution.” A t-test will only test whether there are significant differences between the centers of distributions. To investigate the similarity of distributions, it would be more appropriate to use a Kolmogorov Smirnov test or use a metric such as KL divergence.

Re:

Thanks for your comments. In our previous manuscript, we used the t test to test the distribution similarity between the ground-truth CellProfiler feature and the generated CellProfiler feature. We agree that the t-test could be biased when the feature distribution is not normal and only considers the centers of distributions. Therefore, following your suggestions, we used three additional evaluations, including the Wilcoxon test, the Kolmogorov-Smirnov test and the KL divergence metric to validate the consistency of our conclusion. The results are shown in Response Figure 14. We included both MorphDiff(G2I) and MorphDiff(I2I) in this comparison. As we can see in Response Figure 14, MorphDiff(G2I) and MorphDiff(I2I) both outperform the baseline method IMPA regardless of the types of tests or metrics used, highlighting the effectiveness of our method in capturing the ground-truth CellProfiler feature distribution. We also modified in the revised manuscript as follows:

...We conducted an independent Wilcoxon signed-rank test across the ground-truth perturbation CellProfiler features and the generated CellProfiler features...

...We also experimented with other types of statistical tests and metrics, including t-test, Kolmogorov Smirnov test and KL-divergence, in Appendix Figure 6 for MorphDiff(G2I), MorphDiff(I2I) and IMPA to further demonstrate the superiority of our method.

8. The authors report using t-tests, but the distributions they are comparing do not appear to be normal. It is probably more appropriate to use nonparametric tests (e.g., Wilcoxon).

Re:

Thanks for your comment. We agree that the distributions that we are comparing may not follow the Gaussian distribution. Therefore, in the revised Figure 2a, revised Figure 2d, revised Figure 3a, revised Figure 4a, 4b and 4e, we have changed the t-test to the Wilcoxon test for more reasonable comparisons. Our results still demonstrate statistical significance after changing to Wilcoxon tests:

Response Figure 14: a. Using additional tests for Figure 2d in the manuscript. t-test, KL-divergence metric (with a threshold of 3, i.e., KL divergence lower than 3 means the distribution is similar), Wilcoxon and Kolmogorov Smirnov test. The results are compared between MorphDiff(G2I) and IMPA. b. Similar to a, but the results are compared between MorphDiff(I2I) and IMPA.

...we conducted Wilcoxon signed-rank test across the CellProfiler features of the ground truth and the generated output...

9. The main statistical claim that the authors make is that their model “outperforms the baseline method and gene expression-based retrieval by 29.1% and 9.7% respectively” (mentioned three times and the only numerical result in the abstract). While this statement is technically accurate, it is a bit deceiving because the metric the authors are comparing is a percentage (accuracy). Stating a percentage increase is therefore often interpreted as an accuracy that is 29.1 and 9.7 percentage points higher, which is not the case. It would be better to report the actual accuracy values from the different models and simply state that the improvement is statistically significant.

Re:

Thanks for your comment. Following your comment, we have modified this statement. The average accuracy of MorphDiff(G2I) and MorphDiff(I2I) is 45.86% and 46.95% respectively. The average accuracy of IMPA and gene expression-based retrieval is 30.0% and 38.91%. Thus, we rewrite this statement as follows: outperforms the baseline method and gene expression-based retrieval by 16.9% and 8.0% respectively. We also made the following modifications to the manuscript:

The average MOA retrieval accuracy of MorphDiff-generated morphology is comparable with that of the ground truth cell morphology, and consistently outperforms the baseline method and gene expression-based retrieval by 16.9% and 8.0% respectively. The improvement is statistically significant.

Response Figure 15: **a.** The R^2 score between the ground-truth CellProfiler feature vectors and the generated CellProfiler feature vectors on the Drug ID set. **b.** The R^2 score between the ground-truth CellProfiler feature vectors and the generated CellProfiler feature vectors on the Drug OOD set. **c.** The R^2 score between the ground-truth CellProfiler feature vectors and the generated CellProfiler feature vectors on the LINCS leave-one-out set.

10. Some of the descriptions of trends in the data uses vague or imprecise wording. For instance “Figure 3b shows that most scatter points lie above the diagonal, signifying that MorphDiff outperforms IMPA in most of the 3000 samples. Meanwhile, the y-coordinate corresponding to the highest density is close to one.” Use precise statistics rather than vague and subjective words like “most” or “close.”

Re:

Thank you for pointing that problem out. We replaced the vague sentence with precise statistics as the Response Figure 15 shows. We conducted the Wilcoxon signed-rank test on the pairs of R^2 scores, demonstrating that MorphDiff (both modes) outperforms the baseline. We changed the vague sentence in the revised Section 2.3 as:

MorphDiff(I2I) and MorphDiff(G2I) outperforms the baseline significantly with a p -value less than 0.0001 through Wilcoxon signed-rank test. In more detail, 89% of the generated samples from MorphDiff(I2I) exhibit an R^2 score greater than 0.5, with 27.2% surpass 0.8. For MorphDiff(G2I), 87.6% of the samples have an R^2 score greater than 0.5, and 16.2% exceed 0.8. In contrast, 78.3% of the samples from IMPA exceed 0.5, but none reach 0.8. This indicates that MorphDiff (both modes) can well capture the feature distribution of the drug perturbation dataset. We conducted the same analysis for CDRP OOD set (Appendix Figure 12a) and for LINCS leave-one-out set (Appendix Figure 4a), and found that both modes of MorphDiff consistently outperform the baseline methods with the value $p < 0.0001$, demonstrating the generalizability of our method.

The manuscript also requires significant copy-editing. There are grammatical issues throughout, including a typo in the title (“celluar”). I’ll list several specific issues below with examples, but this list is not exhaustive:

11. Acronyms used before they are defined (e.g., “fid” on page 4), defined multiple times (e.g., “MOA” is defined twice in the abstract), or not defined at all (e.g., “SMILES”).

Thanks for your comments. We have clarified the acronyms the first time they appear and deleted duplicate definitions. Some full definitions are listed below:

- IMPA: IMage Perturbation Autoencoder
- DRIT++: Disentangled Representation for Image-to-Image Translation
- VQGAN: Vector Quantised Generative Adversarial Network
- MDTv2: Masked Diffusion Transformer
- DMIT: Disentanglement for Multi-mapping Image-to-Image Translation
- FID: Frechet Inception Distance
- IS: Inception Score
- SMILES: Simplified Molecular Input Line Entry System

We also added a table including all acronyms and full names in our paper as Response Table 1 shows.

12. *Colloquial or vague language (e.g., "can be pretty noisy").*

Re:

Thanks for your comments. We really appreciate your suggestions in our writing. We have modified these words accordingly. Regarding the specific example you mentioned, we have modified as follows:

...which leads to a high noise level in the cell morphology data.

13. *Incorrectly used words (e.g., "our framework corporates" should be "incorporates"; "in the following article" should probably be "section" or "analysis").*

Re:

Thanks for your comments. We have carefully modified these inappropriate words following your comments.

...our framework incorporates CellProfiler...

14. *Plurality mismatch (e.g., "training and ID set covers" should be "training and ID sets cover").*

Re:

Thanks for your comments. We have carefully modified these grammar errors following your comments.

15. *Missing articles (e.g., "architecture augmented with attention mechanism" should say "...with an attention mechanism"; "We selected most predictable morphological features" should say "...the most predictable...").*

Re:

Thanks for your comments. We have carefully modified these grammar errors following your comments

16. *Awkward phrasing (e.g., "achieves an excellent performance").*

Re:

Thanks for your comments. We have carefully modified these words and tried to make our description clearer. Regarding your specific example, we have modified the manuscript accordingly.

With the designed MOA retrieval pipeline, we found that the generated cell morphology from MorphDiff has comparable performance with the ground truth cell morphology, and outperforms the baseline methods statistically significantly.

17. *Editorializing (e.g., "which leads to a meager coverage score").*

Re:

Thanks for your comments. We have carefully modified the manuscript accordingly.

Other comments:

18. *In figure 4c, in one of the differences marked as significant (CDK2) IMPA is higher, but it is marked with an asterisk indicating MorphDiff is higher.*

Re:

Thanks for your comment. We have corrected this mistake in the revised manuscript (now Figure 4b). We also revised the manuscript accordingly.

Figure 4b demonstrates that MorphDiff(I2I) effectively identifies significantly changed CellProfiler features, outperforming IMPA for 6 out of 10 targets.

19. The authors mention that they "expand each channel to 3 dimensions to match the input shape of VGG," but it is unclear how this was achieved. Were channels duplicated?

Re:

Thank you for highlighting that confusion. Yes, the channels were indeed duplicated. Intuitively, we duplicated the images from each channel (128,128) to create the images with a shape of (3,128,128) to match the input shape specified by VGG. We have revised the Method Section as follows:

We duplicate each channel from a shape of (128, 128) to (3, 128, 128) to ensure that the images from each channel are compatible with the input shape required by VGG.

Minor comments:

20. In appendix figure 1, including the control image would provide helpful context for the results.

Re:

Thanks for your comment. We have included the control image (DMSO) in the revised Figure 1d and Appendix Figure 1. We also attached these figures in Extended Response Figure 3 and Extended Response Figure 4.

21. Figure 1d should be a separate table rather than a subfigure.

Re:

Thanks for your comment. We have converted Figure 1d to Appendix Table 1 to compare the utility between different methods. We also showed the table in Response Table 7.

Model	G2I	I2I	CellProfiler	DeepProfiler
MorphDiff	Y	Y	Y	Y
StarGANv1	N	Y	N	N
IMPA	N	Y	Y	Y
DMIT	N	Y	N	N
DRIT++	N	Y	N	N
MDTv2	N	Y	N	N
VQGAN	N	Y	N	N
MorphNet	Y	N	N	N

Response Table 7: Comparison of MorphDiff with other related tools. G2I means whether this tool supports perturbed gene expression/gene embeddings/drug embeddings to perturbed cell morphology generation. I2I means whether this tool supports control cell morphology to perturbed cell morphology generation. CellProfiler means whether this tool could provide an interpretable feature set for the downstream analysis involving CellProfiler. DeepProfiler means whether this tool could provide an interpretable embedding set for the downstream analysis involving DeepProfiler.

22. It is unclear why q_i needs to be defined since it is identically p_i .

Re:

Thanks for your comments. p_i and q_i represent different directions of the diffusion process, q_i describes the probabilistic distribution of z_t given the probabilistic distribution of z_{t-1} , while p_i describes the probabilistic distribution of z_{t-1} given the probabilistic distribution of z_t . Therefore, we need to define p_i and q_i , respectively. To further enhance clarity, we also revised the following descriptions in the manuscript:

...Concretely, the denoising process can be formulated as follows, where $p_{\theta}(z_{t-1}|z_t)$ denotes the probability distribution of z_{t-1} given z_t ...

...where $q(z_t|z_{t-1})$ can be interpreted as obtaining z_t by adding Gaussian noise to z_{t-1} parameterized by $1 - \alpha_t$...

23. The better performing model includes Y_{DMSO} , so it probably makes sense to have that version of the function in equation 1.

Re:

Thanks for your suggestion. We have included it in the revised version as follows:

The cell morphology of the control group may serve as prior information for modeling the cell morphology after perturbation. Thus, f can also take Y_{DMSO} as input, which can be written as $f(l_{q_i}, Y_{\text{DMSO}})$.

24. "time t " should instead say "step t ."

Re:

Thanks for your suggestion, we have made the modifications in the revised manuscript.

z_t is a noisy version of z_0 at step t .

25. It can be a little hard to read when describing statistical tests together with the results (e.g., second paragraph of 2.4). It would be better to describe the tests with the metrics in the Method Section and only mention significance in the results.

Re:

Thanks for your suggestion, we have carefully revised the second paragraph of Section 2.4 along with other paragraphs to improve the readability. We revised the second paragraph of 2.4 as follows:

Moreover, we investigated whether our method could identify the significantly changed CellProfiler features after being treated with drugs that target different genes. First, we adopted the chi-square test between the ground-truth control and the perturbed morphology to identify significantly changed CellProfiler features with the p -value threshold at 0.05. Then, we calculated the F1 score (details in Section 4.6) between the significantly changed CellProfiler features identified with the generated and ground-truth morphology. We also sampled 10 times for each model to determine the significance of the results. Figure 4b demonstrates that MorphDiff(I2I) effectively identifies significantly changed CellProfiler features, outperforming IMPA for 6 of 10 targets. Among these, five targets have corrected p -values less than 0.005, and one target has a corrected p -value less than 0.01, as determined by the Wilcoxon signed-rank test with Bonferroni correction. MorphDiff(I2I) also outperforms MorphDiff(G2I) for 5 targets, while MorphDiff(G2I) outperforms MorphDiff(I2I) for 2 targets significantly.

We describe the tests with the metrics in the Method Section:

F1 score In this study, we adopt balanced F-score (F1 score) as the evaluation metric in Section 2.4. After the chi-square test on CellProfiler features extracted from ground truth morphology, we term the features with p -values less than 0.05 as positive samples (regarded as significantly changed features), and term other features as negative samples. For generated morphology, we conduct the same test on generated CellProfiler features, then we acquire predicted positive samples and negative samples generated by each method. In this way, we can calculate true positive (TP), false positive (FP) and false negative (FN) for each method. The F1 score can be calculated as:

$$F_1 = \frac{2TP}{2TP + FP + FN} \quad (8)$$

References

- [1] Esser, P., Rombach, R. & Ommer, B. Taming transformers for high-resolution image synthesis. In Proceedings of the IEEE/CVF conference on computer vision and pattern recognition, 12873–12883 (2021).
- [2] Gao, S., Zhou, P., Cheng, M.-M. & Yan, S. Masked diffusion transformer is a strong image synthesizer. In Proceedings of the IEEE/CVF International Conference on Computer Vision, 23164–23173 (2023).
- [3] Jayasumana, S. et al. Rethinking fid: Towards a better evaluation metric for image generation. In Proceedings of the IEEE/CVF Conference on Computer Vision and Pattern Recognition, 9307–9315 (2024).
- [4] Ramesh, A., Dhariwal, P., Nichol, A., Chu, C. & Chen, M. Hierarchical text-conditional image generation with clip latents. arXiv preprint arXiv:2204.06125 **1**, 3 (2022).
- [5] Subramanian, A. et al. A next generation connectivity map: L1000 platform and the first 1,000,000 profiles. Cell **171**, 1437–1452 (2017).
- [6] Chandrasekaran, S. N. et al. Jump cell painting dataset: morphological impact of 136,000 chemical and genetic perturbations. bioRxiv 2023–03 (2023).
- [7] Bray, M.-A. et al. A dataset of images and morphological profiles of 30 000 small-molecule treatments using the cell painting assay. Gigascience **6**, giw014 (2017).
- [8] Haghighi, M., Caicedo, J. C., Cimini, B. A., Carpenter, A. E. & Singh, S. High-dimensional gene expression and morphology profiles of cells across 28,000 genetic and chemical perturbations. Nature methods **19**, 1550–1557 (2022).
- [9] Natoli, T. et al. broadinstitute/lincs-cell-painting: Full release of lincs cell painting dataset (2021).
- [10] Salimans, T. et al. Improved techniques for training gans. Advances in neural information processing systems **29** (2016).
- [11] Deng, J. et al. Imagenet: A large-scale hierarchical image database. In 2009 IEEE conference on computer vision and pattern recognition, 248–255 (Ieee, 2009).
- [12] Naeem, M. F., Oh, S. J., Uh, Y., Choi, Y. & Yoo, J. Reliable fidelity and diversity metrics for generative models. In International Conference on Machine Learning, 7176–7185 (PMLR, 2020).
- [13] Rombach, R., Blattmann, A., Lorenz, D., Esser, P. & Ommer, B. High-resolution image synthesis with latent diffusion models. In Proceedings of the IEEE/CVF conference on computer vision and pattern recognition, 10684–10695 (2022).
- [14] Vaswani, A. et al. Attention is all you need. Advances in neural information processing systems **30** (2017).
- [15] Lee, H. & Welch, J. D. Morphnet predicts cell morphology from single-cell gene expression. bioRxiv 2022–10 (2022).
- [16] Choi, Y. et al. Stargan: Unified generative adversarial networks for multi-domain image-to-image translation. In Proceedings of the IEEE conference on computer vision and pattern recognition, 8789–8797 (2018).
- [17] Moshkov, N. et al. Learning representations for image-based profiling of perturbations. Nature communications **15**, 1594 (2024).
- [18] Carpenter, A. E. et al. Cellprofiler: image analysis software for identifying and quantifying cell phenotypes. Genome biology **7**, 1–11 (2006).
- [19] Seal, S. et al. From pixels to phenotypes: Integrating image-based profiling with cell health data as biomorph features improves interpretability. Molecular Biology of the Cell **35**, mr2 (2024).
- [20] Serrano, E. et al. Reproducible image-based profiling with pycytominer. ArXiv arXiv–2311 (2024).

- [21] Sanchez-Fernandez, A., Rumetshofer, E., Hochreiter, S. & Klambauer, G. Cloome: contrastive learning unlocks bioimaging databases for queries with chemical structures. *Nature Communications* **14**, 7339 (2023).
- [22] Kalinin, A. A. *et al.* A versatile information retrieval framework for evaluating profile strength and similarity. *bioRxiv* (2024).
- [23] Chandrasekaran, S. N. *et al.* Three million images and morphological profiles of cells treated with matched chemical and genetic perturbations. *Nature Methods* 1–8 (2024).
- [24] Tegtmeier, M. *et al.* High-dimensional phenotyping to define the genetic basis of cellular morphology. *Nature Communications* **15**, 347 (2024).
- [25] Esser, P., Rombach, R. & Ommer, B. Taming transformers for high-resolution image synthesis. In *Proceedings of the IEEE/CVF conference on computer vision and pattern recognition*, 12873–12883 (2021).
- [26] Gao, S., Zhou, P., Cheng, M.-M. & Yan, S. Masked diffusion transformer is a strong image synthesizer. In *Proceedings of the IEEE/CVF International Conference on Computer Vision*, 23164–23173 (2023).
- [27] Radford, A. *et al.* Learning transferable visual models from natural language supervision. In *International conference on machine learning*, 8748–8763 (PMLR, 2021).
- [28] Way, G. P. *et al.* Morphology and gene expression profiling provide complementary information for mapping cell state. *Cell systems* **13**, 911–923 (2022).
- [29] Nassiri, I. & McCall, M. N. Systematic exploration of cell morphological phenotypes associated with a transcriptomic query. *Nucleic acids research* **46**, e116–e116 (2018).
- [30] Wakui, T. *et al.* Predicting reprogramming-related gene expression from cell morphology in human induced pluripotent stem cells. *Molecular Biology of the Cell* **34**, ar45 (2023).
- [31] Cortes, C. & Vapnik, V. Support-vector networks. *Machine learning* **20**, 273–297 (1995).
- [32] Berger, V. W. & Zhou, Y. Kolmogorov–smirnov test: Overview. *Wiley statsref: Statistics reference online* (2014).
- [33] Agrawal, A., Ali, A., Boyd, S. *et al.* Minimum-distortion embedding. *Foundations and Trends® in Machine Learning* **14**, 211–378 (2021).
- [34] Xia, L., Lee, C. & Li, J. J. Statistical method scdeed for detecting dubious 2d single-cell embeddings and optimizing t-sne and umap hyperparameters. *Nature Communications* **15**, 1753 (2024).
- [35] Wolf, F. A., Angerer, P. & Theis, F. J. Scanpy: large-scale single-cell gene expression data analysis. *Genome biology* **19**, 1–5 (2018).
- [36] Trapotsi, M.-A., Hosseini-Gerami, L. & Bender, A. Computational analyses of mechanism of action (moa): data, methods and integration. *RSC Chemical Biology* **3**, 170–200 (2022).
- [37] Hu, Y. & Bajorath, J. Many structurally related drugs bind different targets whereas distinct drugs display significant target overlap. *RSC advances* **2**, 3481–3489 (2012).
- [38] Hu, Y., Lounkine, E. & Bajorath, J. Many approved drugs have bioactive analogs with different target annotations. *The AAPS journal* **16**, 847–859 (2014).
- [39] Heusel, M., Ramsauer, H., Unterthiner, T., Nessler, B. & Hochreiter, S. Gans trained by a two time-scale update rule converge to a local nash equilibrium. *Advances in neural information processing systems* **30** (2017).
- [40] Palma, A., Theis, F. J. & Lotfollahi, M. Predicting cell morphological responses to perturbations using generative modeling. *bioRxiv* 2023–07 (2023).
- [41] Meng, C. *et al.* Sdedit: Guided image synthesis and editing with stochastic differential equations. In *International Conference on Learning Representations* (2022).

Extended Response Tables

MOA	Drugs
calcium channel blocker	lidoflazine,nitrendipine,felodipine,levetiracetam,lomerizine,amlodipine,isradipine,gabapentin
dopamine receptor antagonist	haloperidol,prochlorperazine,molindone,thioridazine,triflupromazine
adrenergic receptor antagonist	pindolol,bisoprolol,phentolamine,prazosin
adrenergic receptor agonist	phenylephrine,naphazoline,methyldopa,clonidine
phosphodiesterase inhibitor	theobromine,milrinone,amrinone,cilostazol,pf-02545920
serotonin receptor antagonist	ondansetron,lerisetron,mdl-73005ef,mianserin
dopamine receptor agonist	bromocriptine,piribedil
NFkB pathway inhibitor	sasapyrine,erythromycin
EGFR inhibitor	lapatinib,arry-334543,canertinib,rociletinib
HMGCR inhibitor	pitavastatin,procaine,simvastatin,pravastatin

Extended Response Table 1: Drugs and their corresponding MOAs for the leave-one-out validation for different MOAs.

Target	Drugs
TOP2A	levofloxacin,voreloxin,merbarone,sparfloxacin
HRH1	levomequitazine,homochlorcyclizine,ketotifen,acrivastine
MLNR	erythromycin,camicinal
F10	otamixaban,rivaroxaban
ERBB2	arry-334543,cp-724714
PTGS2	etodolac,pterostilbene
CTSK	odanacatib,balicatib
TRPA1	hc-030031
PREP	baicalin
SLC6A1	ci-966

Extended Response Table 2: Drugs and their corresponding targets for the leave-one-out validation for different targets.

Extended Response Figures

Extended Response Figure 1: **MorphDiff predicts morphological changes on the target level.** Morphological features were extracted from both ground truth and generated images. The perturbed morphological features can be distinguished from the control, and the drugs targeting the same target cause similar changes in morphological space. The images generated by MorphDiff(I2I) and MorphDiff(G2I) are consistent with this distinction, while IMPA generates different clusters consisting of different perturbations. WD represents the Wasserstein Distance between perturbed and control CellProfiler features. The WD of MorphDiff is also closer to ground truth.

Extended Response Figure 2: (cont'd) **MorphDiff predicts morphological changes on the target level.** CellProfiler features were extracted from both ground truth and generated images for the other five targets. WD represents the Wasserstein Distance between perturbed and control CellProfiler features. The WD of MorphDiff is also closer to ground truth.

Extended Response Figure 3: **Overview of the MorphDiff Framework.** **a.** In this study, the multimodal dataset consists of the morphology images with five channels collected using CP (Cell Painting), and the gene expressions collected using L1000 assays. CellProfiler is then used to segment the cells and extract CellProfiler features at the single-cell level. Morphology images and gene expression together characterize the cell morphology responses to specific perturbations. **b.** MorphDiff is composed of two main components: Morphology VAE (MVAE) and Latent Diffusion Model (LDM). The MVAE encoder encodes the multi-channel cell morphology images into latent representation, and the decoder reconstructs the original cell morphology images based on the latent representation. LDM is trained to denoise from Gaussian random noise Z_T to morphology latent representation Z_0 recursively conditioned on L1000 gene expression. **c.** MorphDiff can be applied in two ways to generate cell morphology images with perturbations: L1000 gene expression to cell morphology generation (G2I, Gene to Image) and perturbed L1000 gene expression combined with control morphology images to perturbed cell morphology images generation (I2I, Image to Image). I2I is implemented with SDEdit [41] without re-training. **d.** Visualization of the generated output of different methods. Created with BioRender.com.

Extended Response Figure 4: **Visualization of the generated samples of MorphDiff, ground truth, and several baselines on genetic over-expression perturbation datasets.** We can observe that the generated samples of MorphNet are of relatively low quality and can not be distinguished across perturbations. The images generated by DRIT++ exhibit high variance across different perturbations, but the quality of the generated images is also low. StarGAN, DMIT and IMPA can generate relatively high-quality images, but the generated morphologies are unsatisfactory in some cases. Moreover, the generated results of VQGAN and Masked Diffusion Transformer are good, but may lose some details compared with ground truth. The visual quality of the images generated by the proposed method MorphDiff is the highest, and the generated images are the most visually similar to the ground-truth images.

Prediction of cellular morphology change under perturbations with transcriptome-guided diffusion model

Xuesong Wang^{*†1,2}, Yimin Fan^{*†1,2}, Yucheng Guo^{†1}, Chenghao Fu², Kinhei Lee², Khachatur Dallakyan², Yaxuan Li², Qijin Yin¹, Yu Li^{‡2,4,5,6,7}, and Le Song^{‡1,3}

¹BioMap Research, California, USA

²Department of Computer Science and Engineering, CUHK, Hong Kong SAR, China

³Mohamed bin Zayed University of Artificial Intelligence, Abu Dhabi, UAE

⁴The CUHK Shenzhen Research Institute, Hi-Tech Park, Nanshan, Shenzhen, 518057, China

⁵Institute for Medical Engineering and Science, Massachusetts Institute of Technology, Cambridge, MA, USA

⁶Wyss Institute for Biologically Inspired Engineering, Harvard University, Boston, MA, USA

⁷Broad Institute of MIT and Harvard, Cambridge, MA, USA

Response to reviewers

Manuscript ID: NCOMMS-24-43912A

Manuscript title: Prediction of cellular morphology change under perturbations with transcriptome-guided diffusion model

We sincerely thank the editor and all reviewers for their constructive and thoughtful comments, which have been invaluable in refining our manuscript. We have addressed each comment in detail in our responses. In this revised version, we have thoroughly addressed all reviewers' concerns by conducting additional benchmarking, statistical tests, analyses, evaluations, and textual revisions. We believe these changes have significantly enhanced the comprehensiveness and clarity of our manuscript, thanks to the insightful and valuable suggestions provided by all reviewers.

Blue: our responses to the reviewer's comments	Orange: references to the manuscript
Gray: reviewers' comments	Black: pointers to figures, tables, and sections

Summary of responses to reviewer comments

In response to the reviewers' comments, we present a summary of our key updates and responses below:

- **Enhanced interpretation of results and practical user guidelines:** We have made extensive modifications to enhance the interpretation of our results and have provided practical guidelines for users. Following the suggestion of Reviewer 2, we conducted comprehensive quantitative analyses on how well our models perform on different targets and MOAs and examine how out-of-distribution (OOD) dataset distance affects model performance across multiple levels. These analyses help interpret the applicability and potential use cases of our proposed model. Our findings indicate that our methods perform exceptionally well when input conditions closely resemble those of the training set, while performance may degrade when facing perturbations distant from the training distribution. Nevertheless, our approach still demonstrates substantially higher generalizability compared to baseline methods. Additionally, we have incorporated practical guidelines to assist users in applying our method to novel perturbation scenarios.

*Work was done while interning at BioMap.

†Equal first authorship.

‡Corresponding Author. Email: liyu@cse.cuhk.edu.hk and songle@biomap.com

- **Improving the robustness of performance benchmarking:** We have implemented comprehensive measures to ensure robust and thorough performance benchmarking. In response to the suggestion from Reviewer 4, we re-implemented all baseline methods to utilize five-channel cell morphology images instead of three-channel inputs, thereby establishing a more fair and consistent comparison framework. Additionally, we conducted extensive quantitative analyses to evaluate how the quality and distribution of reference control cell morphology images affect the performance of the I2I mode of MorphDiff. To minimize potential bias caused by the reference control cell morphology images, we sampled multiple groups of reference control cell morphology images for each method requiring such inputs. Furthermore, we revised our three-dimensional projection methodology to align with the standard protocols established in CellProfiler software. Through these collective improvements, we have improved our benchmark in terms of robustness and comprehensiveness.
- **Enhanced visualization of the results:** We have made extensive modifications to improve the visualization and analysis of the results. For visualization, following the suggestion of Reviewer 4, we have performed comprehensive visualization of MorphDiff and seven baseline methods on the JUMP ORF OOD dataset. For analysis, following the suggestion of Reviewer 2, we have used two high-dimensional statistical testing approaches to demonstrate the substantial distribution difference between the training and the OOD set, thus demonstrating that our performance evaluation on the OOD set effectively assesses model performance on distributions outside the domain of the training set.
- **Continuous improvement of manuscript quality:** We have continued to improve the quality of our manuscript following the suggestions from the reviewers. These improvements include enhanced clarity in descriptions of statistical testing, clarified statistical test settings, improved illustration of the model applications in revised Figure 1, improved descriptions of the projection method, enhanced readability of the last two paragraphs of the introduction section, expanded descriptions of acronyms, and revised citations. Additionally, we have carefully proofread the manuscript to minimize grammatical errors.

Datasets we use in the reply letter and revised manuscript

1. **JUMP:** We collected 130 perturbations and a control set that include both gene expression data [1] and morphology images from [2]. We split the dataset into three sets, which are training set, ID (in-distribution) set, and OOD (out-of-distribution) set. We randomly picked 10 percent of perturbations as OOD sets, resulting in 13 perturbations. The remained parts were randomly split into training set (90 percent) and ID set(10 percent), which indicates that the training set and ID set contain both control and perturbation data. We termed them as **JUMP Training set**, **JUMP ID set** and **JUMP OOD set**.
2. **CDRP:** We collected 959 perturbations and control set from CDRP [3;4]. Following the same split principle as JUMP dataset, we obtained 96 perturbations for OOD set, and 863 perturbations along with the control for training set and ID set. We termed them as **CDRP Training set**, **CDRP ID set** and **CDRP OOD set**. We additionally selected 10 targets (68 drugs annotated with 35 MOAs) from CDRP for MOA retrieval analysis, we termed it as **CDRP Target_MOA dataset**.
3. **LINCS:** To construct a dataset for model validation, we collected the bulk gene profiles from [4] (LINCS part) and downloaded the corresponding images from [5]. In this dataset, we selected 61 drugs that have sufficient images, along with available MOA and target labels. The components of these images are the same as those in the **CDRP** dataset. We first divided the data into 10 folds according to the 10 targets corresponding to 21 drugs. We termed it as **LINCS Target leave-one-out set**. For each experiment, we trained the model on another 9 folds and validated it on the remaining fold. We conducted experiments in a similar manner for the 10 MOAs (42 drugs) and termed it as **LINCS MOA leave-one-out set**.

Reviewer 2

The revision by Li et al is extensive and provides additional results to demonstrate the superiority of the proposed methodology over comparators.

Re:

Thanks for your appreciation of our revision and your efforts in providing valuable feedback. We have made comprehensive revisions for your comments in this round.

In summary, in response to your comments, we provided a detailed summary of the context for interpreting the results. We analyzed the key factors that impact model performance when applying our tool in out-of-distribution settings, thereby clarifying the potential use cases and the practical applicability of the methods. We have also conducted extensive analysis on the statistical tests in evaluating the degree of distribution difference of OOD datasets. We have also made substantial revisions in the manuscript, to improve the clarity of statements and avoid any potential confusion. The detailed point-by-point response is shown as follows.

While I appreciate the significant effort the authors have made to revise the manuscript, I remain concerned about the lack of a clear context in which to interpret the results, such as for which of the investigated MOAs/targets morphology prediction is most accurate. This issue is closely tied to potential use cases and limits the applicability of the findings.

Re:

Thank you for your constructive feedback and appreciation of our efforts. We understand your concerns regarding the potential use cases and applicability of our findings. We agree that a more detailed analysis interpreting the results of our work is needed. Following your suggestion, we have analyzed which of the investigated mechanisms of action (MOAs)/targets yield the most accurate morphology predictions, and what factors potentially impact prediction performance for specific MOAs and targets.

We hypothesized that prediction models may perform better on data that bear greater similarity to the training data, which is a fundamental consideration in machine learning that merits careful attention when evaluating predictive capabilities [6]. We applied this principle by inferring our model on unseen data. First, we analyzed the distance between the CDRP Training set and the CDRP Target_MOA dataset, with results shown in Response Figure 1.

Then, we used a straightforward approach to quantify the generalization capability of prediction models when handling out-of-domain test datasets belonging to different drugs, MOAs, and targets. Specifically, we calculated the Wasserstein Distance between the DeepProfiler embeddings of the CDRP Training set and the ground-truth CDRP Target_MOA dataset. We examined how performance on different metrics change with the Wasserstein Distance between the training and testing datasets at both target and MOA levels. The results are shown in Response Figure 2a and Extended Response Figure 1. As demonstrated, we found that all metrics exhibit a consistent negative correlation with respect to the distance with the CDRP Training set. Overall, we observe that model performance is affected by the distance between training and testing datasets, with degraded performance on more remote testing datasets. On the MOA level, our model makes the best predictions on HMGCR inhibitor, PPAR receptor agonist and EGFR inhibitor. On the target level, our model makes the best predictions on PPARG, EGFR and CDK1.

Having found that the distance between ground-truth cell morphology and training cell morphology can impact model performance, we next sought to provide practical user guidelines. In real-world applications, users will input perturbed gene counts directly into the MorphDiff model and utilize the predicted cell morphology responses without knowing the ground-truth cell morphology distribution. Therefore, we quantified whether the Wasserstein Distance between the training gene counts and the inference gene counts affects model performance. As shown in Response Figure 2b and Extended Response Figure 2, there is indeed a negative correlation between this distance and performance metrics. This indicates that models may have degraded and less reliable performance when inferring cell morphology using gene count distributions that are distant from those in the training set. We also compared the performance of MorphDiff with IMPA in Response Figure 2c and Extended Response Figure 3. Specifically, we divided the drug MOAs in the out-of-distribution (OOD) set into five equal-sized bins based on their distance from the training dataset, calculated using drug embeddings (the input of IMPA) and gene counts (the input of MorphDiff). We found that under both settings, regardless of whether the test input conditions were closer to or further from the training distribution, MorphDiff consistently outperforms the baseline method in most cases, highlighting MorphDiff's superior generalizability when predicting OOD perturbations.

Response Figure 1: Statistics showing the Wasserstein Distance between the DeepProfiler embeddings of the CDRP Training set and the CDRP Target_MOA dataset.

In Response Figure 2d, we present an illustrative guideline for method users. When users want to apply the model to infer cell morphological responses from new L1000 gene expression profiles in their applications, we suggest first checking the similarity between the new L1000 profile and the L1000 database used in MorphDiff training. Higher similarity indicates that results are more likely to be reliable; conversely, lower similarity suggests that users should exercise caution, as predicting distant perturbations in out-of-distribution (OOD) scenarios remains challenging for all tools, though MorphDiff demonstrates superior performance compared to other tools in such cases. This phenomenon has also been discussed in previous works related to predicting morphological perturbation response and transcriptional response. For example, in IMPA [7], the authors explicitly acknowledged that “its predictive ability and performance for perturbations that are very different from those in the training set may decrease.” Similarly, CellOT [8], which focuses on predicting transcriptional response, stated that “The ability to make o.o.d. predictions, however, is only feasible if (1) similar samples have been observed in the unperturbed setting and (2) the training set contains cases that are similar not only in their unperturbed state but also their perturbation response.” These observations reinforce our position that predicting distant OOD perturbations remains challenging for current methods. Addressing this challenge will require both the curation of large-scale, high-quality, comprehensive datasets and the development of novel learning paradigms.

Following your comments, we have added some additional analysis in the section 2.4 of the revised manuscript as follows

We further investigated how the performance of MorphDiff methods varies across different MOAs and targets. Specifically, we evaluated how performance is affected by the distance between the CDRP Training set and the evaluated CDRP Target_MOA dataset. As shown in Appendix Figure 25-29, we found that model performance is influenced by the similarity between training and evaluation datasets, with performance degrading as the evaluation compounds become more distant from the training data. Consequently, when applying MorphDiff methods, researchers should consider potential performance limitations for test compounds that differ significantly from those in the training set. More detailed analysis is described in Supplementary Notes 3.

Following your comments, we have added additional discussion in the discussion section of the revised manuscript to better demonstrate the potential use cases and applicability to users.

Response Figure 2: Evaluating the use cases and applicability of MorphDiff. **a.** Generalization capabilities of MorphDiff(I2I) and MorphDiff(G2I) when generalizing to new MOAs on the CDRP OOD dataset. Correlation between the Wasserstein Distance of the training and ground-truth OOD data and the normalized metrics at MOA levels. The x-axis represents the Wasserstein Distance between DeepProfiler embeddings of the training and testing datasets. The y-axis displays the normalized performance metrics. The black line indicates a linear regression between the x-axis and y-axis values. **b.** Correlation between the Wasserstein Distance of training gene count and inference OOD L1000 gene count and the normalized metrics at MOA level. The x-axis represents the Wasserstein Distance between the input gene counts of the training and testing datasets. The y-axis displays the normalized performance metrics. **c.** Comparison of the performance of MorphDiff methods and IMPA method on the generalization capabilities with respect to the distance of drug embedding / gene count. The CDRP Target_MOA dataset were divided into five bins based on their Wasserstein Distance from the training dataset. These Wasserstein Distances were computed using gene counts and drug embeddings. All performance metrics were divided by the best performance metrics achieved on each MOA. **d.** Recommendation and practical guidelines for users of our tool. Extended Response Figure 1-3 are extensions of this figure.

For future work... In addition, while extensive benchmarks have demonstrated that MorphDiff outperforms other baseline methods in predicting out-of-distribution perturbations, we observed that its performance may degrade when applied to perturbations that deviate significantly from the training dataset distributions (Supplementary Notes 3). Although this limitation is expected for most deep learning methods in predicting perturbation response, including previous works such as CellOT [8] and IMPA [7], the curation of large-scale, high-quality, comprehensive datasets and the development of novel learning paradigms may further enhance generalization performance when predicting responses to distant perturbations...

Following your comments, we have added the above analysis in the Supplementary Notes 3 in the revised manuscript. The modifications are shown as follows:

Analyzing the applicability of MorphDiff on OOD MOAs and targets

Prediction models may perform better on data that bear greater similarity to the training data, which is a fundamental consideration in machine learning that merits careful attention when evaluating predictive capabilities. We applied this principle by inferring our model on unseen data. First, we analyzed the distance between the training dataset and the MOA/Target dataset in the CDRP dataset, with results shown in Appendix Figure 25.

Then, we used a straightforward approach to quantify the generalization capability of prediction models when handling out-of-domain test datasets belonging to different drugs, MOAs, and targets. Specifically, we calculated the Wasserstein Distance between the DeepProfiler embeddings of the CDRP Training set and the ground-truth CDRP Target MOA dataset. We examined how performance on different metrics change with the Wasserstein Distance between the training and OOD datasets at both target and MOA levels. The results are shown in Appendix Figure 26a and Appendix Figure 27. As demonstrated, we found that all metrics exhibit a consistent negative correlation with respect to the distance with the CDRP Training set. Overall, we observe that model performance is affected by the distance between training and OOD datasets, with degraded performance on more remote OOD datasets. On the MOA level, our model makes the best predictions on HMGR inhibitor, PPAR receptor agonist and EGFR inhibitor. On the target level, our model makes the best predictions on PPARG, EGFR and CDK1.

Having found that the distance between ground-truth cell morphology and training cell morphology can impact model performance, we next sought to provide practical user guidelines. In real-world applications, users will input perturbed gene counts directly into the MorphDiff model and utilize the predicted cell morphology responses without knowing the ground-truth cell morphology distribution. Therefore, we quantified whether the Wasserstein Distance between the training gene counts and the inference gene counts affects model performance. As shown in Appendix Figure 26b and Appendix Figure 28, there is indeed a negative correlation between this distance and performance metrics. This indicates that models experience degraded and less reliable performance when inferring cell morphology using gene count distributions that are distant from those in the training set. We also compared the performance of MorphDiff with IMPA in Appendix Figure 26c and Appendix Figure 29. Specifically, we divided the drug MOAs in the out-of-distribution (OOD) set into five equal-sized bins based on their distance from the training dataset, calculated using drug embeddings (the input of IMPA) and gene counts (the input of MorphDiff) respectively. All performance metrics were divided by the best performance metrics achieved on each MOA. We found that under both settings, regardless of whether the test input conditions were closer to or further from the training distribution, MorphDiff consistently outperforms the baseline method in most cases, highlighting MorphDiff's superior generalizability when predicting OOD perturbations.

In Appendix Figure 26d, we present an illustrative guideline for method users. When users want to apply the model to infer cell morphological responses from new L1000 gene expression profiles in their applications, we suggest first checking the similarity between the new L1000 profile and the L1000 database used in MorphDiff training. Higher similarity indicates that results are more likely to be

reliable; conversely, lower similarity suggests that users should exercise caution, as predicting distant perturbations in out-of-distribution (OOD) scenarios remains challenging for all tools, though MorphDiff demonstrates superior performance compared to other tools in such cases. This phenomenon has also been discussed in previous works related to predicting morphological perturbation response and transcriptional response. For example, in IMPA [7], the authors explicitly acknowledged that “its predictive ability and performance for perturbations that are very different from those in the training set may decrease.” Similarly, CellOT [8], which focuses on predicting transcriptional response, stated that “The ability to make o.o.d. predictions, however, is only feasible if (1) similar samples have been observed in the unperturbed setting and (2) the training set contains cases that are similar not only in their unperturbed state but also their perturbation response.” These observations reinforce our position that predicting distant OOD perturbations remains challenging for current methods. The curation of large-scale, high-quality, comprehensive datasets and the development of novel learning paradigms could be promising future directions to further improve the generalization capabilities of perturbation prediction models

Some minor concerns:

Lines 203–205: The authors mention that the K-S test and KL divergence (Kullback-Leibler?) were ‘experimented’ with to further demonstrate the superiority of their method. Could the authors clarify how a significant K-S test or KL divergence would substantiate such claims of superiority? Additionally, the term ‘experimenting’ with statistical evaluations might imply exploratory analysis, which could be perceived as inconsistent with best practice.

Re:

Thanks for the comments. We presented a clear explanation as below and modified the manuscript accordingly.

In previous manuscript lines 203-205, we stated that “We also experimented with other types of statistical tests and metrics, including t-test, Kolmogorov-Smirnov test and KL-divergence, in Appendix Figure 6 for MorphDiff(G2I), MorphDiff(I2I) and IMPA to further demonstrate the superiority of our method.” Revised Appendix Figure 8 (Previous Appendix Figure 6) serves as a supplementary analysis to the revised Figure 2e (Response Figure 5e) in the main manuscript.

In the revised Figure 2e (Response Figure 5e), we employed the Wilcoxon signed-rank test to assess the distribution similarity between the ground-truth CellProfiler features and the generated CellProfiler features. If the p -value of the Wilcoxon signed-rank test exceeds 0.05, we conclude that there is insufficient evidence to indicate significant differences between the generated perturbed morphological features and the ground-truth perturbed morphological features. We then calculated the proportion of generated CellProfiler features that do not significantly differ from the ground-truth CellProfiler features according to this statistical test. Consequently, a higher proportion indicates greater similarity between the generated features and the ground-truth features, and we found that the generated features from MorphDiff have a much higher proportion compared with those of IMPA.

In the previous round of revision, Reviewer 4 questioned whether our method would demonstrate consistent advantages across different types of statistical tests. In response, we employed various statistical measures as suggested including t-test, Kolmogorov-Smirnov test, and KL-divergence to evaluate the robustness of our results. For the t-test and Kolmogorov-Smirnov test, we used the same evaluation method as depicted in revised Figure 2e (Response Figure 5e). For KL-divergence, we established a threshold (3), and considered the distribution of a CellProfiler feature in the generated and ground-truth cell morphology images to be sufficiently similar if the KL-divergence was less than this threshold. Similarly, a high proportion of CellProfiler features showing high similarity between generated and ground-truth images indicates better overall consistency between the generated and ground-truth distributions.

As demonstrated in Appendix Figure 8, our method consistently outperforms the baselines across multiple statistical assessments, confirming that our findings are robust and not dependent on the choice of statistical test. We acknowledge that the original wording in lines 203-205 could potentially lead to confusion; therefore, we have clarified this explanation to better contextualize the supplementary analyses in relation to the revised Figure 2e (Response Figure 5e) in the main manuscript as follows:

We also repeated the analysis in Figure 2e with other types of statistical tests and metrics, including t-test, Kolmogorov-Smirnov test and KL-divergence, in Appendix Figure 8 for MorphDiff(G2I),

Response Figure 3: The statistical test results distribution for each dimension in the previous manuscript. a. On CDRP drug dataset. b. on JUMP dataset.

MorphDiff(I2I) and IMPA to further demonstrate the robustness and consistency of our conclusion.

The authors state that the K-S test was used to verify that the OOD datasets had different distributions. Could the authors specify which variable(s) the K-S test was applied to, or clarify if it was applied in a multivariate context?

Re:

Thanks for the comments and feedback. To clarify, in the previous version, we performed the K-S test between the **DeepProfiler embeddings (672 dimension)** of the training and OOD datasets. We used the K-S tests implemented in Scipy (*scipy.stats.ks_2samp*). As this test is implemented to evaluate uni-variate rather than multivariate data, we averaged the statistics across each dimension and reported the results. The statistical test results for each dimension are shown in Response Figure 3. We can observe that on the OOD dataset, the p -value for most dimensions is low, highlighting that the OOD dataset indeed has a significantly different distribution from the training set.

After reading your comments and thinking carefully, we agree that the previous approach is not a standard method for performing tests on high-dimensional data. Indeed, there are multivariate implementations of the K-S test [9]. However, K-S tests on high-dimensional datasets can be quite time-consuming and resource-intensive, and to the best of our knowledge, there are no highly efficient software packages that can perform high-dimensional K-S tests at the scale of our dataset. Therefore, to make the evaluation of the OOD dataset more rigorous and the results more convincing, we explored alternative high-dimensional statistical tests to further justify the distribution difference between the training and the OOD dataset.

To demonstrate that the out-of-distribution (OOD) datasets indeed have a significantly different distribution compared to the training datasets, we implemented other two statistical tests comparing the DeepProfiler embeddings of the training and in-distribution (ID)/out-of-distribution (OOD) datasets. These tests further validate the statistical significance of the differences between the training and evaluation distributions. Specifically, we conducted both tests in a multivariate manner to comprehensively assess distributional differences.

The first test is the Maximum Mean Discrepancy (MMD) test [10] between the DeepProfiler embeddings of the training and ID/OOD datasets. The MMD test determines if two groups come from different distributions by measuring their distance in a high-dimensional feature space. The key steps are: (1) transform data using an RBF kernel, (2) calculate the MMD statistic by comparing within-group similarities to between-group similarities, and (3) assess significance through permutation testing—repeatedly shuffling labels and recalculating MMD statistic to generate a null distribution. The resulting p -value indicates whether the observed difference is statistically significant, with smaller values suggesting the samples come from different distributions. Due to the high computational complexity, we randomly sampled 2000 cell morphology images from each dataset and conducted tests using 999 permutations. To ensure robustness and stability, we repeated these statistical tests for 10 rounds. We presented

the statistic value and p -value (10 rounds) for CDRP dataset, JUMP dataset, LINCS Target leave-one-out set and LINCS MOA leave-one-out set respectively in Extended Response Table 1, 3, 5, 7.

We conducted the second evaluation using the energy distance test to compare DeepProfiler embeddings between the training and ID/OOD datasets, employing the `dcor` package [11]. This widely-used statistical package is designed for analyzing high-dimensional datasets. Specifically, we utilized the function `dcor.homogeneity.energy_test` to perform our statistical analyses. The energy test assesses whether multiple groups originate from the same distribution by examining the Euclidean distances between data points. This method compares point-pairwise distances within each group to those between different groups. If points are more closely clustered within their own groups than with points from other groups, this pattern indicates that these groups likely stem from different underlying distributions. The test employs permutation testing to generate a p -value, with lower values indicating that groups likely originate from different distributions. Due to the computational complexity of this analysis, we randomly sampled 2000 cell morphology images from each dataset and conducted tests using 999 permutations. We used the same sampling strategy as the MMD test. To ensure robustness and stability, we repeated these statistical tests for 10 rounds. The statistic values and p -values (10 rounds) for the CDRP dataset, JUMP dataset, LINCS Target leave-one-out set, and LINCS MOA leave-one-out set are presented in Extended Response Tables 2, 4, 6, and 8, respectively.

The statistical testing results presented in Extended Response Tables 1-8 consistently support our conclusions across two different types of statistical tests and multiple datasets (JUMP, CDRP, and LINCS MOA leave-one-out). These results demonstrate that the training set and the out-of-distribution (OOD) set derive from distributions with statistically significant differences, while the training set and the in-distribution (ID) set derive from distributions with no significant differences. This validation confirms that our performance evaluation on the OOD set effectively assesses model performance on distributions outside the domain of the training set.

To further improve the clarity and readability of our manuscript, we have revised in the manuscript in Section 4.3 as follows:

Statistical tests for assessing the distribution difference We performed statistical tests to assess whether the OOD set and leave-one-out set exhibit statistically significant distribution differences from the training set. Specifically, we conducted two statistical tests: Energy Distance tests [11], and Maximum Mean Discrepancy tests [10] to evaluate the distribution differences between the DeepProfiler embeddings of the training set and the ID/OOD sets. For the Energy Distance test, we employed the `dcor.homogeneity.energy_test` function from the `dcor` package [11]. The energy test assesses whether multiple groups originate from the same distribution by examining the Euclidean distances between data points. This method compares point-pairwise distances within each group to those between different groups. If points are more closely clustered within their own groups than with points from other groups, this pattern indicates that these groups likely stem from different underlying distributions. For the Maximum Mean Discrepancy (MMD) tests [10], we determined if two groups come from different distributions by measuring their distance in a high-dimensional feature space. The key steps are: (1) transform data using an RBF kernel, (2) calculate the MMD statistic by comparing within-group similarities to between-group similarities, and (3) assess significance through permutation testing—repeatedly shuffling labels and recalculating MMD statistic to generate a null distribution. Due to the computational complexity of these two analysis, we randomly sampled 2,000 cell morphology images from each dataset and conducted tests using 999 permutations. To ensure robustness and stability, we repeated all statistical tests for 10 rounds. The statistic values and p -values for the JUMP dataset, CDRP dataset, LINCS Target leave-one-out set, and LINCS MOA leave-one-out set across all three types of tests are presented in Appendix Tables 3-10. The statistical testing results consistently demonstrate that the training set and the OOD or leave-one-out sets derive from distributions with statistically significant differences, while the training set and the ID set derive from distributions with no significant differences. This validation confirms that our performance evaluation on the OOD set and leave-one-out set effectively assesses model performance on distributions outside the domain of the training set.

Reviewer 4

The authors have made substantial changes to the manuscript in response to the previous review. I believe it is greatly improved, but I have some additional comments.

Re:

Thanks for your comments. In this revision, we have made comprehensive revisions on the comments you provided and we believe that they are very helpful and further improve the quality of our work.

In summary, in response to your comments, we have conducted extensive experiments to enhance the robustness of our benchmarking results. We have re-implemented all baseline methods according to your suggestion, enabling them to accept five-channel cell morphology images as input, and applied the standard three-channel image composition method defined in CellProfiler software for benchmark evaluation and visualization. Additionally, we performed comprehensive experiments analyzing how the quality and distribution of reference control DMSO cell morphology images affect the performance of the I2I mode of our MorphDiff model. We have also made substantial improvements to the readability and clarity of the manuscript, including enhanced visualizations in revised Figure 1 and 2, rephrasing of unclear sentences, revision of citations, and careful correction of grammatical errors. We believe that the quality of our manuscript has further improved thanks to your insightful and constructive feedback. The detailed point-by-point responses are as following.

I appreciate the inclusion of example images in figure 1d, but it's current presentation is not ideal. First, it does not fit with the rest of figure 1 as the rest of the figure shows flowcharts of the model architecture. The visualization is a result and works better in conjunction with the results from figure 2a because it can provide examples that demonstrate the trends shown in the metrics. For example, the explanation about why DMIT has higher density could be supported by pointing to specific examples. More details should also be included in the figure and/or caption (e.g., is there significance to the four examples chosen? Are there qualities of the proposed model's outputs that you are trying to highlight?).

Re:

Thanks for your comments and suggestions. We have addressed your comments and suggestions as follows.

We appreciate your suggestions regarding the presentation of Figure 1. In the revised version, we have replaced panel d with an illustration that describes the main applications of our framework. This modification allows readers to gain a more intuitive understanding of how our framework can be applied in practice. The revised Figure 1 is also shown in Response Figure 4. In the revised Section 2.1 in the manuscript, we also made the following modifications:

In Figure 1d, we outlined the key applications of MorphDiff in practice. Primarily, MorphDiff can predict cell morphology treated by novel perturbations not encountered during training, which fulfills our main objective of exploring the vast perturbation space through in-silico inference. Furthermore, our framework incorporates CellProfiler [12] and DeepProfiler [13] to generate biologically meaningful morphological features and embeddings, respectively. This integration enables researchers to probe specific morphological feature changes following perturbation, thereby enhancing interpretability and practical applicability. As a critical application, the generated morphological profiles can be utilized for retrieval and identification of drug MOA, establishing MorphDiff as a powerful tool in phenotypic drug discovery.

We greatly appreciate your suggestion to integrate visualization results in Figure 2a. The revised Figure 2 is shown in Response Figure 5.

For your question “is there significance to the four examples chosen? Are there qualities of the proposed model's outputs that you are trying to highlight?”. In the previous version of our manuscript, we arbitrarily select four examples to demonstrate the visual generation quality of MorphDiff. For comprehensive evaluation, in this revision, we showed diverse JUMP OOD perturbations in Response Figure 5a (Revised Figure 2) and Extended Response Figures 4, 5, and 6 (Revised Appendix Figure 1,2,3). Since the models can generate multiple images for a given perturbation, we established the following criteria for selecting images for visualization. For each row of images, we selected the generated image from each method that had the highest structural similarity (SSIM) [15] to the ground-truth image. This approach prevents selective bias and ensures fair visual comparison across methods.

It is vital to notice that interpreting all generative performance metrics from visual output alone is not feasible,

Response Figure 4: **Overview of the MorphDiff Framework.** **a.** In this study, the multimodal dataset consists of the morphology images with five channels collected using CP (Cell Painting), and the gene expressions collected using L1000 assays. CellProfiler is then used to segment the cells and extract CellProfiler features at the single-cell level. Morphology images and gene expression together characterize the cell morphology responses to specific perturbations. **b.** MorphDiff is composed of two main components: Morphology VAE (MVAE) and Latent Diffusion Model (LDM). The MVAE encoder encodes the multi-channel cell morphology images into latent representation, and the decoder reconstructs the original cell morphology images based on the latent representation. LDM is trained to denoise from Gaussian random noise Z_T to morphology latent representation Z_0 recursively conditioned on L1000 gene expression. **c.** MorphDiff can be applied in two ways to generate cell morphology images with perturbations: L1000 gene expression to cell morphology generation (G2I, Gene to Image) and perturbed L1000 gene expression combined with control morphology images to perturbed cell morphology images generation (I2I, Image To Image). I2I is implemented with SDEdit [14] without re-training. **d.** Illustration of the downstream applications of MorphDiff, including prediction on novel perturbations, feature analysis as well as morphology-based MOA retrieval. DMSO stands for Dimethyl sulfoxide, which is considered as control group without perturbation. Created with BioRender.com.

as these metrics are calculated based on distribution similarity between the generated and ground-truth samples rather than individual image comparisons. Nevertheless, in some cases, visualization could present readers an intuitive understanding of generation quality and can potentially help to interpret certain generative evaluation metrics. From the visualizations presented in Response Figure 5a and Extended Response Figures 4, 5, and 6, the generated outputs of both modes of MorphDiff display colors and morphological structures that closely resemble the ground truth. DMIT, IMPA and MDTv2 produce reasonably good images that adequately capture the

general image features, but they fall short in detail and clarity. MorphNet shows promise in generating reasonable images with some detail, but it lacks diversity. StarGAN produces diverse outputs with good morphological structures but exhibits subtle color bias compared to the ground truth. For DRIT++ and VQGAN, the visual quality appears worse than other methods, with either reduced clarity or noticeable color bias. Overall, MorphDiff (I2I), MorphDiff (G2I), IMPA, and DMIT demonstrate the highest image generation quality, with both MorphDiff variants producing images that closely resemble the ground truth.

Notably, you also commented that “the explanation about why DMIT has higher density could be supported by pointing to specific examples”. In response to your last comments on the model performance inconsistency, we sampled 10 groups of control morphology images and used them as reference sets to evaluate the performance of MorphDiff(I2I) and other methods that rely on reference control morphology images as input, ensuring the robustness of the results (details are provided in our response to your last comments on model performance). The 10 distinct groups of reference sets introduce variance for each method requiring a reference set, which may potentially influence the statistical difference analysis between methods. As shown in Response Figure 5b (revised Figure 2b), there is no longer a significant performance difference in the density score between the MorphDiff (both modes) and the DMIT. Therefore, we did not further elaborate on this comment in the revised manuscript.

To improve the clarity of our manuscript, following your suggestion, we have also made substantial modifications in the revised manuscript. Regarding the discussion on the visualization results, we have added in the revised manuscript:

First, we conducted a visual assessment of the generation quality between MorphDiff methods and baseline approaches, as illustrated in Figure 2a. For this visual comparison, we presented results from the SREBF1 genetic perturbation, which represents an out-of-distribution (OOD) perturbation (all JUMP OOD perturbations). Visualizations for the additional nine OOD genetic perturbations are available in Appendix Figures 1-3. The specific criteria used for selecting these images for visualization are detailed in Supplementary Notes 1. The generated outputs of both modes of MorphDiff display colors and morphological structures that closely resemble the ground truth. DMIT, IMPA and MDTv2 produce reasonably good images that adequately capture the general morphological features, but they fall short in detail and clarity. MorphNet shows promise in generating reasonable images with some detail, but it lacks diversity. StarGAN produces diverse outputs with good morphological structures but exhibits subtle color bias compared to the ground truth. For DRIT++ and VQGAN, the visual quality appears worse than other methods, with either reduced clarity or noticeable color bias. Overall, MorphDiff (I2I), MorphDiff (G2I), IMPA, and DMIT demonstrate the highest image generation quality, with both MorphDiff variants producing images that closely resemble the ground truth.

...The results also show that MorphNet achieves a nearly zero coverage score, demonstrating that the generative output of MorphNet lacks diversity, which aligns with the visualization in Figure 2a...

Regarding the criteria for choosing the images for visualization, we have added in the Supplementary Notes 1 in the revised manuscript:

As the models can generate multiple images for a given perturbation, we established the following criteria for selecting images for visualization. For each row of images, we selected the generated image from each method that had the highest structural similarity (SSIM) [15] to the ground-truth image. This approach prevents selective bias and ensures fair visual comparison across methods.

*I have some additional comments about the projection into three channels. First, how were these weights determined? Is there a biological justification for them or is it optimized in some way? The description of the projection is also overly complicated with several equations and variables defined that aren't used elsewhere. It is sufficient to just say the channels are $(1 * DNA)$, $(1 * ER + 0.4 * AGP + 0.4 * Mito)$, $(1 * RNA + 0.4 * AGP + 0.2 * Mito)$ or use a matrix: $C2 = [1, 0, 0, 0, 0; 0, 1, 0, 0.4, 0.4; 0, 0, 1, 0.4, 0.2] * C1$*

Re:

We greatly appreciate your feedback. In our initial approach, we selected weights based on the visual quality of the composed images. For this revision, however, we have enhanced the robustness and fidelity of our experimental results by adopting the standardized weights used in CellProfiler—an established cell morphology analysis software [12]. CellProfiler incorporates a composition function specifically designed to convert five-channel cell

Response Figure 5: **MorphDiff predicts changes in cell morphology with genetic perturbations.** **a.** Visualization of the generated outputs from MorphDiff and baseline methods for the SREBF1 perturbation demonstrates that MorphDiff methods achieve the highest generation quality and more closely resemble the ground-truth images. **b.** Performance benchmarking on the JUMP OOD set. MorphDiff(G2I) denotes generating the perturbation morphology images with only the perturbation gene expression. MorphDiff(I2I) denotes generating the perturbation morphology with the perturbation gene expression and a control reference image. Linear normalization was conducted on the reciprocals of FID, CMMD, and on the other three metrics separately to convert their values to the range of 0 to 1. A larger value indicates better performance. **c.** UMAP visualization of the generated cell morphology CellProfiler feature on two genetic perturbations for IMPA and MorphDiff(G2I). DMSO stands for Dimethyl sulfoxide, which is considered as control group without perturbation. **d.** Distribution of CellProfiler features between the ground truth, generated morphology from IMPA, generated morphology from MorphDiff(G2I) and MorphDiff(I2I) on RAC1 genetic perturbation. *E* stands for *ERSyto*. *EB* stands for *ERSytoBleed*. *H* stands for *Hoechst*. *Pg* stands for *Ph_golgi*. *LQ* stands for *LowerQuartile*. *Var* stands for *Variance*. *Tex* stands for *Texture*. *Corr* stands for *Correlation*. *Int* stands for *Intensity*. **e.** The Wilcoxon signed-rank test results to test the difference between the generated cell morphology CellProfiler features and the ground-truth CellProfiler features. The *y*-axis indicates the proportion of generated CellProfiler features not significantly differ from the ground-truth CellProfiler features through a Wilcoxon signed-rank test ($p < 0.05$). Across different types of perturbations, morphology generated by MorphDiff(G2I) consistently has more CellProfiler features that do not have significant differences with ground truth, which indicates the high generation quality of MorphDiff(G2I). **f.** The heatmap of the correlation between the CellProfiler features of the ground-truth morphology, MorphDiff(G2I)-generated morphology, and IMPA-generated morphology with L1000 gene expression.

morphology images into three-channel images for visualization purposes, and we have implemented the same weights in our analysis. The weights assigned to the different channels are as follows: ER(1, 0, 0), RNA (0, 1, 0), DNA(0, 0, 1), Mito(0.5,0.5,0), AGP (0.5, 0, 0.5). Regarding the writing issues, we agree that it is sufficient to describe the formulation of the composition function as follows: $(1 * ER + 0.5 * Mito + 0.5 * AGP, 1 * RNA + 0.5 * Mito, 1 * DNA + 0.5 * AGP)$. This formulation reflects the weights employed in CellProfiler. We have revised Section 4.4 “Evaluation Setup” accordingly:

...For performance benchmarking, the five generative metrics can only be applied to three-channel RGB images. Therefore, we converted the five-channel output to three-channel images for benchmarking, following the methodology outlined in CellProfiler [12]. The weights assigned to the different channels are as follows: ER(1, 0, 0), RNA (0, 1, 0), DNA(0, 0, 1), Mito(0.5,0.5,0), AGP (0.5, 0, 0.5). Consequently, the resulting composited image is calculated as: $(1 * ER + 0.5 * Mito + 0.5 * AGP, 1 * RNA + 0.5 * Mito, 1 * DNA + 0.5 * AGP)$.

Finally, is there a reason why this projection was used for the baseline models rather than adjusting the architecture to accept five channel input? It shouldn't be too difficult to modify these models to accept five channels. While pretrained model weights could not be used, I believe the models are being retrained for this task anyway. It seems like a significant disadvantage for these baselines to not be able to use the full input data.

Response Figure 6: **a.** Revised Figure 2b. Benchmarking on JUMP OOD dataset [2]. **b.** Revised Figure 3a. Benchmarking on CDRP OOD dataset [16; 3] (Drug perturbation).

Re:

Thank you for your valuable comments. In the original manuscript, we noted that six baseline methods in our study originally accept three-channel images as input, as they are designed for general image generation tasks. We did not modify their model configurations to ensure the integrity of the baseline methods as they were intended by their original developers.

Following your suggestion, in this revision, we modified the baseline methods from using three-channel images input to accepting five-channel images input. As these baseline methods employ different implementation styles and most are based on widely used computer vision architectures, modifying them to accept five-channel input is not trivial and requires substantial technical adjustments.

Nevertheless, despite these technical challenges, we have modified all the architectures to accept five-channel input and retrained the adjusted models (MorphNet [17], StarGANv1 [18], DMIT [19], VQGAN [20], MDTv2 [21], and DRIT++ [22]) for the revised evaluation, as shown in Response Figure 6. This adjustment ensures that each model can utilize the complete input data, allowing for a more equitable comparison. After modifying the baseline methods from three-channel to five-channel inputs, the performance comparison shows some variations, as illustrated in Extended Response Figures 7 and 8. However, the overall trend remains largely consistent. Importantly, our method continues to achieve the highest performance in most evaluation scenarios. We have also implemented the following modifications in the revised manuscript’s methods section.

During training, methods that directly support five-channel cell morphology images as input (MorphDiff and IMPA) were trained using these five-channel images. For methods that originally could only process RGB three-channel images, we modified their implementations to enable them to accept and generate five-channel images.

In the previous manuscript, we mistakenly used the Wilcoxon rank sum test instead of the Wilcoxon signed-rank test as stated in some analysis. We identified this issue and revised the statistical test results, as shown in

Response Figure 6. Additionally, we updated the statistical tests in Revised Figure 3a, 4a, b, and e, as well as in Revised Appendix Figure 13 and 20. Similar to previous analyses, MorphDiff exhibits statistically significant advantages compared to baseline methods in most cases, with p -values less than 0.05. We revised the analysis in the manuscript accordingly. In revised section 2.2:

...It should be noted that star (***) means that MorphDiff surpasses the baseline method with a p -value less than 0.05. The 'ns' represents the situation of no significant difference...

In revised section 2.3:

We first benchmarked all methods on the CDRP OOD set with the general evaluation metrics, in which the meanings of stars are the same as in Section 2.2 and the hashtag (#) indicates the situation that the baseline performs better significantly...

in revised section 2.4:

...MorphDiff(I2I) significantly outperforms IMPA across all image generation quality metrics, with a corrected p -value of less than 0.05...

...outperforming IMPA for 6 of 10 targets with corrected p -values less than 0.05...

...MorphDiff not only outperforms gene expression but also significantly eclipses IMPA regarding retrieval accuracy, as indicated by p -values of less than 0.05, based on Wilcoxon signed-rank test with Bonferroni correction...

Minor comments: The last two paragraphs of the introduction section largely summarize the results. The manuscript would flow better if the introduction were limited to setting up the goals and hypotheses and leaving the results to later sections.

Re:

Thanks for your comments. Following your suggestions, we have revised the last two paragraphs of the introduction as follows:

To evaluate MorphDiff's capability for predicting cell morphology responses to novel perturbations, we established a comprehensive assessment framework. Our methodology tested the model's performance across three extensive datasets encompassing a wide range of genetic and drug perturbations. We assessed the model using both standard image generation metrics and measures of biological relevance by analyzing interpretable morphological features extracted through established computational tools such as CellProfiler [12] and DeepProfiler [23]. Furthermore, we examined MorphDiff's effectiveness in capturing correlations between transcriptional and morphological responses to perturbations, potentially providing insights into how changes in gene expression manifest as alterations in cellular morphology.

A key application we aim to investigate is MorphDiff's potential in phenotypic drug discovery, particularly for MOAs retrieval. While traditional approaches focus on drug structure analysis and transcriptome response analysis [24; 25], we emphasize that cell morphology information—whether directly observed or computationally inferred—provides complementary and valuable signals for mechanism of action (MOA) identification. Through our designed MOA retrieval pipeline, we demonstrated that MorphDiff-generated morphologies achieve comparable performance to ground-truth morphologies and outperform existing baseline methods and gene expression-based approaches. Furthermore, we explored the potential of our approach to discover drugs with different molecular structures but similar MOA. In summary, MorphDiff is a powerful tool in phenotypic drug discovery by accurately generating cell morphology under unseen perturbations, with promising applications in facilitating the exploration of the vast phenotypic perturbation screening space and assisting in determining the MOAs of structurally diverse drugs.

Predicting cell morphological responses to perturbations using generative modeling is cited as biorxiv, but it was recently published in Nature Communications

Re:

Thanks for your comments. We have changed the citation from biorxiv to the published version.

Palma, A., Theis, F. J. & Lotfollahi, M. Predicting cell morphological responses to perturbations using generative modeling. *Nature Communications* 16, 505 (2025)

The acronym DMSO appears several times (figure 1d, figure 2b, and section 4.1), but it is not defined and it also is not mentioned in the figure captions.

Re:

Thanks for your comments. Dimethyl sulfoxide (DMSO) is a versatile organosulfur solvent (C₂H₆OS) widely used in biological research as a vehicle control. It serves as the baseline “no perturbation” condition in experiments because it dissolves both polar and nonpolar compounds effectively. DMSO control groups are essential for distinguishing between effects caused by experimental compounds versus the solvent itself. For clarity, we have explained in the revised Figure 1d (Response Figure 4) caption as follows:

..DMSO stands for Dimethyl sulfoxide, which is considered as control group without perturbation...

We also added the definition in section 4.1:

...DMSO refers to cells cultured in Dimethyl sulfoxide, which serves as the control group without perturbation...

I looked over the responses to reviewer 1's comments and most of them seem to be well addressed. The only one that I think has some issues is the response to point 4: 4. The results from G2I (Gene to Image) and I2I (Image to Image) within the MorphDiff framework exhibit discrepancies. Could you provide an explanation? I'm not completely clear on what the reviewer was asking because they weren't specific about what they meant by "discrepancies." I think the authors took the right approach here in trying to explain the differences in performance between the two models. I do think there are a couple of issues in their response, however.

Re:

Thank you for your valuable comments. We have conducted comprehensive experiments to investigate the reasons for these discrepancies. Our detailed analysis has yielded reasonable explanations for these observations, which we present below.

As your concerns and comments in the following three paragraphs are interconnected, we have structured our response to improve clarity. In response to your first comment, we presented our analysis on what caused the observed discrepancies (specifically, the selection of reference control cell morphology images, Response Figure 7a), discussed our results from more robust and comprehensive evaluations (experiments with 10 distinct sets of control cell morphology images for methods rely on cell control morphology images as input, Response Figure 7b), explored the potential factors leading to these discrepancies (particularly the quality of the control cell morphology images, measured by kurtosis—the degree to which the reference set contains outliers, Response Figure 7c), and provided practical recommendations for users when applying the MorphDiff(I2I) mode in practice (Response Figure 7d). This analysis provides a coherent explanation addressing all your comments. In response to your second and third comments, we mainly focused on the specific modifications made to the corresponding sections (Section 2 (Results) and Section 4 (Discussion)). Response Figure 7 is a summary diagram. Detailed point-to-point answers and analysis are given below.

The main source of confusion stems from the trends shown in response figure 1. In the results for the two held out datasets, we see opposite trends between G2I and I2I (i.e., I2I is higher for density and coverage and lower for FID and CMMD on the JUMP dataset, all of which are reversed in the CDRP dataset). While it is expected that the two methods would have differences in performance, it is unexpected that the trends between the two methods would reverse between these two datasets. This reversal suggests that there is something different about the datasets that makes the methods perform differently on them.

Re:

We sincerely appreciate your insightful observation and comments. We acknowledge that the trends between the two modes are not expected to be reversed between these two datasets. To identify the specific differences between the datasets, we carefully investigated the factors contributing to the performance disparity between the G2I mode and the I2I mode of MorphDiff. As shown in Response Figure 7a, these two modes represent different inference approaches for the pre-trained MorphDiff model, both incorporating perturbed L1000 gene expression as input. It is important to highlight that these two approaches are fundamentally the same, with the only distinction being that

Response Figure 7: Overview of our analysis on the performance discrepancies. a. We identified that the use of control morphology images is the key difference between the G2I mode and I2I mode of MorphDiff inference. b. We randomly sampled 10 different DMSO control groups and analyzed the performance variance among these groups for methods rely on control morphology images as input. c. We evaluated the impact of kurtosis—a statistical measure reflecting the likelihood of a distribution to contain outliers—on benchmarking I2I mode performance. d. We provided practical recommendations for users when applying the MorphDiff(I2I) mode in practice. Given a large candidate pool of DMSO images, the following steps can be performed to improve the robustness in I2I mode inference: Filtering outliers in the DMSO reference control morphology images based on their Mahalanobis distance from the distribution center in the DeepProfiler [23] embedding space.

MorphDiff(I2I) also requires a set of control DMSO morphology images as input. In our previous manuscript, we assumed that all control DMSO morphology images perfectly have the same properties and therefore randomly sampled from the complete set of control DMSO images to serve as reference control morphology images during generation. Upon observing the performance inconsistency in our evaluation results, we have recognized that this assumption may be flawed. In fact, as we discussed in the manuscript, the control morphology images could have variation considering various technical factors during the imaging process [23].

To comprehensively investigate the impact of control reference morphology images on model performance, we randomly sampled 10 groups of DMSO images (each group with the same number of images as before) to serve as reference control morphology images and generated outputs using all methods that take the reference control morphology images as input. For comparison, we also established 10 different random seeds to sample images for methods that do not use the reference control morphology images as input. We visualized the results in Response Figure 8(a-b) for both the JUMP OOD set and CDRP OOD set. As shown in the figure, methods that take the control reference morphology images as input (left) exhibit substantially greater variance in performance compared to methods do not use the control reference morphology images (right), demonstrating that changes in reference DMSO images can have a considerably larger effect than alterations in random seed during sampling. Notably, depending on which set of random DMSO samples were used, the performance of MorphDiff(I2I) could either exceed or fall below that of MorphDiff(G2I). Therefore, the performance discrepancies between I2I and G2I reported in the original manuscript can be attributed to bias introduced by the specific DMSO reference images utilized in the I2I mode. This phenomenon is particularly evident in the two modes of MorphDiff, as both modes generally utilize the same pre-trained MorphDiff checkpoint. Consequently, they exhibit very similar performance characteristics and their relative performance is more sensitive to variations induced by differences in the control reference morphology images. To reduce the impact of such variations during benchmarking, we reported the summary results with randomly sampled 10 control morphology image sets for all image-to-

image methods (IMPA [7], DRIT++ [22], DMIT [19], StarGANv1 [18]). The revised results, shown in Response Figure 6a and b, indicate that the previously observed confusing discrepancies are no longer present. Overall, we have established that variation in cell morphology reference images can impact model performance and made considerable efforts in making our benchmarking more robust and reliable.

We also conducted extensive statistical analysis and tests to ensure that our other analysis involving the CellProfiler features and DeepProfiler embeddings is robust to the variation of the reference DMSO control cell morphology images. As shown in Response Figure 8c, we sampled MorphDiff(I2I) and IMPA using 10 different DMSO control cell morphology groups and obtained 10 output groups for each method on the CDRP OOD set. We then extracted the CellProfiler features and DeepProfiler embeddings for each output group. Multiple statistical tests were performed to assess the significance of differences between each output group. Intuitively, if there is no significant difference between the i th output group of IMPA and the j th output group of IMPA, as well as no significant difference between the i th output group of MorphDiff(I2I) and the j th output group of MorphDiff(I2I), but significant differences exist between the i th/ j th output group of IMPA and the j th/ i th output group of MorphDiff(I2I), it means that the change of control cell morphology images will only bring subtle variation within the output of each method, but will not change the difference gap between our method and the baseline method, which can prove the robustness of our analysis using CellProfiler features and DeepProfiler embeddings.

As each CellProfiler feature measures one aspect of the property of the cell morphology images, for each pair of sampled groups on the CDRP OOD set, we performed Wilcoxon signed-rank tests on each CellProfiler feature and calculated the proportion of CellProfiler features with no significant difference (p -value > 0.05). As shown in Response Figure 8d, we demonstrate that almost every IMPA sample group has no significant differences with other IMPA sample groups on almost every feature. The same holds between every MorphDiff(I2I) group. However, each MorphDiff(I2I) sample group has a majority of CellProfiler features that differ significantly from those in the IMPA sample groups, which means that CellProfiler features only have subtle variation within MorphDiff(I2I) sampled outputs, but still exhibit significant differences between the outputs of IMPA and MorphDiff(I2I). We conducted the same analysis but with a different statistical test—the Kolmogorov-Smirnov (K-S) test—as shown in Extended Response Figure 9a. We also used another statistical test, the Friedman test [26], which tests whether multiple samples have the same distribution. We used N groups of IMPA sampled output and $(10 - N)$ groups of MorphDiff(I2I) sampled output and performed the Friedman test on 10 groups of sampled output for each CellProfiler feature. The proportion of CellProfiler features with significant difference is shown in Response Figure 8e. We found that when $N = 0$ (all MorphDiff samples) and $N = 10$ (all IMPA samples), almost every CellProfiler feature does not show significant differences, but in other combinations (mixing samples from both methods), the majority of CellProfiler features have significant differences. All these tests have clearly demonstrated that the change of reference control cell morphology images will not affect our analysis using CellProfiler features, as only subtle variation is found within each method but significant variation exists between methods. We also evaluated the intra-method and inter-method variation of DeepProfiler embeddings on CDRP OOD set. As shown in Response Figure 8f, we performed energy distance tests between the DeepProfiler embeddings of each sampled group. For the Energy Distance test [11], we employed the `dcor.homogeneity.energy` test function from the `dcor` package. The energy test assesses whether multiple groups originate from the same distribution by examining the Euclidean distances between data points. This method compares point-pairwise distances within each group to those between different groups. If points are more closely clustered within their own groups than with points from other groups, this pattern indicates that these groups likely stem from, different underlying distributions. We also used Maximum Mean Discrepancy (MMD) test and calculated the Wasserstein Distance and details are shown in Extended Response Figure 9(b-c). Overall, we can see that our subsequent analysis and conclusions using the DeepProfiler embeddings can hardly be affected by the variation from the control cell morphology images, as this variation will only cause subtle differences within each method but does not affect the gap between methods.

Furthermore, we conducted preliminary experiments to explore how specific properties of control morphology images affect MorphDiff’s I2I mode performance during inference. Like most large-scale generative models, diffusion models are black-box systems whose internal mechanisms are not fully transparent. Therefore, rather than providing theoretical analysis, our approach focuses on empirical observations to identify potential factors inducing performance variation, which aligns with the current state of machine learning research, where full interpretability of such models remains an ongoing challenge in the field.

Concretely, we investigate the impact of kurtosis properties [27] of the reference DMSO control morphology image set on the performance of the sampled output images produced by the I2I mode of MorphDiff. Kurtosis, which quantifies the degree of “tailedness” in a distribution, is potentially relevant in this context because high kurtosis

Response Figure 8: Evaluating the impact of different reference control morphology image sets on the generative evaluation metrics, CellProfiler features and DeepProfiler embeddings. a. Performance comparison on JUMP OOD dataset between methods using DMSO control reference morphology images as input and methods do not use DMSO control reference morphology images as input. The results for 10 runs using different control reference image sets are shown and methods using DMSO control reference morphology images have larger performance variance. b. The same as a, but on the CDRP OOD dataset. c. We sampled both IMPA and MorphDiff(I2I) outputs using 10 different DMSO control cell morphology image groups. Multiple statistical tests were applied to the CellProfiler features and DeepProfiler embeddings of both MorphDiff(I2I) and IMPA outputs to assess the significance of differences between each sampled group. A total of 263 CellProfiler features were evaluated. d. We performed Wilcoxon signed-rank tests between each pair of MorphDiff(I2I)'s and IMPA's generated CellProfiler features on the CDRP OOD dataset. We determine whether there is a significant difference between two groups for a given feature at a significance level of $p\text{-value} = 0.05$ (where $p\text{-value} > 0.05$ indicates no significant difference and $p\text{-value} < 0.05$ indicates a significant difference). The values in the heatmap represent the proportion of features for which there is no significant difference. e. We combined N groups of IMPA output and $(10-N)$ groups of MorphDiff(I2I) output to perform Friedman tests on each CellProfiler feature, where N varied from 0 to 10. The Friedman test was used to assess whether substantial differences existed between multiple groups. f. We performed Energy distance tests between each pair of MorphDiff(I2I)'s and IMPA's generated DeepProfiler embeddings. The values represent the p -values of these tests, with p -values smaller than 0.05 indicating significant differences.

suggests a higher likelihood of extreme outliers in the control morphology image data. We hypothesize that the presence of outliers in the control reference morphology images might influence generative model performance, suggesting kurtosis could serve as a potential metric for assessing dataset characteristics. We sampled 100 sets of control morphology images from the whole pool of control DMSO morphology images and calculated the kurtosis of DeepProfiler embeddings for each set in a multivariate manner [27]. We then sampled MorphDiff with the I2I mode using each set of control DMSO morphology images and assessed the generative performance. Through this approach, we explored the possible relationship between the kurtosis of the reference control DMSO morphology image sets and their resulting generative performance.

The results on the JUMP OOD dataset and the CDRP OOD dataset are shown in Response Figure 9a and b. We observed that the kurtosis of the reference DMSO morphology image set (x-axis) has a negative correlation with the performance metrics of the generated output (y-axis) on both datasets. This supports our hypothesis that higher kurtosis is associated with a higher likelihood of outliers, which may contribute to degraded model performance in the generated outputs. Importantly, we also highlighted the kurtosis of the previously used reference DMSO morphology image set for the I2I mode of MorphDiff sampling (green dot/line) as well as their G2I counterpart. This comparison explains the performance discrepancies between MorphDiff(G2I) and MorphDiff(I2I) on the two datasets as shown in the previous manuscript. Our analysis revealed that the kurtosis of the reference DMSO images can be one of the potential factors causing these discrepancies.

We also provided practical tips for applying the I2I mode of the MorphDiff model to enhance the method’s robustness. As revealed in Response Figure 9(a-b), higher kurtosis in control reference morphology images suggests a greater propensity for outliers, which may negatively impact generative model performance. Therefore, as demonstrated in Response Figure 7d, filtering outlier samples from the control reference cell morphology set could potentially improve the robustness of results. To provide quantitative validation of this approach, we conducted the following analysis as illustrated in Response Figure 9c: For a given control reference cell morphology set, we used Mahalanobis distance (the distance used to compute kurtosis [27]) in the DeepProfiler embedding space to measure the distance from distribution center to each sample, where closer distance indicates samples less likely to be outliers [28]. We then compared the performance between two scenarios: (1) retaining only the closest $X\%$ of samples to the center (those least likely to be outliers) and (2) retaining only the furthest $X\%$ of samples from the center (those most likely to be outliers). As shown in Response Figure 9d, on the JUMP OOD dataset, using the closest $X\%$ of samples led to performance improvements compared to using the furthest $X\%$ of samples. The improvements were particularly notable in FID(-1) and Inception scores, demonstrating that filtering outliers based on Mahalanobis distance from the center before applying MorphDiff (I2I) can effectively enhance the robustness of the sampling results.

We added the following discussion in Supplementary Notes 4:

Analysis of the impact of the quality of the control morphology images on I2I mode performance as well as practical tips

Furthermore, we conducted preliminary experiments to explore how specific properties of control morphology images affect MorphDiff’s I2I mode performance during inference. Like most large-scale generative models, diffusion architectures operate as complex systems whose internal mechanisms are not fully transparent. Rather than attempting theoretical analysis, our approach will focus on empirical observations to identify potential factors influencing performance variation. This practical investigation aligns with the current state of machine learning research, where complete interpretability of such models remains an ongoing challenge in the field.

Concretely, we investigate the impact of kurtosis properties [27] of the reference DMSO control morphology image set on the performance of the sampled output images produced by the I2I mode of MorphDiff. Kurtosis, which quantifies the degree of “tailedness” in a distribution, is potentially relevant in this context because high kurtosis suggests a higher likelihood of extreme outliers in the control morphology image data. We hypothesize that the presence of outliers in the control reference morphology images might influence generative model performance, suggesting kurtosis could serve as a potential metric for assessing dataset characteristics. We sampled 100 sets of control morphology

Response Figure 9: a. b. Analysis of the potential impact of the kurtosis of the DMSO control morphology images on the performance of the I2I mode of MorphDiff. For each point (blue cross), the x-axis represents the kurtosis of a reference set of DMSO control morphology images, and the y-axis indicates the corresponding performance metrics of the inference output using this set of reference images (I2I mode). The orange line indicates the performance of the G2I mode of MorphDiff, and the green dot/line indicates the kurtosis and performance of the DMSO reference images in the previous manuscript. We conducted the same analysis for the JUMP OOD dataset (a) and CDRP OOD dataset (b), the results are linearly normalized. c. Illustration of our approach to improving the robustness of MorphDiff’s I2I mode. We quantified the Mahalanobis distance between each DMSO sample and the center of the DMSO reference image set. We then compared the generative performance of MorphDiff(I2I) when using two different subsets: (1) the closest to center $X\%$ samples (least likely to be outliers) versus (2) the furthest to center $X\%$ samples (most likely to be outliers). d. Performance Illustration of the proposed filtering approach. y-axis means performance improvement by calculating the performance difference between the sampling output of the Closest $X\%$ and the Furthest $X\%$.

images from the whole pool of control DMSO morphology images and calculated the kurtosis of DeepProfiler embeddings for each set in a multivariate manner [27]. We then sampled MorphDiff with the I2I mode using each set of control DMSO morphology images and assessed the generative performance. Through this approach, we explored the possible relationship between the kurtosis of the reference control DMSO morphology image sets and their resulting generative performance.

We assume that higher kurtosis is associated with a higher likelihood of outliers, which may contribute to degraded model performance in the generated outputs. The results on the JUMP OOD dataset and the CDRP OOD dataset are shown in Appendix Figure 30a and b. We observed that the kurtosis of the reference DMSO morphology image set (x-axis) has a negative correlation with the performance metrics of the generated output (y-axis) on both datasets. This supports Our analysis revealed that the kurtosis of the reference DMSO images can be one of the potential factors causing these discrepancies.

We also provided practical tips for applying the I2I mode of the MorphDiff model to enhance the

method’s robustness. As revealed in Appendix Figure 30 a and b, higher kurtosis in control reference morphology images suggests a greater propensity for outliers, which may negatively impact generative model performance. Therefore, filtering outlier samples from the control reference cell morphology set could potentially improve the robustness of results. To provide quantitative validation of this approach, we conducted the following analysis as illustrated in Appendix Figure 30c: For a given control reference cell morphology set, we used Mahalanobis distance (the distance used to compute kurtosis [27]) to measure the distance from distribution center to each sample, where closer distance indicates samples less likely to be outliers [28]. We then compared the performance between two scenarios: (1) retaining only the closest $X\%$ of samples to the center (those least likely to be outliers) and (2) retaining only the furthest $X\%$ of samples from the center (those most likely to be outliers). As shown in Appendix Figure 30d, on the JUMP OOD dataset, using the closest $X\%$ of samples led to performance improvements compared to using the furthest $X\%$ of samples. The improvements were particularly notable in FID(-1) and Inception scores, demonstrating that filtering outliers based on Mahalanobis distance from the center before applying MorphDiff (I2I) can effectively enhance the robustness of the sampling results as shown in Appendix Figure 30e.

In the authors’ revision, they include a discussion of the differences in performance between these methods on the two datasets in sections 2.2 and 2.3, respectively. These descriptions are independent and don’t acknowledge that they are complete opposites of each other. For example, in section 2.2 they write: “MorphDiff(I2I) performs better on the metrics of density and coverage, suggesting that it generates a more diverse set of samples, and the generated distribution better aligns with the ground truth.” while in section 2.3 they write: “MorphDiff(G2I) outperforms on density, and coverage.. [meaning it] excels in generating samples of higher diversity and closer to the ground-truth distribution.” Presenting the results like this is confusing because it seems to attribute opposite strengths to the two methods depending on the dataset. This discrepancy makes it difficult to connect experimental results to the designs of the underlying models.

Re:

Thank you for your comments. As we discussed in your first sub-question and Response Figure 6 shows, after using multiple control cell morphology image sample sets for benchmarking I2I methods to ensure robust evaluation, we found that the previously observed discrepancies between the two modes of MorphDiff are no longer present, although the I2I mode shows larger variance than the G2I mode. Consequently, we removed these descriptions and added new analyses in Sections 2.2 and 2.3, respectively.

In section 2.2, we made the following modifications:

To ensure statistical reliability and robustness of our results, we employed different sampling approaches based on method requirements. For methods requiring reference control DMSO morphology images as input (see Appendix Table 2), we generated the output using 10 randomly sampled, distinct groups of control DMSO morphology images. For methods that do not rely on reference control DMSO images, we conducted 10 separate sampling iterations, each with a different random seed. We performed the Wilcoxon signed-rank test for each comparison with Bonferroni correction to further enhance the statistical significance of our tests. It should be noted that star (“*”) means that MorphDiff surpasses the baseline method with a p -value less than 0.05. The ‘ns’ represents the situation of no significant difference. The same notation for performance tests is used in the following sections. As depicted in Figure 2b, MorphDiff (I2I) and MorphDiff (G2I) rank as the top two methods across these evaluation metrics compared with other baseline approaches, highlighting the effectiveness of our model. We also observed that all methods relying on reference control DMSO morphology images exhibit larger performance variance compared to those that do not, as the quality of the control cell morphology images may impact the model performance in terms of general evaluation metrics (See Supplementary Notes 4). The results also show that MorphNet achieves a nearly zero coverage score, demonstrating that the generative output of MorphNet lacks diversity, which aligns with the visualization in Figure 2a. Based on both qualitative visualizations and quantitative metrics, MorphDiff (both modes) demonstrates more superior performance in terms of generalization, fidelity, and diversity compared to other generative models, ultimately resulting in superior generation quality.

In section 2.3, we made the following modifications:

We first benchmarked all methods on the CDRP OOD set with the general evaluation metrics, in which the meanings of stars are the same as in Section 2.2 and the hashtag (#) indicates the situation that the baseline performs better significantly. As shown in Figure 3a, both modes of MorphDiff substantially outperform other methods in the most cases. While DMIT and MDTv2 achieve high scores in IS, density, and coverage, they obtain relatively lower scores in FID(-1) and CMMD(-1) compared to MorphDiff (both modes). This suggests that DMIT and MDTv2 struggle to produce high-quality samples. Similar to the results on the JUMP dataset, MorphNet performs well in terms of density score but is nearly ineffective in terms of coverage, demonstrating its lack of generalizability and diversity. Ideally, an effective generative model should achieve low FID, CMMD, and high scores in IS, density, and coverage to ensure that it produces outputs of high quality and fidelity, while also generating diverse outputs that closely align with the ground-truth distribution. MorphDiff (both modes) outperforms baseline methods considering these criteria, demonstrating its comprehensive generative capability and stability.

In the discussion section, the authors acknowledge the reversal in trends, but do not investigate it further. Instead, they speculate on factors that could have caused it: “One reason for this discrepancy may be the quality of the control morphology images used as input in the I2I mode. Although the control cell morphology provides valuable information about cellular components, it can also be affected by batch effects and technical variations in Cell Painting experiments.” It would be much more valuable to actually look at the quality of the images and see if there are differences between those provided by JUMP and CDRP. The authors could also provide specific examples to show where the inclusion of an input image had opposite effects for the two datasets.

Re:

Thank you for your comments. As we discussed in the first sub-question, we have carefully examined how the control morphology images used as input in the I2I mode can impact final model performance. We have studied the impact of quality (particularly kurtosis) of the reference control cell morphology images on generated output performance. The details of these results were presented in the first sub-question. We have made the following modifications in the revised manuscript:

MorphDiff operates in two distinct modes: I2I and G2I. The G2I mode utilizes gene expression data exclusively as input, whereas the I2I mode incorporates both gene expression data and control cell morphology images. We have observed that the I2I mode exhibits greater variance in performance compared to the G2I mode. This variability can be attributed primarily to the quality and distribution characteristics of the selected control cell morphology sets used as input for the I2I mode. Our analysis indicates that the quality of the control cell morphology may potentially influence the quality of the generated output. Detailed analysis and tips for filtering DMSO control images can be referred to the Supplementary Notes 4 and Appendix Figure 30. Both modes of MorphDiff satisfy the criteria for effective generative modeling, achieving low FID, CMMD, and high scores in Inception Score, density, and coverage metrics. For most application scenarios, either mode is appropriate. However, when users specifically aim to analyze morphological feature changes relative to certain control cells—as discussed in Figure 3c and Figure 4b—MorphDiff (I2I) represents the more suitable option. Nevertheless, users should carefully examine the quality and distribution of selected control morphology images, as these factors can potentially impact the generative performance.

References

- [1] Subramanian, A. et al. A next generation connectivity map: L1000 platform and the first 1,000,000 profiles. Cell **171**, 1437–1452 (2017).
- [2] Chandrasekaran, S. N. et al. Jump cell painting dataset: morphological impact of 136,000 chemical and genetic perturbations. bioRxiv 2023–03 (2023).
- [3] Bray, M.-A. et al. A dataset of images and morphological profiles of 30 000 small-molecule treatments using the cell painting assay. Gigascience **6**, giw014 (2017).

- [4] Haghghi, M., Caicedo, J. C., Cimini, B. A., Carpenter, A. E. & Singh, S. High-dimensional gene expression and morphology profiles of cells across 28,000 genetic and chemical perturbations. *Nature methods* **19**, 1550–1557 (2022).
- [5] Natoli, T. et al. broadinstitute/lincs-cell-painting: Full release of lincs cell painting dataset (2021).
- [6] Ben-David, S. et al. A theory of learning from different domains. *Machine learning* **79**, 151–175 (2010).
- [7] Palma, A., Theis, F. J. & Lotfollahi, M. Predicting cell morphological responses to perturbations using generative modeling. *Nature Communications* **16**, 505 (2025).
- [8] Bunne, C. et al. Learning single-cell perturbation responses using neural optimal transport. *Nature methods* **20**, 1759–1768 (2023).
- [9] Naaman, M. On the tight constant in the multivariate dvoretzky–kiefer–wolfowitz inequality. *Statistics & Probability Letters* (2021).
- [10] Gretton, A., Borgwardt, K. M., Rasch, M. J., Schölkopf, B. & Smola, A. A kernel two-sample test. *The Journal of Machine Learning Research* **13**, 723–773 (2012).
- [11] Székely, G. J., Rizzo, M. L. & Bakirov, N. K. Measuring and testing dependence by correlation of distances (2007).
- [12] Carpenter, A. E. et al. Cellprofiler: image analysis software for identifying and quantifying cell phenotypes. *Genome biology* **7**, 1–11 (2006).
- [13] Moshkov, N. et al. Learning representations for image-based profiling of perturbations. *Nature Communications* **15**, 1594 (2024).
- [14] Meng, C. et al. Sdedit: Guided image synthesis and editing with stochastic differential equations. In *International Conference on Learning Representations* (2022).
- [15] Wang, Z., Bovik, A. C., Sheikh, H. R. & Simoncelli, E. P. Image quality assessment: from error visibility to structural similarity. *IEEE transactions on image processing* **13**, 600–612 (2004).
- [16] Bray, M.-A. et al. Cell painting, a high-content image-based assay for morphological profiling using multiplexed fluorescent dyes. *Nature protocols* **11**, 1757–1774 (2016).
- [17] Lee, H. & Welch, J. D. Morphnet predicts cell morphology from single-cell gene expression. *bioRxiv* 2022–10 (2022).
- [18] Choi, Y. et al. Stargan: Unified generative adversarial networks for multi-domain image-to-image translation. In *Proceedings of the IEEE conference on computer vision and pattern recognition*, 8789–8797 (2018).
- [19] Yu, X., Chen, Y., Liu, S., Li, T. & Li, G. Multi-mapping image-to-image translation via learning disentanglement. *Advances in Neural Information Processing Systems* **32** (2019).
- [20] Esser, P., Rombach, R. & Ommer, B. Taming transformers for high-resolution image synthesis. In *Proceedings of the IEEE/CVF conference on computer vision and pattern recognition*, 12873–12883 (2021).
- [21] Gao, S., Zhou, P., Cheng, M.-M. & Yan, S. Masked diffusion transformer is a strong image synthesizer. In *Proceedings of the IEEE/CVF International Conference on Computer Vision*, 23164–23173 (2023).
- [22] Lee, H.-Y. et al. Drit++: Diverse image-to-image translation via disentangled representations. *International Journal of Computer Vision* **128**, 2402–2417 (2020).
- [23] Moshkov, N. et al. Learning representations for image-based profiling of perturbations. *Nature communications* **15**, 1594 (2024).
- [24] Batool, M., Ahmad, B. & Choi, S. A structure-based drug discovery paradigm. *International journal of molecular sciences* **20**, 2783 (2019).
- [25] Lotfollahi, M. et al. Predicting cellular responses to complex perturbations in high-throughput screens. *Molecular systems biology* **19**, e11517 (2023).

- [26] Demšar, J. Statistical comparisons of classifiers over multiple data sets. Journal of Machine learning research **7**, 1–30 (2006).
- [27] Korkmaz, S., Gökşülük, D. & Zararsiz, G. Mvn: An r package for assessing multivariate normality. R journal **6** (2014).
- [28] DeCarlo, L. T. On the meaning and use of kurtosis. Psychological methods **2**, 292 (1997).

Extended Response Tables

Extended Response Table 1: Detailed results of the Maximum Mean Discrepancy (MMD) test on the CDRP dataset (Repeat 10 times).

Dataset	Rounds	Statistics	p -value	Dataset	Rounds	Statistics	p -value
ID set		-2.62E-05	0.714	ID set		3.92E-05	0.164
OOD set	0	1.88E-03	0.001	OOD set	5	2.05E-03	0.001
Target_MOA set		7.92E-02	0.001	Target_MOA set		7.79E-02	0.001
ID set		5.28E-05	0.103	ID set		-4.06E-05	0.851
OOD set	1	1.94E-03	0.001	OOD set	6	2.26E-03	0.001
Target_MOA set		7.85E-02	0.001	Target_MOA set		8.18E-02	0.001
ID set		-1.15E-05	0.550	ID set		2.32E-05	0.251
OOD set	2	1.99E-03	0.001	OOD set	7	2.44E-03	0.001
Target_MOA set		7.86E-02	0.001	Target_MOA set		8.12E-02	0.001
ID set		-2.58E-05	0.713	ID set		-9.71E-06	0.509
OOD set	3	1.99E-03	0.001	OOD set	8	1.54E-03	0.001
Target_MOA set		7.83E-02	0.001	Target_MOA set		7.75E-02	0.001
ID set		-4.59E-06	0.463	ID set		-3.44E-05	0.803
OOD set	4	1.98E-03	0.001	OOD set	9	1.88E-03	0.001
Target_MOA set		7.78E-02	0.001	Target_MOA set		7.99E-02	0.001

Extended Response Table 2: Detailed results of the Energy Distance test on the CDRP dataset (Repeat 10 times).

Dataset	Rounds	Statistics	p -value	Dataset	Rounds	Statistics	p -value
ID set		9.815	0.726	ID set		12.412	0.163
OOD set	0	82.562	0.001	OOD set	5	88.560	0.001
Target_MOA set		2808.427	0.001	Target_MOA set		2761.942	0.001
ID set		12.983	0.114	ID set		9.501	0.829
OOD set	1	83.840	0.001	OOD set	6	96.208	0.001
Target_MOA set		2773.373	0.001	Target_MOA set		2895.760	0.001
ID set		10.608	0.553	ID set		11.916	0.253
OOD set	2	86.011	0.001	OOD set	7	102.112	0.001
Target_MOA set		2776.654	0.001	Target_MOA set		2867.429	0.001
ID set		10.218	0.608	ID set		10.627	0.518
OOD set	3	85.052	0.001	OOD set	8	69.067	0.001
Target_MOA set		2773.926	0.001	Target_MOA set		2742.465	0.001
ID set		10.790	0.481	ID set		9.620	0.817
OOD set	4	84.535	0.001	OOD set	9	81.002	0.001
Target_MOA set		2750.999	0.001	Target_MOA set		2828.478	0.001

Extended Response Table 3: Detailed results of the Maximum Mean Discrepancy (MMD) test on the JUMP dataset (Repeat 10 times).

Dataset	Rounds	Statistics	p -value	Dataset	Rounds	Statistics	p -value
ID set	0	1.61E-05	0.308	ID set	5	-2.25E-06	0.425
OOD set		1.14E-02	0.001	OOD set		9.91E-03	0.001
ID set	1	-1.62E-05	0.568	ID set	6	-4.22E-05	0.816
OOD set		9.77E-03	0.001	OOD set		1.01E-02	0.001
ID set	2	-5.82E-05	0.941	ID set	7	4.48E-05	0.154
OOD set		1.06E-02	0.001	OOD set		1.07E-02	0.001
ID set	3	5.34E-05	0.129	ID set	8	-2.28E-05	0.638
OOD set		1.08E-02	0.001	OOD set		1.07E-02	0.001
ID set	4	1.75E-05	0.300	ID set	9	-8.38E-06	0.507
OOD set		1.03E-02	0.001	OOD set		1.03E-02	0.001

Extended Response Table 4: Detailed results of the Energy Distance test on the JUMP dataset (Repeat 10 times).

Dataset	Rounds	Statistics	p -value	Dataset	Rounds	Statistics	p -value
ID set	0	9.815	0.726	ID set	5	12.412	0.163
OOD set		82.562	0.001	OOD set		88.560	0.001
ID set	1	12.983	0.114	ID set	6	9.501	0.829
OOD set		83.840	0.001	OOD set		96.208	0.001
ID set	2	10.608	0.553	ID set	7	11.916	0.253
OOD set		86.011	0.001	OOD set		102.112	0.001
ID set	3	10.218	0.608	ID set	8	10.627	0.518
OOD set		85.052	0.001	OOD set		69.067	0.001
ID set	4	10.790	0.481	ID set	9	9.620	0.817
OOD set		84.535	0.001	OOD set		81.002	0.001

Extended Response Table 5: Detailed results of the Maximum Mean Discrepancy (MMD) on the LINCS Target leave-one-out set (Repeat 10 times).

Dataset	Rounds	Statistics	p -value	Dataset	Rounds	Statistics	p -value
CTSK	0	0.021	0.001	CTSK	5	0.033	0.001
ERBB2		0.021	0.001	ERBB2		0.035	0.001
F10		0.021	0.001	F10		0.034	0.001
MELK		0.022	0.001	MELK		0.035	0.001
MLNR		0.021	0.001	MLNR		0.034	0.001
PREP		0.021	0.001	PREP		0.035	0.001
PTGS2		0.022	0.001	PTGS2		0.035	0.001
SRGAP1		0.020	0.001	SRGAP1		0.034	0.001
TOP2A		0.021	0.001	TOP2A		0.033	0.001
TRPA1		0.020	0.001	TRPA1		0.035	0.001
CTSK	1	0.009	0.001	CTSK	6	0.004	0.001
ERBB2		0.010	0.001	ERBB2		0.004	0.001
F10		0.010	0.001	F10		0.004	0.001
MELK		0.011	0.001	MELK		0.004	0.001
MLNR		0.010	0.001	MLNR		0.004	0.001
PREP		0.010	0.001	PREP		0.004	0.001
PTGS2		0.010	0.001	PTGS2		0.004	0.001
SRGAP1		0.010	0.001	SRGAP1		0.004	0.001
TOP2A		0.010	0.001	TOP2A		0.004	0.001
TRPA1		0.010	0.001	TRPA1		0.004	0.001
CTSK	2	0.016	0.001	CTSK	7	0.006	0.001
ERBB2		0.018	0.001	ERBB2		0.006	0.001
F10		0.017	0.001	F10		0.006	0.001
MELK		0.018	0.001	MELK		0.006	0.001
MLNR		0.017	0.001	MLNR		0.006	0.001
PREP		0.018	0.001	PREP		0.006	0.001
PTGS2		0.017	0.001	PTGS2		0.006	0.001
SRGAP1		0.018	0.001	SRGAP1		0.006	0.001
TOP2A		0.017	0.001	TOP2A		0.006	0.001
TRPA1		0.017	0.001	TRPA1		0.006	0.001
CTSK	3	0.004	0.001	CTSK	8	0.008	0.001
ERBB2		0.004	0.001	ERBB2		0.009	0.001
F10		0.004	0.001	F10		0.008	0.001
MELK		0.004	0.001	MELK		0.008	0.001
MLNR		0.005	0.001	MLNR		0.008	0.001
PREP		0.004	0.001	PREP		0.008	0.001
PTGS2		0.005	0.001	PTGS2		0.009	0.001
SRGAP1		0.004	0.001	SRGAP1		0.009	0.001
TOP2A		0.004	0.001	TOP2A		0.009	0.001
TRPA1		0.004	0.001	TRPA1		0.009	0.001
CTSK	4	0.005	0.001	CTSK	9	0.027	0.001
ERBB2		0.004	0.001	ERBB2		0.029	0.001
F10		0.005	0.001	F10		0.029	0.001
MELK		0.005	0.001	MELK		0.029	0.001
MLNR		0.005	0.001	MLNR		0.028	0.001
PREP		0.005	0.001	PREP		0.027	0.001
PTGS2		0.005	0.001	PTGS2		0.027	0.001
SRGAP1		0.004	0.001	SRGAP1		0.027	0.001
TOP2A		0.005	0.001	TOP2A		0.027	0.001
TRPA1		0.004	0.001	TRPA1		0.029	0.001

Extended Response Table 6: Detailed results of the Energy Distance Test on the LINC5 Target leave-one-out set (Repeat 10 times).

Dataset	Rounds	Statistics	p -value	Dataset	Rounds	Statistics	p -value
CTSK		774.683	0.001	CTSK		1271.801	0.001
ERBB2		791.469	0.001	ERBB2		1322.075	0.001
F10		799.256	0.001	F10		1301.862	0.001
MELK		834.459	0.001	MELK		1346.745	0.001
MLNR		790.459	0.001	MLNR		1290.705	0.001
PREP	0	779.179	0.001	PREP	5	1328.801	0.001
PTGS2		821.662	0.001	PTGS2		1318.103	0.001
SRGAP1		765.466	0.001	SRGAP1		1304.944	0.001
TOP2A		779.625	0.001	TOP2A		1275.148	0.001
TRPA1		763.520	0.001	TRPA1		1325.110	0.001
CTSK		370.440	0.001	CTSK		164.550	0.001
ERBB2		407.110	0.001	ERBB2		168.736	0.001
F10		379.061	0.001	F10		169.201	0.001
MELK		419.972	0.001	MELK		151.882	0.001
MLNR		394.402	0.001	MLNR		173.193	0.001
PREP	1	379.386	0.001	PREP	6	176.399	0.001
PTGS2		404.070	0.001	PTGS2		152.814	0.001
SRGAP1		380.579	0.001	SRGAP1		176.289	0.001
TOP2A		377.674	0.001	TOP2A		163.058	0.001
TRPA1		400.459	0.001	TRPA1		169.827	0.001
CTSK		630.308	0.001	CTSK		266.772	0.001
ERBB2		670.262	0.001	ERBB2		246.107	0.001
F10		640.987	0.001	F10		266.530	0.001
MELK		674.095	0.001	MELK		255.561	0.001
MLNR		639.816	0.001	MLNR		248.198	0.001
PREP	2	674.203	0.001	PREP	7	242.636	0.001
PTGS2		664.330	0.001	PTGS2		249.065	0.001
SRGAP1		684.513	0.001	SRGAP1		245.500	0.001
TOP2A		660.435	0.001	TOP2A		247.638	0.001
TRPA1		657.089	0.001	TRPA1		246.702	0.001
CTSK		183.508	0.001	CTSK		312.442	0.001
ERBB2		177.683	0.001	ERBB2		339.042	0.001
F10		180.307	0.001	F10		315.281	0.001
MELK		182.482	0.001	MELK		309.171	0.001
MLNR		192.208	0.001	MLNR		328.685	0.001
PREP	3	166.677	0.001	PREP	8	323.465	0.001
PTGS2		186.878	0.001	PTGS2		348.552	0.001
SRGAP1		162.949	0.001	SRGAP1		338.639	0.001
TOP2A		173.872	0.001	TOP2A		340.584	0.001
TRPA1		173.300	0.001	TRPA1		331.113	0.001
CTSK		191.854	0.001	CTSK		1069.068	0.001
ERBB2		181.950	0.001	ERBB2		1132.253	0.001
F10		215.084	0.001	F10		1119.731	0.001
MELK		198.643	0.001	MELK		1109.266	0.001
MLNR		205.765	0.001	MLNR		1087.651	0.001
PREP	4	194.802	0.001	PREP	9	1069.201	0.001
PTGS2		197.199	0.001	PTGS2		1053.327	0.001
SRGAP1		183.812	0.001	SRGAP1		1047.547	0.001
TOP2A		187.783	0.001	TOP2A		1072.184	0.001
TRPA1		179.384	0.001	TRPA1		1129.584	0.001

Extended Response Table 7: Detailed results of the Maximum Mean Discrepancy on the LINCS MOA leave-one-out set (Repeat 10 times).

Dataset	Rounds	Statistics	p -value	Dataset	Rounds	Statistics	p -value
EGFR inhibitor	0	0.004	0.001	EGFR inhibitor	5	0.001	0.001
HMGCR inhibitor		0.004	0.001	HMGCR inhibitor		0.001	0.001
NFkB pathway inhibitor		0.004	0.001	NFkB pathway inhibitor		0.001	0.001
adrenergic receptor agonist		0.004	0.001	adrenergic receptor agonist		0.001	0.001
adrenergic receptor inhibitor		0.003	0.001	adrenergic receptor inhibitor		0.001	0.001
calcium channel blocker		0.004	0.001	calcium channel blocker		0.001	0.001
dopamine receptor agonist		0.004	0.001	dopamine receptor agonist		0.001	0.001
dopamine receptor inhibitor		0.004	0.001	dopamine receptor inhibitor		0.001	0.001
phosphodiesterase inhibitor		0.004	0.001	phosphodiesterase inhibitor		0.001	0.001
serotonin receptor antagonist		0.004	0.001	serotonin receptor antagonist		0.001	0.001
EGFR inhibitor	1	0.010	0.001	EGFR inhibitor	6	0.018	0.001
HMGCR inhibitor		0.009	0.001	HMGCR inhibitor		0.018	0.001
NFkB pathway inhibitor		0.010	0.001	NFkB pathway inhibitor		0.018	0.001
adrenergic receptor agonist		0.009	0.001	adrenergic receptor agonist		0.018	0.001
adrenergic receptor inhibitor		0.010	0.001	adrenergic receptor inhibitor		0.019	0.001
calcium channel blocker		0.009	0.001	calcium channel blocker		0.018	0.001
dopamine receptor agonist		0.009	0.001	dopamine receptor agonist		0.018	0.001
dopamine receptor inhibitor		0.010	0.001	dopamine receptor inhibitor		0.017	0.001
phosphodiesterase inhibitor		0.009	0.001	phosphodiesterase inhibitor		0.018	0.001
serotonin receptor antagonist		0.010	0.001	serotonin receptor antagonist		0.019	0.001
EGFR inhibitor	2	0.002	0.001	EGFR inhibitor	7	0.002	0.001
HMGCR inhibitor		0.003	0.001	HMGCR inhibitor		0.003	0.001
NFkB pathway inhibitor		0.003	0.001	NFkB pathway inhibitor		0.002	0.001
adrenergic receptor agonist		0.002	0.001	adrenergic receptor agonist		0.002	0.001
adrenergic receptor inhibitor		0.003	0.001	adrenergic receptor inhibitor		0.003	0.001
calcium channel blocker		0.003	0.001	calcium channel blocker		0.003	0.001
dopamine receptor agonist		0.003	0.001	dopamine receptor agonist		0.002	0.001
dopamine receptor inhibitor		0.002	0.001	dopamine receptor inhibitor		0.003	0.001
phosphodiesterase inhibitor		0.003	0.001	phosphodiesterase inhibitor		0.003	0.001
serotonin receptor antagonist		0.002	0.001	serotonin receptor antagonist		0.002	0.001
EGFR inhibitor	3	0.012	0.001	EGFR inhibitor	8	0.002	0.001
HMGCR inhibitor		0.012	0.001	HMGCR inhibitor		0.002	0.001
NFkB pathway inhibitor		0.012	0.001	NFkB pathway inhibitor		0.002	0.001
adrenergic receptor agonist		0.013	0.001	adrenergic receptor agonist		0.002	0.001
adrenergic receptor inhibitor		0.012	0.001	adrenergic receptor inhibitor		0.002	0.001
calcium channel blocker		0.011	0.001	calcium channel blocker		0.002	0.001
dopamine receptor agonist		0.012	0.001	dopamine receptor agonist		0.002	0.001
dopamine receptor inhibitor		0.012	0.001	dopamine receptor inhibitor		0.002	0.001
phosphodiesterase inhibitor		0.012	0.001	phosphodiesterase inhibitor		0.002	0.001
serotonin receptor antagonist		0.012	0.001	serotonin receptor antagonist		0.002	0.001
EGFR inhibitor	4	0.016	0.001	EGFR inhibitor	9	0.005	0.001
HMGCR inhibitor		0.016	0.001	HMGCR inhibitor		0.005	0.001
NFkB pathway inhibitor		0.017	0.001	NFkB pathway inhibitor		0.006	0.001
adrenergic receptor agonist		0.015	0.001	adrenergic receptor agonist		0.005	0.001
adrenergic receptor inhibitor		0.015	0.001	adrenergic receptor inhibitor		0.005	0.001
calcium channel blocker		0.016	0.001	calcium channel blocker		0.006	0.001
dopamine receptor agonist		0.016	0.001	dopamine receptor agonist		0.005	0.001
dopamine receptor inhibitor		0.016	0.001	dopamine receptor inhibitor		0.005	0.001
phosphodiesterase inhibitor		0.016	0.001	phosphodiesterase inhibitor		0.005	0.001
serotonin receptor antagonist		0.016	0.001	serotonin receptor antagonist		0.006	0.001

Extended Response Table 8: Detailed results of the Energy Distance Test on the LINCS MOA leave-one-out set (Repeat 10 times).

Dataset	Rounds	Statistics	p -value	Dataset	Rounds	Statistics	p -value
EGFR inhibitor	0	162.594	0.001	EGFR inhibitor	5	35.816	0.001
HMGCR inhibitor		154.919	0.001	HMGCR inhibitor		37.866	0.001
NFkB pathway inhibitor		147.968	0.001	NFkB pathway inhibitor		52.126	0.001
adrenergic receptor agonist		159.153	0.001	adrenergic receptor agonist		53.919	0.001
adrenergic receptor inhibitor		138.315	0.001	adrenergic receptor inhibitor		51.239	0.001
calcium channel blocker		161.838	0.001	calcium channel blocker		40.528	0.001
dopamine receptor agonist		146.804	0.001	dopamine receptor agonist		45.129	0.001
dopamine receptor inhibitor		147.282	0.001	dopamine receptor inhibitor		46.806	0.001
phosphodiesterase inhibitor		161.655	0.001	phosphodiesterase inhibitor		45.250	0.001
serotonin receptor antagonist		145.992	0.001	serotonin receptor antagonist		42.078	0.001
EGFR inhibitor	1	364.421	0.001	EGFR inhibitor	6	679.458	0.001
HMGCR inhibitor		362.582	0.001	HMGCR inhibitor		687.616	0.001
NFkB pathway inhibitor		366.814	0.001	NFkB pathway inhibitor		702.750	0.001
adrenergic receptor agonist		363.883	0.001	adrenergic receptor agonist		683.194	0.001
adrenergic receptor inhibitor		383.216	0.001	adrenergic receptor inhibitor		716.722	0.001
calcium channel blocker		361.565	0.001	calcium channel blocker		703.348	0.001
dopamine receptor agonist		350.055	0.001	dopamine receptor agonist		698.971	0.001
dopamine receptor inhibitor		371.723	0.001	dopamine receptor inhibitor		662.825	0.001
phosphodiesterase inhibitor		349.038	0.001	phosphodiesterase inhibitor		674.459	0.001
serotonin receptor antagonist		379.420	0.001	serotonin receptor antagonist		708.494	0.001
EGFR inhibitor	2	106.280	0.001	EGFR inhibitor	7	98.334	0.001
HMGCR inhibitor		119.521	0.001	HMGCR inhibitor		113.771	0.001
NFkB pathway inhibitor		113.592	0.001	NFkB pathway inhibitor		92.020	0.001
adrenergic receptor agonist		109.492	0.001	adrenergic receptor agonist		105.139	0.001
adrenergic receptor inhibitor		119.698	0.001	adrenergic receptor inhibitor		110.331	0.001
calcium channel blocker		123.264	0.001	calcium channel blocker		112.086	0.001
dopamine receptor agonist		130.497	0.001	dopamine receptor agonist		105.902	0.001
dopamine receptor inhibitor		106.897	0.001	dopamine receptor inhibitor		110.970	0.001
phosphodiesterase inhibitor		116.367	0.001	phosphodiesterase inhibitor		107.899	0.001
serotonin receptor antagonist		103.285	0.001	serotonin receptor antagonist		97.183	0.001
EGFR inhibitor	3	467.787	0.001	EGFR inhibitor	8	76.362	0.001
HMGCR inhibitor		470.290	0.001	HMGCR inhibitor		86.119	0.001
NFkB pathway inhibitor		460.696	0.001	NFkB pathway inhibitor		77.671	0.001
adrenergic receptor agonist		491.312	0.001	adrenergic receptor agonist		74.922	0.001
adrenergic receptor inhibitor		466.398	0.001	adrenergic receptor inhibitor		81.949	0.001
calcium channel blocker		448.481	0.001	calcium channel blocker		74.335	0.001
dopamine receptor agonist		479.226	0.001	dopamine receptor agonist		77.346	0.001
dopamine receptor inhibitor		476.312	0.001	dopamine receptor inhibitor		78.339	0.001
phosphodiesterase inhibitor		462.820	0.001	phosphodiesterase inhibitor		82.342	0.001
serotonin receptor antagonist		464.876	0.001	serotonin receptor antagonist		77.242	0.001
EGFR inhibitor	4	615.468	0.001	EGFR inhibitor	9	199.137	0.001
HMGCR inhibitor		627.009	0.001	HMGCR inhibitor		196.891	0.001
NFkB pathway inhibitor		645.583	0.001	NFkB pathway inhibitor		221.902	0.001
adrenergic receptor agonist		592.643	0.001	adrenergic receptor agonist		208.717	0.001
adrenergic receptor inhibitor		592.216	0.001	adrenergic receptor inhibitor		200.890	0.001
calcium channel blocker		625.183	0.001	calcium channel blocker		220.990	0.001
dopamine receptor agonist		631.829	0.001	dopamine receptor agonist		201.646	0.001
dopamine receptor inhibitor		618.807	0.001	dopamine receptor inhibitor		192.499	0.001
phosphodiesterase inhibitor		616.610	0.001	phosphodiesterase inhibitor		203.380	0.001
serotonin receptor antagonist		611.908	0.001	serotonin receptor antagonist		226.804	0.001

Extended Response Figures

Extended Response Figure 1: Correlation between the Wasserstein Distance of the training and ground-truth OOD data and the normalized metrics at Target and MOA levels. The x-axis represents the Wasserstein Distance between DeepProfiler embeddings of the training and testing datasets. The y-axis displays the normalized performance metrics. The black line indicates a linear regression between the x-axis and y-axis values.

Extended Response Figure 2: Correlation between the Wasserstein Distance of training gene count and inference OOD L1000 gene count and the normalized metrics at Target and MOA levels. The x-axis represents the Wasserstein Distance between the input gene counts of the training and testing datasets. The y-axis displays the normalized performance metrics. The black line indicates a linear regression between the x-axis and y-axis values.

Extended Response Figure 3: Comparison of the performance of MorphDiff methods and IMPA method on the generalization capabilities with respect to the distance of drug embedding / gene count. The CDRP Target_MOA dataset were divided into five bins based on their Wasserstein Distance from the training dataset. These Wasserstein Distances were computed using gene count and drug embeddings. All performance metrics were normalized according to the best performance achieved on each MOA.

Extended Response Figure 4: Visualization of the generated images from each baseline method as well as the ground-truth from the JUMP dataset. We randomly chose three ground-truth images for each genetic perturbation, and select generated images of baseline by SSIM[15] for visualization. MorphDiff (both modes) stands out as the top performer in terms of visual quality and detail across all perturbations. DMIT, IMPA and MDTv2 offer decent performance but generally fall short in detail and clarity. MorphNet generates reasonable images with some detail, but lacks diversity, and the detail is inconsistent with the ground truth. DRIT++, StarGANv1 and VQGAN exhibit varying levels of quality, often resulting in less clear images. The visual quality can be inconsistent with ground truth, with some images appearing blurry. This figure visualizes the results for CARD11, CDKN11 and DVL1 genetic perturbations.

Extended Response Figure 5: (cont'd) Visualization of the generated images from each baseline method as well as the ground-truth from the JUMP dataset. We randomly chose three ground-truth images for each genetic perturbation, and select generated images of baseline by SSIM[15] for visualization. MorphDiff (both modes) stands out as the top performer in terms of visual quality and detail across all perturbations. DMIT, IMPA and MDTv2 offer decent performance but generally fall short in detail and clarity. MorphNet generates reasonable images with some detail, but lacks diversity, and the detail is inconsistent with the ground truth. DRIT++, StarGANv1 and VQGAN exhibit varying levels of quality, often resulting in less clear images. The visual quality can be inconsistent with ground truth, with some images appearing blurry. This figure visualizes the results for MAP3K2, NOTCH2 and PIK3C genetic perturbations.

Extended Response Figure 6: (cont'd) Visualization of the generated images from each baseline method as well as the ground-truth from the JUMP dataset. We randomly chose three ground-truth images for each genetic perturbation, and select generated images of baseline by SSIM[15] for visualization. MorphDiff (both modes) stands out as the top performer in terms of visual quality and detail across all perturbations. DMIT, IMPA and MDTv2 offer decent performance but generally fall short in detail and clarity. MorphNet generates reasonable images with some detail, but lacks diversity, and the detail is inconsistent with the ground truth. DRIT++, StarGANv1 and VQGAN exhibit varying levels of quality, often resulting in less clear images. The visual quality can be inconsistent with ground truth, with some images appearing blurry. This figure visualizes the results for PRKCZ, PTEN and SMAD5 genetic perturbations.

Extended Response Figure 7: The performance changes for each method across the various metrics are evaluated following the transition from accepting three-channel images to five-channel images on the JUMP OOD dataset.

Extended Response Figure 8: The performance changes for each method across the various metrics are evaluated following the transition from accepting three-channel images to five-channel images on the CDRP OOD dataset.

Extended Response Figure 9: a. We performed Kolmogorov-Smirnov (K-S) tests between each pair of MorphDiff(I2I)'s and IMPA's generated CellProfiler features. We determine whether there is a significant difference between two groups for a given feature at a significance level of $p\text{-value} = 0.05$ (where $p\text{-value} > 0.05$ indicates no significant difference and $p\text{-value} < 0.05$ indicates a significant difference). The values in the heatmap represent the proportion of features for which there is no significant difference. b. We performed Maximum Mean Discrepancy tests between each pair of MorphDiff(I2I)'s and IMPA's generated DeepProfiler embeddings. The values represent the p -values of these tests, with p -values smaller than 0.05 indicating significant differences. For the Maximum Mean Discrepancy (MMD) tests [10], we determined if two groups come from different distributions by measuring their distance in a high-dimensional feature space. The key steps are: (1) transform data using an RBF kernel, (2) calculate the MMD statistic by comparing within-group similarities to between-group similarities, and (3) assess significance through permutation testing—repeatedly shuffling labels and recalculating MMD statistic to generate a null distribution. We can see that using both tests, the p -value between each IMPA group and MorphDiff(I2I) group is 0.01, highlighting that there is indeed significant difference between the two groups, while for each pair of IMPA groups and each pair of MorphDiff(I2I) groups, the p -value is high, indicating there is no significant difference. c. We calculated the Wasserstein Distance between each pair of MorphDiff(I2I)'s and IMPA's generated DeepProfiler embeddings. The distance between each IMPA sampled group and MorphDiff(I2I) group remains similar across different sampled groups, and is much larger than the distance within each method.